# Entorhinal grid-like codes and time-locked network dynamics track others navigating through space

Isabella C. Wagner [1,2,3] ✉, Luise P. Graichen[1], Boryana Todorova [1], Andre Lüttig [1], David B. Omer [4], Matthias Stangl [5] & Claus Lamm [1]

Navigating through crowded, dynamically changing environments requires the ability to keep track of other individuals. Grid cells in the entorhinal cortex are a central component of self-related navigation but whether they also track others' movement is unclear. Here, we propose that entorhinal grid-like codes make an essential contribution to socio-spatial navigation. Sixty human participants underwent functional magnetic resonance imaging (fMRI) while observing and re-tracing different paths of a demonstrator that navigated a virtual reality environment. Results revealed that grid-like codes in the entorhinal cortex tracked the other individual navigating through space. The activity of grid-like codes was time-locked to increases in co-activation and entorhinal-cortical connectivity that included the striatum, the hippocampus, parahippocampal and right posterior parietal cortices. Surprisingly, the grid-related effects during observation were stronger the worse participants performed when subsequently re-tracing the demonstrator's paths. Our findings suggests that network dynamics time-locked to entorhinal grid-cell-related activity might serve to distribute information about the location of others throughout the brain.

Maneuvering a crowded sidewalk or coordinating with one's team members to move towards the goal on a soccer field not only requires the planning of one's own movement through space, but depends upon the ability to track the location of conspecifics. Such socio-spatial navigation was recently tied to neuronal processes similar to self-related spatial navigation. For instance, work using intracranial recordings from the human medial temporal lobe revealed representations that coded for environmental boundaries when participants or others moved through space[1], and hippocampal social place cells were found to code for the location of others in bats[2] and rodents[3]. A central component of navigation is the integrity of grid cells in the entorhinal cortex that express periodic firing fields arranged along the vertices of regular hexagons, tessellating the environment and providing a spatial map[4,5]. In humans, it has been discussed that the neural firing signature of grid cell populations might relate to measures obtained non-invasively using fMRI (for a recent discussion, see[6]), in the form of so-called "grid-like codes"[7]. Moreover, these grid-like codes were shown to support spatial[7–12] as well as mental self-related navigation[13,14], but whether they also track others' movement (or the movement of non-social features) through space is unclear. Here, we propose that entorhinal grid cells (and related grid-like codes) make an essential contribution to (socio-)spatial navigation in humans.

Similar to the joint actions of different spatial cell types such as place and grid cells[15–18], flexible navigation relies on the orchestration of

[1]Social, Cognitive and Affective Neuroscience Unit, Department of Cognition, Emotion, and Methods in Psychology, Faculty of Psychology, University of Vienna, 1010 Vienna, Austria. [2]Vienna Cognitive Science Hub, University of Vienna, 1090 Vienna, Austria. [3]Centre for Microbiology and Environmental Systems Science, University of Vienna, 1030 Vienna, Austria. [4]Edmond and Lily Safra Center for Brain Sciences, The Hebrew University of Jerusalem, Givat Ram, 9190401 Jerusalem, Israel. [5]Department of Psychiatry and Biobehavioral Sciences, Jane and Terry Semel Institute for Neuroscience and Human Behavior, University of California Los Angeles, Los Angeles, CA 90095, USA. ✉e-mail: isabella.wagner@univie.ac.at

a set of regions including medial temporal, posterior-medial, parietal, striatal and prefrontal structures[19–23]. Beyond the hippocampal-entorhinal circuit, the parahippocampal and retrosplenial cortices are regarded as key players involved in the visuospatial processing of scenes and their orientation in the broader spatial environment[24–27]. The retrosplenial cortex specifically was proposed to translate information between allocentric (as supported by the hippocampus[28]) and ego-centric reference frames[23,24,26], interacting with the (right) posterior parietal cortex to guide the visuospatial coordination of locomotion[29–31] and the coding of spatial routes[32]. Similar aspects of navigation such as goal-directed behavior[19] and route following[33] were linked to dorsal striatal processes. Presumably, the dorsal striatum contributes to navigation via associative reinforcement and thus functions in parallel to the hippocampal system that rapidly encodes new experiences[19,34,35]. Interactions between these two systems appear mediated by the prefrontal cortex[34], which also regulates the top-down control of navigation via planning and goal tracking[20]. Since socio-spatial navigation also involves tracing others' movement through space, it likely engages additional brain regions concerned with biological motion processing, such as the posterior superior temporal sulcus[36,37]. Altogether, this suggests a complex interplay of multiple brain regions, but it remains elusive how they interact with the putative "spatial map" hosted by entorhinal grid cells. We thus asked whether the activity of entorhinal grid-like codes is coupled to functional connectivity changes with medial temporal, parietal, striatal, and prefrontal areas and whether such network dynamics may explain differences in navigation performance.

To tackle these questions, we built upon the design of previous animal work that investigated social place cells in bats[2] and translated it into the human context. Sixty healthy participants underwent functional magnetic resonance imaging (fMRI). Our modified navigation task projected them into a first-person perspective within a virtual reality (VR) environment while they were asked to observe different paths of a demonstrator (Fig. 1a, b). The paths had to be held in memory and needed to be re-traced after a short delay. Crucially, participants were positioned at a fixed viewpoint during observation, allowing us to dissociate potential grid-like codes between other- and self-related movements. Behavioral performance was quantified as cumulative distance error in virtual meters (vm) indicating the deviation from the demonstrator's paths (Fig. 1c). We reasoned that if underlying grid cells supported the tracking of others, we should observe increased grid-like codes in the entorhinal cortex during observation of the demonstrator moving through VR-space. Furthermore, we expected entorhinal grid-like codes to be dovetailed by entorhinal-cortical connectivity changes with regions typically involved in spatial navigation and visuospatial processing, altogether modulated by behavioral performance.

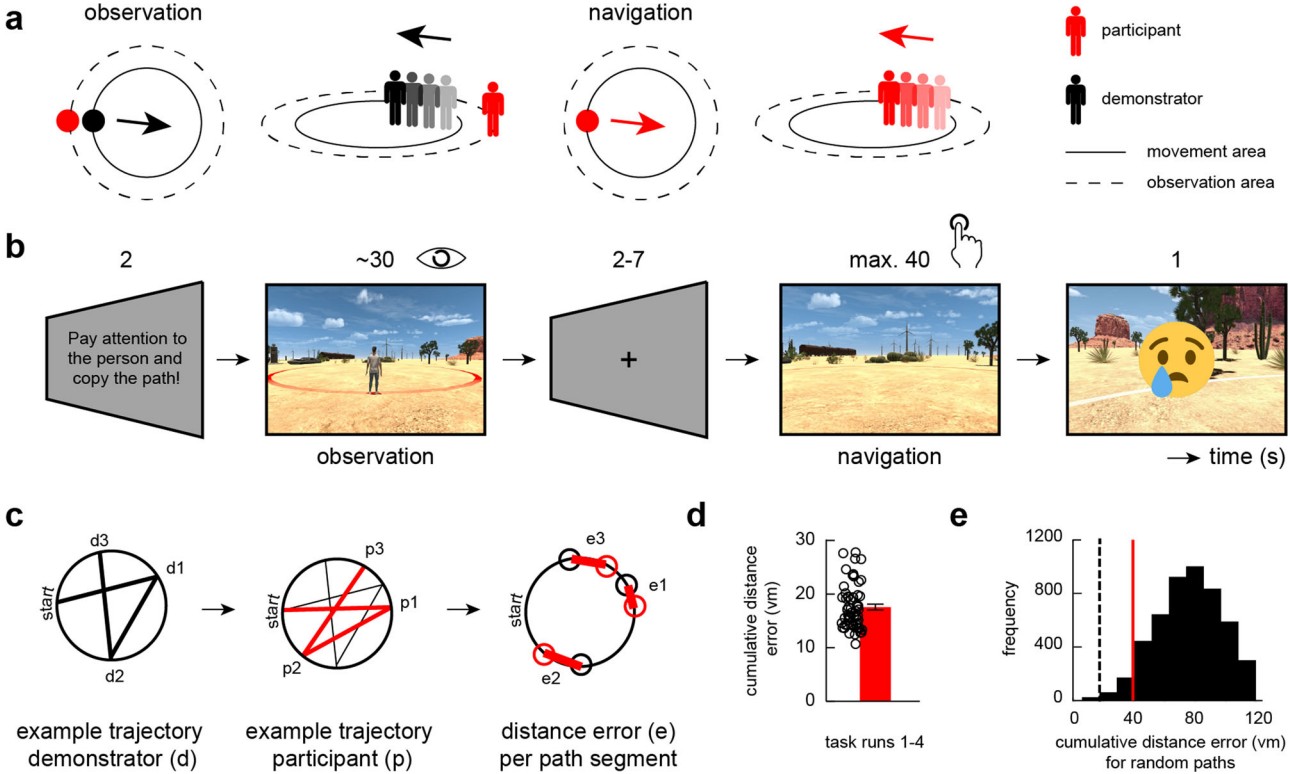

Fig. 1 | Modified navigation task. a The task projected participants into a first-person perspective within a virtual reality (VR) environment while they were asked to observe and subsequently re-trace the paths of a demonstrator. The movement area was marked with a red circle on the sandy desert plane and was surrounded by an observation area (solid and dashed circles in bird's eye and street views, respectively). During observation (left panel), participants were placed directly behind the demonstrator's starting point, were not able to move, and viewed the demonstrator walking through the circular arena (movement trajectory schematically indicated). During navigation (right panel), the demonstrator disappeared and the participant was projected onto the same starting position to re-trace the previously observed path. b Timeline of one example trial during the modified navigation task (s, seconds). A performance threshold of 20 virtual meters (vm) determined the feedback that participants received on each trial (i.e., happy emoji, cumulative distance error ≤ 20 vm; sad emoji, cumulative distance error > 20 vm). c Example path of demonstrator and participant obtained from observation and navigation periods, respectively. A trial comprised three random path segments that each started and ended at the edge of the movement area (i.e., participants observed/walked paths from one edge to the other and were not able to stop and turn within the movement area). We then calculated the distance error per path segment, yielding the cumulative distance error per trial. d Average cumulative distance error within the sample (N = 58). Error bars reflect the standard error of the mean, s.e.m. e Permutation distribution obtained from simulating the paths of a random agent 5000 times ("Methods"). Chance level was calculated as the 5th percentile of the distribution, yielding a value of 39.3 vm (red line). The dashed line indicates the value of the observed group mean (17.57 vm, shown in d). Source data are provided as a Source Data file.

## Results

### Participants are successfully able to re-trace the demonstrator's paths

Participants completed a single MRI session that started out with an initial task familiarization period, followed by four runs of the modified navigation task, each involving different paths. Across the four task runs, participants reached an average cumulative distance error of $17.57 \pm 0.54$ vm (mean ± standard error of the mean, s.e.m.; calculated as the average distance between the demonstrator's and the participant's endpoints across the three different segments of a given path, Fig. 1c, d; $N = 58$, 2 participants were excluded from this and all following analyses, see "Methods"). To determine whether participants performed better than chance, we simulated the performance of a random agent using permutation testing (see "Methods"). Navigation performance in our sample was significantly below the permutation-based chance level of 39.3 vm (one-sample $t$-test, $N = 58$, $t(57) = −39.6$, $p_{\text{two-tailed}} < 0.0001$; Fig. 1e). Moreover, performance was stable throughout the experiment (Supplementary Fig. S1), showing that participants were successfully able to re-trace the demonstrator's paths, and providing us with a solid basis for investigating grid-like codes during both observation and navigation.

We next took a closer look at navigation performance on individual segments of a given path (each path consisted of three segments). We found that across-run performance was significantly better when participants re-traced the first path segment ($6.86 \pm 0.16$ vm) compared to the second ($23.3 \pm 0.32$ vm) or third ($22.64 \pm 0.37$ vm; as indicated by the path segment-wise distance error; Supplementary Fig. S1). This pattern was evident in each of the four runs (Supplementary Fig. S1) and was likely due to the fact that the observer's viewpoint was directly fixed behind the demonstrator's starting position at the beginning of each trial.

### Observation is associated with increased activation in the hippocampus and striatum

We then turned towards the fMRI data and started out by investigating changes in brain activation when participants tracked and re-traced the paths of others, irrespective of potential grid-like codes. We hypothesized largely overlapping activation profiles during both observation and navigation periods, involving regions typically engaged in spatial navigation and visuospatial processing, such as medial temporal, posterior-medial, parietal, striatal, and prefrontal areas[19–23]. Additionally, we expected that observation of the demonstrator's paths would be associated with increased activation in brain areas associated with biological motion processing, including the posterior superior temporal sulcus[36,37].

While participants observed the demonstrator's paths a set of regions showed increased activation, including a large cluster centered on the bilateral hippocampus and adjacent structures of the medial temporal lobe (MTL), extending towards the anterior temporal pole, the caudate nucleus, pre- and post-central gyri, as well as the ventromedial prefrontal and posterior cingulate cortices (one-sample $t$-test, $N = 58$, contrast observation > navigation; Fig. 2a, Supplementary Table S1). Results further included stronger bilateral activation in lateral occipital areas, the fusiform gyrus, and the posterior superior temporal sulcus.

When participants re-traced the previously observed paths during navigation periods an activation profile with strongest response in the bilateral occipital cortex, including the fusiform gyrus and the right temporo-parietal junction emerged (one-sample $t$-test, $N = 58$, contrast navigation > observation; Fig. 2b, Supplementary Table S1). Activation was also increased in the bilateral caudate nucleus and putamen, the insula, as well as the adjacent inferior prefrontal cortex.

Since visual input was considerably different between observation and navigation conditions (stationary viewpoint vs. navigation), we also compared each of the conditions against the (fixation) baseline. Activation changes appeared largely similar for both conditions,

including increased activation in visual, parietal and lateral prefrontal regions (Fig. 2c, Supplementary Fig. S2, Supplementary Table S1). Notably, the general effect of navigation (contrast navigation > baseline) was also associated with increased activation in the hippocampus and entorhinal cortex.

Altogether, observing the demonstrator's paths rather than re-tracing them was associated with stronger brain activity in a set of regions that included the hippocampus and adjacent MTL structures, the striatum, anterior and posterior midline regions, as well as superior and inferior temporal areas.

### Increased activation in the right posterior parietal cortex during observation is associated with better performance

To investigate whether brain activation during observation and navigation scaled with individual performance, we went on to test the cross-participant relationship between whole-brain activity during the modified navigation task and the average cumulative distance error when re-tracing the demonstrator's paths. We performed this analysis in two steps to be able to clearly interpret the direction of potential effects, first focusing on observation and then on navigation periods (each contrasted against baseline). Results showed an activation increase within the right posterior parietal cortex ($x = 44$, $y = −78$, $z = 24$, $z$-value = 4.68, 297 voxels) during observation that negatively correlated with the individual cumulative distance error [linear regression, $N = 58$, contrast observation > baseline, cumulative distance error added as a covariate of interest; $p < 0.05$ family-wise-error (FWE) corrected at cluster level using a cluster-defining threshold of $p < 0.001$, cluster size = 80 voxels; Fig. 2d]. In other words, stronger activation in this region was coupled to better subsequent performance as participants observed the demonstrator's paths (and see Supplementary Information for additional analysis). We did not find a significant relationship between behavioral performance and brain activation during navigation periods.

### Entorhinal grid-like codes when observing the demonstrator's paths

Central to our analyses was the question whether grid-like codes supported the tracking of others. We were motivated by findings of place cells in animals that were recently shown to signal the location of conspecifics[2,3], as well as by previous work probing entorhinal grid cell activity (or grid-like codes) during spatial (and mental) self-navigation in both animals[4] and humans[7–14]. Hence, we expected to find a significant hexadirectional modulation (i.e., a 6-fold periodicity) of the fMRI signal in the entorhinal cortex during observation and possibly also during navigation periods.

To test this, we split the fMRI data of each task run into independent sets and estimated/tested individual grid orientations using a 12-fold cross-validation regime (Fig. 3a–c; for a detailed description, please see "Methods"). We used 11 trials (each trial included an observation and a navigation period) to estimate individual grid orientations based on the demonstrator's (or the participant's) movement trajectories through VR-space using a General Linear Model (GLM1; note that grid orientations were estimated separately for observation and navigation periods). The grid orientations' fit was then tested on the path segments of the remaining trial, which served to quantify the magnitude of grid-like codes in the left-out data set (GLM2). This procedure was repeated for each cross-validation fold and the resulting grid magnitudes were averaged across the different iterations. Note that grid analyses were based on a reduced participant sample (entorhinal cortex: automatic segmentation, $N = 49$; manual segmentation, $N = 51$; control regions: $N = 58$; see "Methods").

We found significant grid-like codes in the entorhinal cortex as participants were observing the demonstrator's paths. This effect was only present for the 6-fold symmetrical model [mean grid magnitude (arbitrary units, arb. units) ± s.e.m., $0.158 \pm 0.06$; one-sample $t$-test,

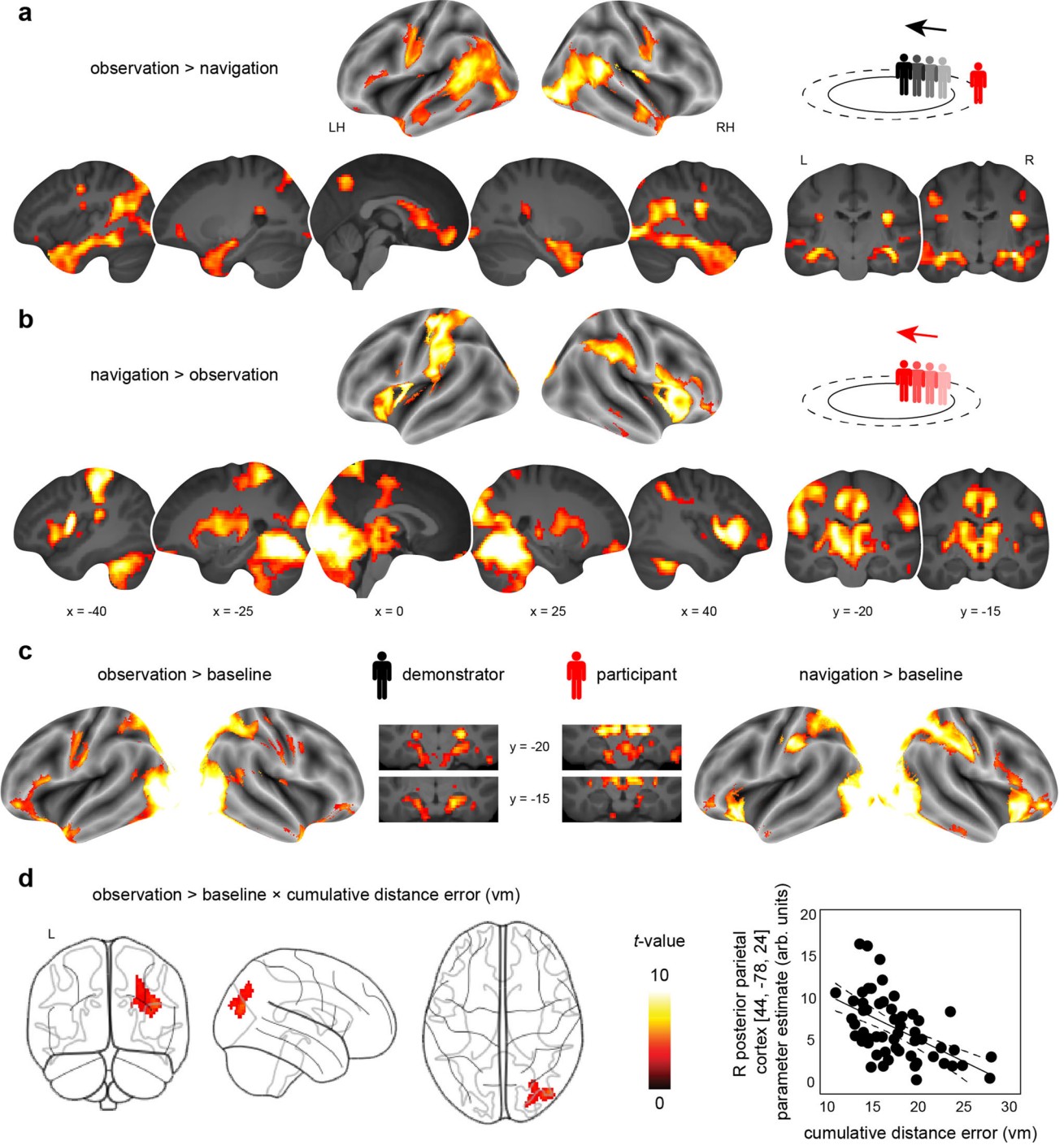

**Fig. 2 | Brain activation profiles during observation and navigation, and association with performance.** Brain activation **a** during observation (compared to navigation) periods, and **b** vice versa (separate one-sample *t*-tests, $N = 58$; Supplementary Table S1). **c** Brain activation changes when contrasting each condition against the implicit (fixation) baseline (separate one-sample *t*-tests, $N = 58$; see also Supplementary Fig. S2, Supplementary Table S1). **d** Brain activation during observation (compared to implicit baseline) and association with performance across participants (indexed by the average cumulative distance error in virtual meters,

vm; linear regression, $N = 58$). The scatter plot shows the relationship between the change in parameter estimates (arbitrary units, arb. units) extracted from the significant cluster and the cumulative distance error (vm; Supplementary Information). Given the clear inferential circularity, we would like to highlight that this plot serves visualization purposes only, solely illustrating the direction of the brain-behavior relationship. All results are shown at $p < 0.05$ FWE-corrected at cluster level (cluster-defining threshold of $p < 0.001$). L, Left; R, Right; H, Hemisphere. Source data are provided as a Source Data file.

$N = 47$ (excluding 2 outliers); $t(46) = 2.73$, $d = 0.4$, 95% CI = [0.04, 0.3], $p_{\text{one-tailed}} = 0.005$, Bonferroni-corrected for multiple comparisons using a threshold of $\alpha_{\text{Bonferroni}} = 0.05$ / a total of 6 entorhinal and control ROIs $= 0.008$; Fig. 3d; results for this and all following analyses of this section remained stable when using the full data set of $N = 49$ and when using a robust test framework, see also Supplementary Table S2]. To establish

the reliability of results obtained from the 6-fold symmetrical model, we repeated the analysis with different control symmetries that were not expected to yield significantly increased grid magnitudes. Different symmetrical models such as 5-fold or 7-fold signal periodicities did not yield significant results [separate one-sample *t*-tests; 5-fold: $N = 42$ (excluding 7 outliers); $p_{\text{one-tailed}} = 0.245$; 7-fold: $N = 48$

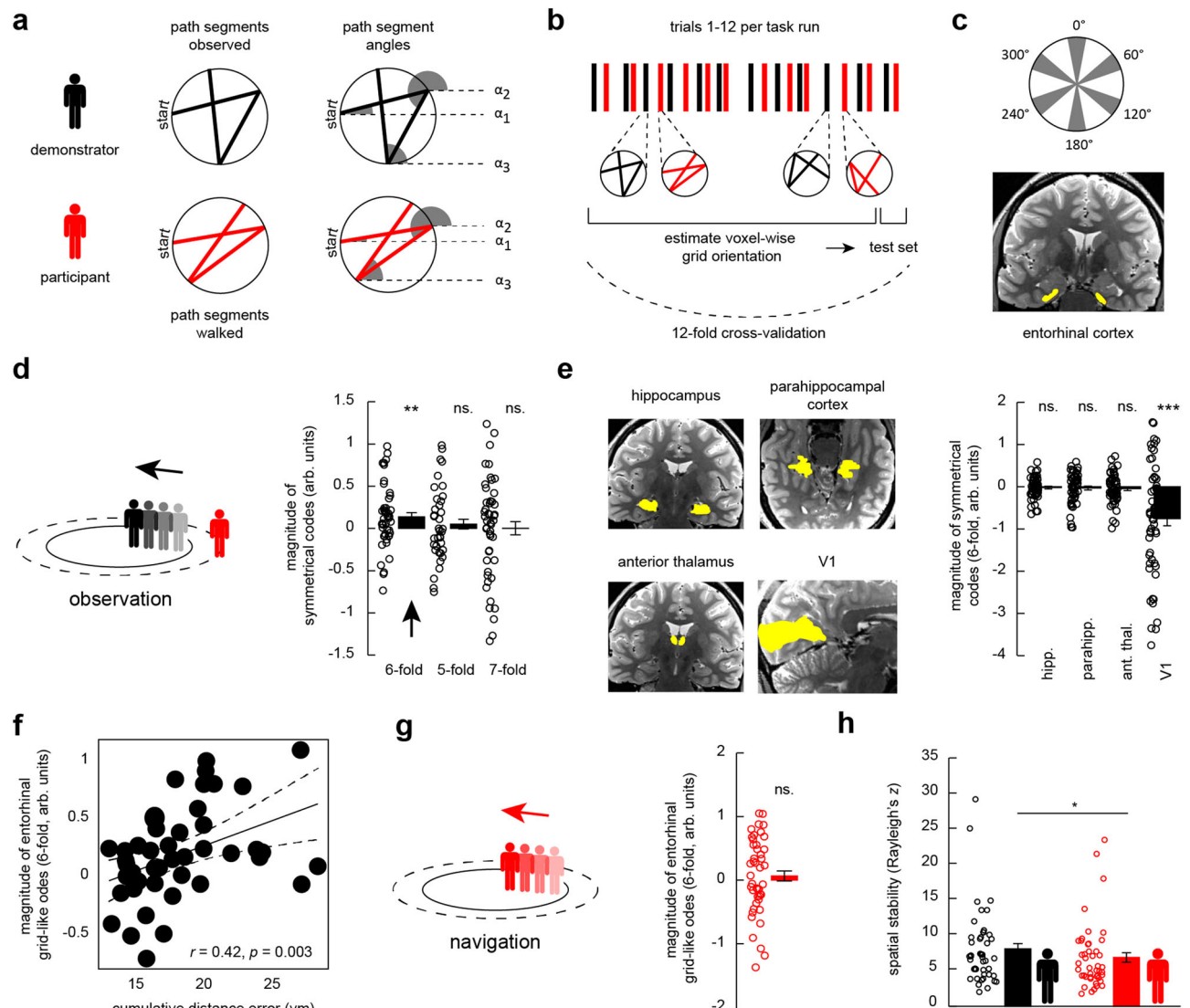

**Fig. 3 | Grid-like codes during observation. a** We calculated path-wise trajectory angles $\alpha_1$–$\alpha_3$ referenced to an arbitrary point on the VR desert plane (dashed lines, angles indicated in gray). **b** Schematic of the trial composition within a task run, showing observation (black) and navigation (red) periods that consisted of three path segments each (1 trial = 1 observation period followed by 1 navigation period). Grid orientations were estimated and tested by employing a 12-fold cross-validation (CV) regime. **c** Schematic of angular differences in 360°-VR-space. We expected increased entorhinal cortex signal for directions observed/walked that were aligned to individual grid orientations (in gray). Entorhinal cortex region-of-interest (ROI, in yellow) projected onto the T2-weighted structural scan of one participant. **d** Magnitude of symmetrical codes (separate one-sample $t$-tests, one-tailed, multiple comparisons corrected; 6-fold: $N = 47$, $p = 0.005$; 5-fold: $N = 42$, $p = 0.245$; 7-fold: $N = 48$, $p = 0.489$) in the entorhinal cortex during observation periods

(arbitrary units, arb. units.; Supplementary Information). **e** Magnitude of observation-related symmetrical codes (sixfold periodicity) in the control ROIs (separate one-sample $t$-tests, one-tailed, multiple comparisons corrected; hippocampus: $N = 56$, $p = 0.271$; parahippocampal cortex: $N = 55$, $p = 0.264$; anterior thalamus: $N = 56$, $p = 0.159$; V1: $N = 58$, $p < 0.0001$). **f** Pearson-correlation (two-tailed, $N = 47$, $p = 0.003$) between grid magnitudes (6-fold periodicity) and the cumulative distance error across participants (vm; Supplementary Information). **g** No significant entorhinal grid-like codes (6-fold periodicity) during navigation periods (one-sample $t$-test, one-tailed, $N = 45$, $p = 0.233$; Supplementary Information). **h** Spatial stability of entorhinal grid-like codes (paired-sample $t$-test, two-tailed, $N = 44$, $p = 0.048$). ***$p < 0.0001$, **$p < 0.01$, *$p < 0.05$; error bars reflect the standard error of the mean, s.e.m. Source data are provided as a Source Data file.

(excluding 1 outlier), $p_{\text{one-tailed}} = 0.489$; Fig. 3d]. The same pattern of findings emerged when we delineated the entorhinal cortex using manual segmentation (results in Supplementary Information and in Supplementary Fig. S3) and when using a different, less nested cross-validation regime that split each task run into four data parts (Supplementary Information). Grid orientations also partly generalized across separate task runs (Supplementary Information). Exploratory follow-up analysis showed that grid-like codes during observation appeared predominantly left lateralized (Supplementary Information). Additionally, prior work reported 4-fold modulation of entorhinal cortex signal during self-related navigation[11]. We thus tested whether such an effect was also present during observation but did not find evidence for a

4-fold signal periodicity in the entorhinal cortex [$N = 47$ (2 outliers excluded), $0.07 \pm 0.05$, $p_{\text{one-tailed}} = 0.082$, Supplementary Fig. S4; $N = 49$ (full sample), $0.07 \pm 0.06$, $p_{\text{one-tailed}} = 0.113$].

To test whether grid-like codes were also present in other areas, we chose several control ROIs known to be involved in spatial navigation and visuo-spatial processing but for which no grid-like codes have been reported so far (including the hippocampus, parahippocampal cortex, anterior thalamus, and the primary visual cortex/V1; Fig. 3e, see "Methods"). There were no significantly increased grid-like codes in any of the control ROIs [separate one-sample $t$-tests; hippocampus: $N = 56$ (excluding 2 outliers), $p_{\text{one-tailed}} = 0.271$; parahippocampal cortex: $N = 55$ (excluding 3 outliers), $p_{\text{one-tailed}} = 0.264$;

anterior thalamus: $N = 56$ (excluding 2 outliers), $p_{one\text{-}tailed} = 0.159$; V1: $N = 58$ (full data set, no outliers excluded), significantly negative grid magnitude, $d = -0.59$, 95% CI = [−1.1, −0.4], $p_{one\text{-}tailed} < 0.0001$; Fig. 3e; and see Supplementary Table S2 for virtually identical results when including the full data set]. We further explored the finding of negative grid-like codes in V1 during observation with additional analyses. First, results remained stable when using a different, less-nested cross-validation regime (mitigating the potential effect of specific directions that could have affected grid magnitudes disproportionally, Supplementary Information). Second, we found that V1 activation levels appeared increased for observed directions that were misaligned with the individual V1 grid orientation (0 modulo 30°) compared to aligned directions (0 modulo 60°). This effect was not driven by single directional bins but appeared relatively consistent across the range of all misaligned directions (Supplementary Fig. S5). We provide an in-depth discussion of this finding in our supplementary materials (Supplementary Information).

In summary, we found significantly increased grid-like signals in the entorhinal cortex (as well as significantly decreased grid-like codes in V1) when participants were observing and putatively encoding the demonstrator's paths.

### Entorhinal grid-like codes during observation are stronger at lower performance

Previous studies highlighted a significant association between entorhinal grid-like codes during navigation and behavioral performance[7–10,38]. We thus reasoned that variations in observation-based grid-like codes (as indexed by grid magnitudes) should be coupled to individual differences in the average cumulative distance error during navigation periods. Interestingly, results yielded a significant positive association such that increased observation-based grid magnitudes were coupled to larger cumulative distance errors ($N = 47$, same sample as for main analysis above: $r_{Pearson} = 0.42$, 95% CI = [0.15, 0.63], $p_{two\text{-}tailed} = 0.003$; Fig. 3f).

We performed several control analyses to validate the brain-behavior relationship. First, we could show that the results did not stem from specific path patterns, such as potentially longer paths during low- compared to high-performance trials (the paths of low-performance trials could have crossed the center of the movement area more often, yielding a longer average path length and thus perhaps a stronger grid-like signal, but this was not the case; Supplementary Information). Second, we repeated the correlation analysis with a newly-defined performance measure, taking into account that chance performance for re-tracing a given path correctly differed for different endpoints along the circumference of the movement area. Using this normalized accuracy measure[39], results remained virtually identical (results in Supplementary Information and in Supplementary Fig. S6). Altogether, these analyses corroborated our result of stronger grid-like codes the worse participants performed.

### No significant entorhinal grid-like codes when re-tracing the demonstrator's paths

In analogy to previous work that reported entorhinal grid cell activity, or grid-like codes, during spatial (and mental) self-navigation[7–14], we expected to replicate this finding in our data set and hypothesized significant grid-like codes in the entorhinal cortex as participants were re-tracing the demonstrator's paths during navigation periods (see "Methods"). Surprisingly, results did not show significantly increased grid-like codes in the entorhinal cortex [separate one-sample $t$-tests for all following analyses; $N = 45$ (excluding 4 outliers), $p_{one\text{-}tailed} = 0.233$; Fig. 3g; as above, results for this and all following analyses of this section remained the same when using the full data set and when using a robust test framework, Supplementary Table S3], also not when manually delineating the region (Supplementary Information), when using a reduced data set (Supplementary Information), when using a

different cross-validation regime (Supplementary Information), or when testing for grid-like codes in the left and right hemisphere separately (but note that grid magnitudes appeared numerically increased in the right hemisphere and were significantly negative in the left hemisphere; Supplementary Information). When investigating grid-like codes in the control ROIs, we found significantly increased grid magnitudes in the anterior thalamus during navigation but not in any of the other regions (results in Supplementary Information and in Supplementary Fig. S3). As above, we also tested for a potential 4-fold signal periodicity since previous work demonstrated such an effect during navigation[11]. Indeed, we found that the 4-fold symmetrical model yielded significantly increased navigation-related responses in the entorhinal cortex [$N = 43$ (6 outliers excluded), mean grid magnitude (arbitrary units, arb. units) ± s.e.m., 0.1 ± 0.05, $t(42) = 1.83$, Cohen's $d = 0.28$, 95% CI = [−0.01, 0.21], $p_{one\text{-}tailed} = 0.037$, Supplementary Fig. S4; $N = 49$ (full sample), 0.15 ± 0.13, $p_{one\text{-}tailed} = 0.137$].

We also explored whether grid orientations during observation periods served as spatial reference frames when participants re-traced the demonstrator's paths during navigation. If this was the case, we should find matching grid orientations between the two conditions, leading to significantly increased grid magnitudes when testing grid orientations obtained from observation on navigation periods. However, results did not yield any significant results, indicating grid-like codes during observation but neither the same nor differently-oriented grid-like codes during navigation periods (results in Supplementary Information and in Supplementary Fig. S7).

### Control analyses: unbiased estimation of grid orientations

We performed several control analyses to verify that our results were actually related to entorhinal grid-like codes that represented the demonstrator's path in space. For instance, the estimation of individual grid orientations can be biased by the distance walked along different directions in space (in other words, it would not be possible to estimate individual grid orientations for a specific direction if the participants never walked in that direction). To circumvent this issue, we designed our task such that each segment of an individual path was oriented along one of 36 directions that could be divided into 6 directional bins, spanning the 360°-VR-space with 10° angular resolution[7]. We then generated the demonstrator's paths (i.e., the entire trajectories that consisted of three path segments each) to maximize the distance walked in each of the directional bins and could thus avoid any biases in the estimation of individual grid orientations during observation periods (see "Methods", Supplementary Fig. S8).

Since participants were not always able to perfectly re-trace the demonstrator's paths, we also checked whether there was a difference in the participant's distance walked across the 6 directional bins. As above, there was no significant difference in the average distance walked suggesting that we were able to avoid any biases in grid orientation estimation during navigation periods (Supplementary Fig. S8).

### Control analyses: Higher spatial stability of entorhinal grid-like codes during observation

Grid-like codes might be affected by variations in spatial and temporal signal stability[8]. In the case of spatial instability, estimated grid orientations are assumed to vary across the different voxels within the ROI, resulting in more variable mean grid orientations and an overall decrease in grid magnitude. To test whether differences in spatial stability between observation and navigation conditions contributed to our results, we calculated individual voxel-wise grid orientations within the bilateral entorhinal cortex ROI (see "Methods").

Results revealed that the spatial stability [quantified as Rayleigh's $z$ that describes non-uniformity of circular data; i.e., data clustering towards a specific (grid) orientation] was significantly higher for grid-like codes during observation compared to navigation periods [paired-

sample *t*-test, *N* = 44, participant sample from which both observation- and navigation-based grid values were available, *t*(43) = 2.04, *d* = 0.28, 95% CI = [0.01, 2.4], $p_{two\text{-}tailed}$ = 0.048; Fig. 3h]. Voxel-wise entorhinal grid orientations thus varied more when participants re-traced the demonstrator's paths, potentially contributing to the lack of significant grid magnitudes during navigation.

This led us to reason that it might still be possible to obtain significant grid-like coding during navigation when performing the analysis separately for each voxel (i.e., when estimating and testing grid orientations on a voxel-by-voxel basis rather than averaging across the entorhinal cortex ROI). We thus repeated the abovementioned analysis of representational stability for which we had partitioned each task run into data halves and adopted the approach to quantify grid magnitudes during navigation on a voxel-by-voxel basis (that is, we repeated GLM2 for each voxel within the bilateral entorhinal cortex ROI), and then averaged across grid magnitudes to obtain a summary score. However, in line with all previous analyses, results did not yield significant grid-like coding during navigation [one-sample *t*-test, *N* = 46 (3 outliers excluded), mean grid magnitude (arbitrary units, arb. units) ± s.e.m., −0.02 ± 0.03, $p_{one\text{-}tailed}$ = 0.199; *N* = 49 (full sample), 0.09 ± 0.08, $p_{one\text{-}tailed}$ = 0.14].

Regarding temporal stability, there was no significant difference between voxel-wise entorhinal grid magnitudes of observation and navigation periods over time (i.e., no difference in grid orientations in the estimation and test data sets, quantified as % voxels with same/different orientations across time; see "Methods", Supplementary Fig. S8).

### Control analyses: no effect of neural adaption during navigation

Another possible reason for the lack of significant entorhinal grid-like codes during navigation could be neural adaption (or "repetition suppression") during navigation. Neural adaption describes the phenomenon that the neural signal is reduced upon repeated presentation of the same stimulus or when associated stimuli are presented in succession (compared to the sequential presentation of two unrelated stimuli), potentially reflecting the presence of a "memory trace" (for a review, see ref. [40]). In the current task, observation periods were followed by navigation periods and neural adaption might thus indicate an association (or similarity) between the periods. To be able to compare between navigation periods with high/low similarity to the previous observation period, we took into account individual performance: we expected that the observation-navigation similarity should be stronger for trials during which participants performed well (compared to trials during which participants performed less well; i.e., smaller vs. larger cumulative distance error, respectively). In other words, during high-performance trials, participants should have encoded the path trajectory well during observation, indicated by stronger neural adaption during the subsequent navigation period.

However, we did not find evidence for neural adaption (results in Supplementary Information and in Supplementary Fig. S9), suggesting that this phenomenon did not impact our ability to detect grid-like codes during navigation periods. Additionally, it could be the case that overall entorhinal activation levels were stronger during observation compared to navigation periods, leading to significantly increased grid-like codes during observation (and no significant findings during navigation). Analysis showed that we could rule out this potential explanation as overall activation levels appeared comparable between the conditions (results in Supplementary Information and in Supplementary Fig. S9).

### Control analyses: no effects of eye movements on grid-like codes

To account for the potential impact of eye movements on grid-like codes[41–44], we recorded eye gaze during the modified navigation task and leveraged the saccade directions of each participant during observation and navigation periods (note that this analysis included a subset of 37 participants from which both eye-tracking and entorhinal fMRI data were available). We repeated our initial grid analysis but now modeled eye gaze directions (i.e., the angle between successive saccades with respect to an arbitrary reference point on the computer screen) instead of movement trajectories in VR-space (Supplementary Information). Findings did not reveal a significant increase in entorhinal saccade-based grid magnitudes when testing for a 6-fold symmetrical model during observation (or navigation) periods (separate one-sample *t*-tests, all $p_{two\text{-}tailed}$ > 0.05; results in Supplementary Information and in Supplementary Fig. S8). Additionally, we tested whether the magnitude of grid-like codes in the entorhinal cortex during observation scaled with the average number of saccades that participants performed. Correlation analysis showed that this was not the case [*N* = 37 (same sample as above); $p_{two\text{-}tailed}$ = 0.427], altogether suggesting that our result of significant grid-like codes in the entorhinal cortex when observing the demonstrator's paths was based on spatial information rather than the number or direction of saccadic eye movements.

We also verified whether participants were actually following the demonstrator with their eye gaze during observation by defining an area-of-interest (AOI) for each path segment. AOIs were defined by the two-dimensional coordinates of a given path trajectory on the computer screen as well as by the demonstrator's height, and we calculated the percentage of eye movements that were located within the AOI boundaries. The majority of eye movements were located within AOIs, emphasizing that participants' viewing behavior was related to the observation of the demonstrator (Supplementary Information).

### Striatal activation increases are time-locked to entorhinal grid-like codes during observation and negatively scale with performance

We next asked whether the activity of entorhinal grid-like codes during observation was paralleled by activation changes of regions typically involved in spatial navigation and visuospatial processing (e.g., medial temporal, parietal, striatal, and prefrontal structures[19–23]), and whether this would be linked to behavioral performance. Similar to above, we modeled observation and navigation periods based on the individual path segments but now added the participant- and path segment-specific grid magnitudes (obtained from the previous grid code analysis) as parametric modulators (GLM3; grid magnitudes were obtained from GLM2 and were then averaged across the 12 cross-validation folds, Fig. 4a, see "Methods"). We then performed group comparisons to estimate voxel-wise activation changes that scaled with participant- and path segment-specific grid magnitudes (i.e., contrasting the parametric modulation regressor that captured fluctuations in entorhinal grid magnitude during observation/navigation against the implicit baseline).

During observation periods, we found significantly increased activation in the right putamen and caudate nucleus (*x* = 30, *y* = 8, *z* = −6, *z*-value = 4.96, 102 voxels) that positively scaled with individual entorhinal grid magnitudes as participants were observing the demonstrator's paths (one-sample *t*-test, *N* = 47, same participant sample as during initial grid analysis, contrast parametric modulation through entorhinal grid magnitude during observation > baseline, *p* < 0.05 FWE-corrected at cluster level using a cluster-defining threshold of *p* < 0.001, cluster size = 68 voxels; Fig. 4b). Put differently, striatal activation was stronger when participants observed the demonstrator walking aligned with their individual entorhinal grid orientation. There were no significant activation changes during navigation periods.

We next explored whether such grid-related activation increases were associated with individual variations in performance across participants. In line with our result of increased observation-based grid-like codes in the entorhinal cortex at lower performance (Fig. 3f), we found significantly increased activation in the bilateral putamen and caudate nucleus (left: *x* = −18, *y* = −2, *z* = 12, *z*-value = 3.72, 222 voxels; right: *x* = 14, *y* = 4, *z* = 18, *z*-value = 3.68, 245 voxels), as well as in the right orbito-frontal cortex (*x* = 32, *y* = 54, *z* = −9, *z*-value = 4.18, 197 voxels) that was

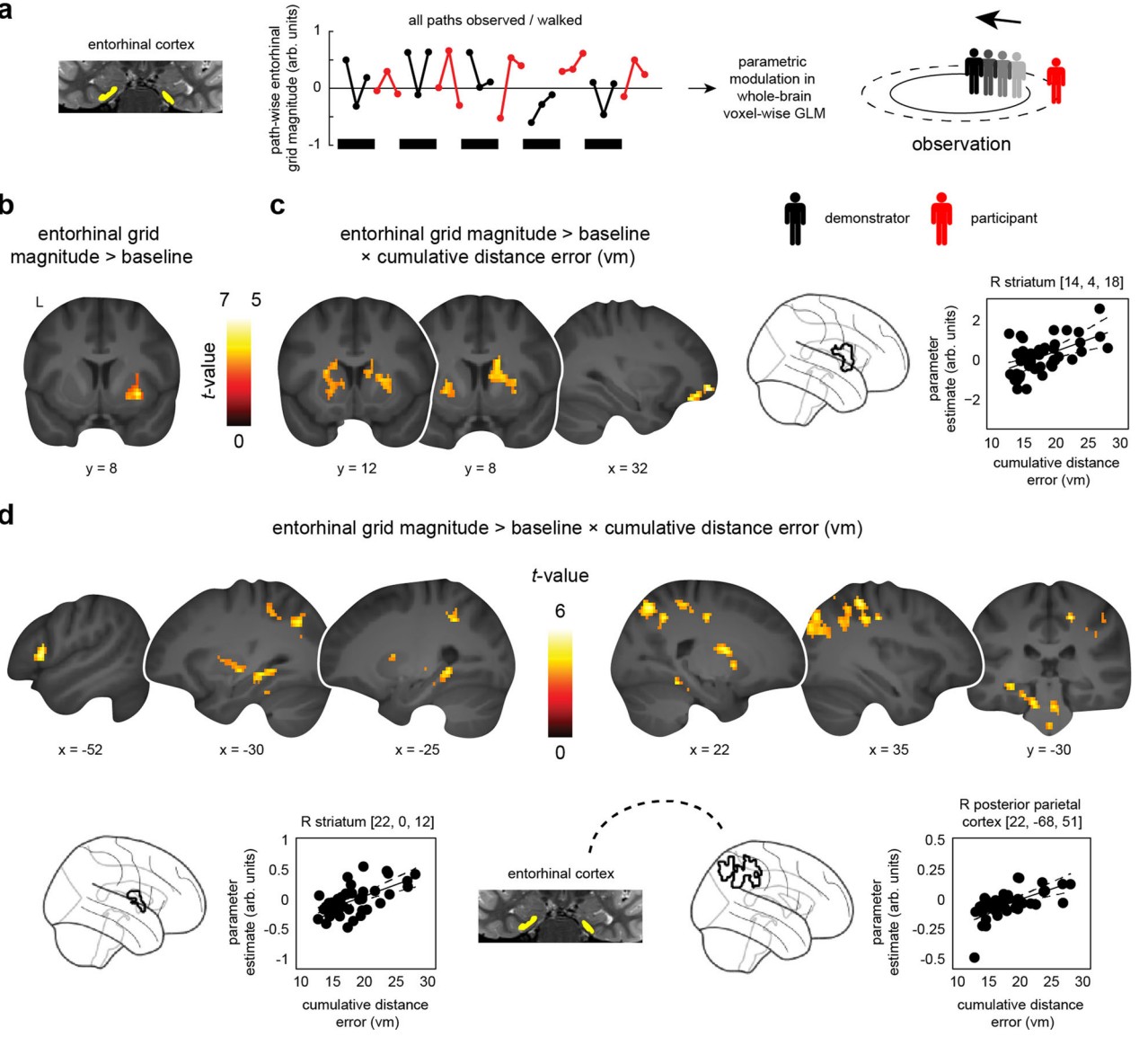

**Fig. 4 | Brain activation profiles and connectivity changes time-locked to entorhinal grid-like codes during observation, and association with performance.** **a** Grid orientations and magnitudes were estimated and tested on independent data sets (GLM1 and GLM2), focusing on data within the entorhinal cortex (yellow). This yielded participant- and path segment-specific grid magnitudes for each of the 36 path segments per condition and task run. Observation periods are indicated in black (navigation periods in red, 1 example task run). Segment-wise grid magnitudes were then used to parametrically modulate observation/navigation events in a third, independent analysis (GLM3) that tested for voxel-wise brain activation/connectivity changes time-locked to fluctuations in entorhinal grid magnitude during observation. **b** Increased brain activation during observation periods (compared to the implicit baseline), positively scaling with entorhinal grid magnitudes (one-sample $t$-test, $N = 47$). **c** Same contrast associated with behavioral

performance (linear regression, $N = 47$). **d** Entorhinal connectivity changes during observation periods, time-locked to increased entorhinal grid magnitudes and associated with behavioral performance (linear regression, $N = 47$; Supplementary Table S4). **c, d** The scatter plots show the relationships between the changes in parameter estimates (arbitrary units, arb. units) extracted from the respective clusters (black outlines in glass brains) and the cumulative distance error (vm). Given the clear inferential circularity, we would like to highlight that these plots serve visualization purposes only, solely illustrating the direction of the brain-behavior relationship. Connectivity is schematically indicated through dashed line. Results are shown at $p < 0.05$ FWE-corrected at cluster level [cluster-defining threshold of (**a**, **d**) $p < 0.001$ or (**c**) $p < 0.005$]. L, Left; R, Right. Source data are provided as a Source Data file.

time-locked to fluctuations in entorhinal grid magnitude (i.e., stronger when participants observed the demonstrator walking aligned with their individual entorhinal grid orientation) and larger the worse participants performed when re-tracing the demonstrator's paths (linear regression, $N = 47$, same participant sample as during initial grid analysis, contrast parametric modulation through entorhinal grid magnitude during observation > baseline, cumulative distance error across all paths added as a covariate of interest; please note that we here deviated from our a priori cluster-defining threshold of $p < 0.001$, showing results that reached significance only at a more lenient statistical threshold, $p < 0.05$ FWE-corrected at cluster level using a cluster-defining threshold of

$p < 0.005$, cluster size = 157 voxels; Fig. 4c). To restate, the activation in these regions scaled more strongly with entorhinal grid-like codes during observation the worse participants performed in re-tracing the demonstrator's paths thereafter. There were no significant activation changes related to better navigation performance.

**Entorhinal connectivity with striatum and right posterior parietal cortex is time-locked to grid-like codes during observation and negatively scales with performance**
So far, we presented evidence that grid-like codes in the entorhinal cortex were significantly enhanced during observation periods and

that striatal activation was time-locked to such entorhinal signals. These results were stronger the less accurately participants performed when re-tracing the demonstrator's paths during navigation periods. To tackle entorhinal-cortical interactions[45] potentially underlying socio-spatial navigation, we tested functional connectivity changes during observation and navigation. More specifically, we asked whether entorhinal grid-like codes would trigger changes in network connectivity whenever participants would observe the demonstrator walking aligned with their individual grid orientation, and whether such changes would be associated with individual differences in performance.

We tested this using the above model (GLM3) and performed generalized psychophysiological interaction analysis (gPPI, see "Methods"). In brief, we took the anatomical boundaries of the bilateral entorhinal cortex as a seed, calculated its whole-brain connectivity during observation (and navigation) periods (i.e., contrasting the parametric modulation regressor that captured fluctuations time-locked to participant- and path segment-specific entorhinal grid magnitudes with the implicit baseline), and tested whether changes in connectivity varied as a function of the average cumulative distance error per participant.

Results showed that functional connectivity between the entorhinal cortex with the hippocampus and parahippocampal cortex, as well as with the inferior temporal and left lateral prefrontal cortex positively scaled with entorhinal grid magnitudes and that this relationship was stronger at lower performance [linear regression; $N = 47$ (same participant sample as during initial grid analysis), contrast parametric modulation through entorhinal grid magnitude during observation > baseline, cumulative distance error across all paths added as a covariate of interest, $p < 0.05$ FWE-corrected at cluster level using a cluster-defining threshold of $p < 0.001$, cluster size = 61 voxels; Fig. 4d, Supplementary Table S4]. This connectivity profile further included enhanced entorhinal coupling with the bilateral striatum and the right posterior parietal cortex (see also Supplementary Fig. S10 showing the overlap with the right posterior parietal cluster from the grid-independent activation analysis that we reported in Fig. 2d). Again, these findings appeared specific to lower performance (there were no significant connectivity changes related to better performance) and to observation periods (there were no significant effects during navigation). General connectivity changes (independent of grid-like coding) further showed that entorhinal-cortical coupling was increased during both observation and navigation periods (but stronger during observation; Supplementary Fig. S11, Supplementary Table S5).

Taken together, functional connectivity between the entorhinal cortex, striatum, right posterior parietal cortex and a wide-spread set of cortical regions was time-locked to fluctuations in entorhinal grid magnitudes when participants were tracking the demonstrator's paths during observation periods and this relationship was stronger at lower individual performance.

## Discussion

In the current study, we investigated whether grid-like codes in the human entorhinal cortex supported participants as they tracked and subsequently re-traced the paths of a virtual demonstrator. Our key findings advance current knowledge in several ways: We found that entorhinal grid-like codes supported the tracking of another individual's movement through space. Crucially, these signals were decoupled from self-movement as participants' viewpoint remained stationary while observing the demonstrator. Fluctuations in grid magnitudes were associated with the co-activation of striatal regions during observation and these results appeared stronger the less accurate participants performed when subsequently re-tracing the demonstrator's paths. The profile of co-activation was paralleled by entorhinal connectivity increases with the striatum and the hippocampus, parahippocampal, right posterior parietal and lateral

prefrontal cortices when participants observed the demonstrator walking aligned with their internal grid orientations, and this pattern was again more pronounced the worse participants performed. The study yielded several surprising findings as well: we found significantly negative grid-like codes in V1 during observation and we did not detect significant grid-like codes during navigation periods. Overall, our findings are the first to demonstrate that grid-like signaling in the human entorhinal cortex is related to the spatial tracking of others, that it is linked to network dynamics, and modulated by individual task performance.

The main goal of our research was to probe whether grid-like codes in the entorhinal cortex supported the tracking of others, potentially yielding insights into human socio-spatial navigation. Confirming our hypothesis, results showed a significant increase in grid magnitudes (thought of as a proxy for putative grid-cell-related activity) while participants were observing the demonstrator's paths from a stationary viewpoint (Fig. 3d). This was specific to hexadirectional coding (i.e., a 6-fold, grid-like modulation of the fMRI signal), specific to the entorhinal cortex where grid cells were previously detected[4], and specific to spatial information rather than being driven by the direction of saccadic eye movements[41–44]. Several recent findings suggest that socio-spatial navigation relies on neuronal substrates similar to those that process self-related spatial navigation. For instance, Stangl et al.[1] used intracranial recordings from the human MTL to reveal boundary-anchored representations during self-navigation as well as during the observation of others moving through space, highlighting a neural mechanism that signals an individual's vicinity to environmental borders. Further evidence comes from animal work with bats[2] and rodents[3] showing so-called "social" place cells in the hippocampus that were specifically tuned to the location of others in space. In contrast, separate subpopulations of CA1 pyramidal cells exclusively coded for self-location or showed a conjoint firing pattern for both self- and other-related location information (but see ref. [46] who did not find evidence for "social" coding). Besides hippocampal place cells, grid cell activity (or fMRI-based grid-like coding) is considered central to spatial[4,5,7–10] and mental navigation[13,14] but was so far not discussed in the context of socio-spatial navigation. Our findings suggest that grid-like codes in the human entorhinal cortex are indeed involved in such a process, allowing us to learn about the spatial routes that others take, and so potentially contributing to our ability to maneuver through crowded and dynamically changing environments as we encounter them in everyday situations.

On the broader scale of cortical structures, flexible self-navigation through the physical environment has been associated with an ensemble of brain regions that includes MTL structures, posterior-medial, parietal, striatal and prefrontal areas[19–23]. Our results are in line with this notion as we found that observing a demonstrator moving through space was associated with increased activation in the hippocampus and adjacent MTL, the caudate nucleus, as well as the ventromedial prefrontal and posterior cingulate cortices (contrast observation > navigation, Fig. 2a). The hippocampus is regarded as crucial for spatial navigation[21,22,47–49], and is assumed to work in parallel with the dorsal striatum (including the caudate nucleus) which was associated with response-learning and goal-directed navigation[19,34,35], as well as route following[33]. Especially the latter aspect seems important in the context of the present task since participants were asked to observe, memorize, and to re-trace the paths of a virtual demonstrator akin to following spatial routes. The ventromedial prefrontal and posterior cingulate cortices are often linked to successful memory retrieval[50–52] rather than encoding[53–55]. However, encoding and retrieval processes may act in parallel[56,57], especially for tasks that require the association of novel information with previously encountered material[58]. In the present modified navigation task, participants were required to encode a series of three path segments that constituted an

entire path trajectory. While observing the demonstrator moving from one path segment to the next, participants probably retrieved previous path segments as well. Furthermore, the ventromedial prefrontal and posterior cingulate cortices have also been linked to social cognition[59–65], suggesting that they likely supported memory encoding of the different paths in the present socio-spatial setting. Observation additionally triggered activation changes in the posterior superior temporal sulcus and fusiform gyrus, which have both been shown to process biological motion[36,37]. Thus, we revealed a profile of brain regions specifically concerned with tracking others' complex and goal-directed movements through space.

Evidence on how such macro-scale activation profiles might interact with the putative "spatial map" supported by entorhinal grid codes is sparse. The entorhinal cortex is located at the interface of the hippocampal-neocortical information processing system[66,67]. Its medial entorhinal division, which houses grid cells[4], receives anatomical projections from the hippocampus[67], the parahippocampal (or post-rhinal cortex in rodents[68]), retrosplenial and posterior parietal cortices[69], as well as from prefrontal areas[70,71]. In turn, output from the medial entorhinal cortex is routed back towards the hippocampus[72–74] and to a distributed set of cortical areas[66,75]. As such, the entorhinal cortex seems ideally positioned to integrate information and to communicate the current layout of the "spatial map" to the brain-wide navigation network[45]. In the present study, we attempted to characterize whether the activity of entorhinal grid-like codes was time-locked to co-activation of and to communication with regions involved in spatial navigation and visuospatial processing. Results indeed yielded increased co-activation of the striatum (caudate nucleus and putamen) when participants observed the demonstrator walking aligned with their individual entorhinal grid orientation and this effect was stronger the worse participants performed when subsequently re-tracing the paths (Fig. 4b, c). Going further, we next tested for functional interactions and found increased entorhinal connectivity with the caudate nucleus and hippocampus, the parahippocampal, posterior parietal and left lateral prefrontal cortex at larger grid magnitudes during observation, again negatively scaling with performance across participants (Fig. 4d). Chen et al.[76] recently investigated the relationship between grid-like codes in the entorhinal and ventromedial prefrontal cortex using human intracranial recordings during virtual, self-related spatial navigation. The authors could show that prefrontal theta oscillations exhibited a hexadirectional signal modulation similar to theta power in the entorhinal cortex and that both types of grid-like codes revealed a comparable grid orientation. These results resonate with our findings of increased co-activation and entorhinal-cortical connectivity as participants observed another individual walking aligned with their internal grid orientation. Given that human grid-like signals have been detected not only in the entorhinal cortex but also in other brain regions[76,77], it will be interesting to test how regions beyond the entorhinal cortex support the tracking of others. One could envision a mechanism of time-locked network dynamics that are triggered by the activity of grid-like codes in the entorhinal cortex, potentially coordinating information transfer about the current "socio-spatial map" across the brain, informing the observer about how to navigate space either by recalling their own movements, or by learning through observation, as shown here. We would like to highlight that this interpretation is speculative and in need of follow-up verification, and that we encourage others to provide more insights into the causal role of entorhinal grid-like signals within the broader network of brain regions representing (socio-)spatial information.

Our analyses revealed several surprising findings as well. For instance, we found that entorhinal grid-like codes, as well as co-activation and entorhinal-cortical connectivity that were timed to the activity of entorhinal grid-like codes, were associated with performance in the modified navigation task. In other words, participants

with higher grid magnitudes during observation performed worse when re-tracing the demonstrator's paths (Fig. 3f). This fits with a recent finding by Nau et al.[78] who reported stronger fMRI-based directional coding in the medial temporal lobe (including the posterior-medial entorhinal cortex) in participants who displayed low memory performance in a virtual spatial navigation task (i.e., measured by a larger drop-error when trying to place objects at their correct locations). We speculate that accurate performance might go hand-in-hand with neuronally efficient processing[79], which would be indexed by lower entorhinal grid magnitudes and a reduced need to employ additional co-activation of and connectivity with other brain regions. To provide an example, memory training was shown to decrease brain activity during memory processing while increasing performance[80]. Individual inability to accurately memorize and re-trace paths might thus be coupled to higher grid-like signals as there was increased need for support through the cognitive map provided by the entorhinal cortex. Notably, previous research reported mixed results, showing increased grid-like codes at better spatial memory performance[7–9] (i.e., a smaller drop-error when trying to place objects at their correct locations), or not showing any significant association[14,41]. Understanding the behavioral implications as well as the exact mechanistic contributions of grid-like codes for observation and navigation thus requires further scrutiny. Moreover, our grid-independent analyses showed that increased activation levels in the right posterior parietal cortex during observation were related to better performance when re-tracing the paths thereafter (Fig. 2d). The posterior parietal cortex was implicated in converging information about target location, movement, and the position of body parts to be able to plan or make movements towards a target[30], as well as in tracking spatial routes[32]. This activation cluster was only partly overlapping with grid-modulated voxels (Supplementary Fig. S10) but highlights a possible dissociation between overall activation levels and grid-like codes within an anatomical region, and how those signals support performance.

Furthermore, we found a strong effect of negative grid-like codes in V1 during observation (Fig. 3e, Supplementary Fig. S5). V1 activation levels were increased for observed directions that were misaligned with the individual V1 grid orientation (0 modulo 30°) compared to aligned directions (0 modulo 60°). Multiple explanations of how grid cell firing activity relates to macroscopic (fMRI-based) signals exist (for a discussion, see[6]). One possibility is that grid cells repeatedly fire during aligned movement and that this repetition would cause neural adaption (leading to a relative increase in the fMRI signal for misaligned directions). Signals might potentially be triggered by grid cells in the visual system (although it is currently unclear whether grid cells exist in V1) or by (entorhinal) mechanisms upstream. At present, we do not know which factors drive this effect and thus cannot provide a firm interpretation of why negative grid-like coding in V1 appears associated with tracking others navigating through space. We encourage future research to elucidate this finding, as well as the relationship between grid-like codes in the medial temporal lobe and visuals systems.

Contrary to previous work[7–14,81], we did not find significantly increased entorhinal grid-like codes during self-navigation (Fig. 3g; see Supplementary Information for numerically increased, as well as significantly decreased grid-like codes in the right and left entorhinal cortex, respectively, and see Supplementary Fig. S3 for significant grid-like coding in the anterior thalamus). The spatial stability of entorhinal grid-like codes was significantly decreased during navigation compared to observation periods (Fig. 3h), indicating that grid orientations in the entorhinal cortex were less clustered towards a specific direction. Grid orientations hence displayed larger variability across voxels, which might have resulted in an overall decrease in grid magnitude values during navigation. The finding might also be explained by the specific setup of our modified navigation task, which differed from

previous setups in several points. Participants were passively moved (after indicating and confirming their intended walking direction with a button press) and were required to re-trace path trajectories rather than freely navigating towards object locations in virtual reality space. Also, participant's viewpoints were directly placed behind the demonstrator's starting positions (but note that these randomly varied across trials). This design aspect could have enforced egocentric (striatal) rather than allocentric (hippocampal) processing[19], and could explain the striatal co-activation and connectivity time-locked to observation-related entorhinal grid-like codes. We speculate that decoupling the participant's viewpoint from the starting position might have rendered the task more hippocampal dependent[47]. In contrast to this argument, however, it appears that participants formed an allocentric representation since we detected entorhinal grid-like codes during observation, and since results showed generally increased activation levels in the hippocampus and entorhinal cortex during navigation (Fig. 2c). Future research might resolve this issue by expanding our current task-setup and decoupling view- and starting points. Interestingly, we found a significant 4-fold symmetrical modulation of the entorhinal cortex signal during navigation, in line with previous work[11]. A significant 4-fold modulation could reflect increased activation when moving in the cardinal directions (north, south, east, west). It is currently unclear whether such modulation is driven by grid cells but we speculate that the cardinal directions might act as mental coordinate system, allowing us to compare other movement directions with these major axes. He and Brown[11] reported such 4-fold modulation when barriers compartmentalized the environment and disrupted grid-like coding. Thus, it is possible that the "borders" of our movement area (indicated on the sandy desert plane) within the larger environment disrupted grid-like codes, pushing towards a 4-fold modulation of the entorhinal cortex signal.

On a final note, we would like to discuss two potential limitations of the present study. First, the design of the modified navigation task did not allow us to disentangle the tracking of others through space from participants planning their own future paths. Prior work demonstrated that entorhinal grid-like codes could be detected as participants imagined movement through space while they remained stationary[13,14]. It is possible that participants were imaging their own movements while observing the demonstrator and that this mental navigation caused elevated grid-like codes during observation. However, we consider this an unlikely explanation of the results. Participants completed a post-MRI interview regarding their individual strategies that they had adopted during observation periods. From the 47 participants that were included in the main grid analysis, only one person reported to have imagined the demonstrator's perspective. We excluded this participant from the sample and repeated the main analysis which left the results unchanged (see Supplementary Information, also for a general description of the reported strategies). We acknowledge that this does not fully preclude that processes related to planning and mental navigation contributed to our findings. Future studies might adjust our task design to disentangle potential grid-like processes related to planning, mental navigation, and observation.

Second, while it is plausible that entorhinal grid-like codes support socio-spatial navigation, we would like to emphasize that we are unable to make claims about the social specificity of our results. Entorhinal grid-like codes during observation might also be triggered by non-social features that need to be tracked (such as moving cars when crossing the road) and might be related to feature relevance (keeping an eye on the moving car is important to cross the road safely). While such results would speak for a general role of entorhinal grid-like codes in tracking moving features[82], the debate on the social specificity of brain processes[83] is fueled by initial evidence for "social" and "non-social" place cells[2]. Omer and colleagues dissociated signals by either presenting another individual (a demonstrator bat) or an object (a football) moving through space while the observer bat remained stationary. A similar task design might be helpful to resolve this issue in follow-up work with humans. Hence, we cannot draw firm conclusions regarding the involvement of entorhinal grid-like coding specifically in socio-spatial navigation but consider our work an important first step in this direction.

To summarize, we found that grid-like codes in the human entorhinal cortex track other individuals that navigate through space. Grid signals during observation were tied to increases in co-activation and entorhinal-cortical connectivity with an ensemble of regions, including the striatum, hippocampus, parahippocampal and right posterior parietal cortices, altogether modulated by accuracy when subsequently re-tracing the paths. While we are currently unable to answer whether these results are specifically related to *social* processing, findings might indicate that grid-like codes could be involved in socio-spatial navigation, concerned with tracking others' complex and goal-directed movements through space. We suggest that grid-like codes and their associated network dynamics could serve to distribute information about the location of others throughout the brain, laying the foundation for an internal "compass" that enables us to maneuver through crowded and dynamically changing environments as we encounter them in everyday situations.

## Methods

### Participant sample
Sixty participants volunteered for this study (aged 18–29 years, mean = 22 years, 45 females, biological sex determined by self-report, 7 left-handed). All participants were healthy, had normal or corrected-to-normal vision, gave written informed consent prior to participation, and received monetary compensation. Two participants were excluded from data analyses (one participant due to an anatomical brain abnormality, and one participant due to low performance in the modified navigation task), which left 58 participants for all following analyses (aged 18-28 years, mean = 22 years, 44 females, 7 left-handed). The study was reviewed and ethically approved by the ethics committee of the University of Vienna (Vienna, Austria; reference number 00538).

### Study procedures
Each participant underwent a single MRI session, starting out with the acquisition of the structural brain images while completing the task familiarization period (participants trained the subsequent modified navigation task during one run, all completed the same trials). This was followed by four runs of the actual modified navigation task and the acquisition of functional brain images.

### Modified navigation task: Virtual reality environment
To investigate whether grid-like codes supported the tracking of others, we translated previous animal work[2] to the human context. Our task projected participants into a first-person perspective within a virtual reality (VR) environment and asked them to observe a demonstrator moving through a circular arena. The path had to be held in memory and had to be re-traced after a short delay (for an example video, see https://osf.io/mhtgp/).

The VR environment depicted a desert scene with landmarks (sandstone towers, Joshua trees, cacti and other desert plants) and objects (railway wagons, car wrecks, an old gas station, wooden sheds, barrels, water towers, and wind turbines) that were placed around a circular arena. The sandy ground was contrasted by a blue sky filled with clouds, while the sun was fixed at the circular arena's zenith. The circular arena consisted of a movement area inside which the avatar (i.e., the demonstrator) and the participants could walk (radius = 60 virtual meters, vm; marked in red), as well as an observation area surrounding it (radius = 90 vm, marked in white; Fig. 1a). All landmarks, objects, and avatars were retrieved from the Unity asset store (https://assetstore.unity.com) and Sketchfab (https://sketchfab.com).

Movement speed of both the demonstrator and participant was set to 15 vm/sec and it was not possible to make rotational and translational movements at the same time (it was only possible to walk straight lines but not curves). The rotation speed was set to 50 deg/s and the participant's camera height was fixed to 1.7 vm. The VR environment and task were developed using Unity (software version 2019.4.5f1, https://unity.com).

Each trial consisted of an observation and a navigation period (Fig. 1b). Starting out, a cue signaled participants to pay attention to the demonstrator's upcoming path (2 s, "Please pay attention to the person and copy the path afterwards!"). To be able to visually follow the demonstrator's path throughout the entire movement area, the observer (i.e., the participant) was projected onto the border of the surrounding observation area, directly placed behind the demonstrator's starting point. The observer remained stationary during the entire observation period (i.e., the observer could not move or rotate but was able to see the entire movement area) since our goal was to disentangle potential grid-like codes supporting the tracking of others from those associated with self-related spatial navigation. A path consisted of three path segments between successive locations, each segment with a random length between 60–120 vm. Thus, single segments could be walked within 4–8 s and an entire observation period could last between 18–30 s (consisting of three consecutive segments, the rotation periods, plus a 1.5 s duration before/after the path was started/concluded). This was followed by a jittered delay during which a fixation cross was presented on the computer screen (ranging between 2–7 s, mean = 5 s).

At the start of the navigation period, the participant (i.e., the former observer) was projected onto the demonstrator's starting point and was asked to re-trace the path. Using an MR-compatible button box, participants could adjust their orientation and, once confirmed through button press, automatically walked to the opposite border of the movement area (it was not possible to pause and re-adjust inside the movement area). Participants were instructed to follow the previously observed path as close as possible without spending too much time on orientation adjustments. After 30 s, a time warning appeared (a red frame was shown around the screen), signaling that participants needed to reach the path's end point within the next 10 s.

Once the border of the movement area was reached for the third time, performance was quantified as (Euclidian) cumulative distance error (vm) averaged across all three points that were visited along the circumference of the movement area (thus, the average difference between the correct and actually visited points). Participants then received feedback about their navigation performance (1 s; happy emoji, distance error ≤ 20 vm; sad emoji, distance error > 20 vm). The choice of a 20 vm feedback threshold was inspired by previous work from Stangl et al.[8] who investigated grid-like coding as participants navigated through a virtual reality environment to place objects at their correct location. We subsequently confirmed that this threshold was appropriate by performing a small behavioral pre-test (with $N = 4$ participants) which showed that participants' performance stabilized at a cumulative distance error of ~20 vm after one training run. The feedback period was followed by another jittered delay (fixation cross presented on computer screen, duration ranging between 2–7 s, mean = 5 s), and a new trial started.

Participants engaged in four task runs of 12 trials each (i.e., consisting of 12 observation and 12 navigation periods). One run had a maximum duration of approx. 17 min.

## Modified navigation task: Path randomization
Each of the demonstrator's paths was oriented along one of 36 directions that could be divided into 6 directional bins, spanning the 360°-space with 10° angular resolution (see also Supplementary Fig. S8). To estimate individual grid orientations in unbiased fashion[7], we maximized the distance walked (vm) across the different directional bins in two steps.

First, a complete path within one trial consisted of three consecutive path segments. To evenly sample directions from the different directional bins, each individual segment was drawn from a different directional bin (thus, a trial included three path segments sampled from three different directional bins). Across the 12 trials per run (3 path segments per trial), we sampled from each directional bin 6 times ($6 \times 6 = 36$ segments per run). Participants completed a total of four runs of the navigation task; each directional bin thus appeared 24 times throughout the experiment. Second, to maximize the distance walked per path segment, we divided the movement area into six equally-sized sectors. Consecutive segments were randomized so that transitions between directly neighboring sectors were prohibited, enforcing a minimum path segment length of 60 vm (the length of a given path segment could vary between 60–120 vm). Starting points of paths were randomly generated within a sector and each sector hosted a start position twice within the same run, whereby the endpoint of the first path segment represented the starting point of the next path segment (and so forth).

Based on these restrictions, we created two trial sequences that were counter-balanced across the full sample of 60 participants. Overall, there was no significant difference in the distance walked across the 6 directional bins [one-way ANOVA, $N = 48$ (each data point indicating the trial-wise distance walked in each of the directional bins and in each of the four task runs), no significant main effect of directional bin, $p_{two\text{-}tailed} = 0.896$, see Supplementary Fig. S8a). The specific order of trials was randomly shuffled for each participant.

## Modified navigation task: determining chancel level performance with permutation testing
To determine whether participants performed better than chance (i.e., whether they were able to re-trace the demonstrator's paths more accurately than a random agent), we performed permutation-based testing. We pooled all paths that were presented to participants during the observation period, randomly selected paths, and simulated the performance of a "random agent". That is, we randomly chose three endpoints (for each of the three path segments) out of all possible endpoints along the circumference of the movement area (spanning the circular border in 1°-degree steps). We then calculated the Euclidean distance between each randomized endpoint and the "correct" endpoint of a specific path segment that was selected from the pool of possible paths, averaged across the three error distances, and accumulated a permutation distribution by iterating over these steps 5000 times (Fig. 1d). We then determined the performance value (cumulative distance error, vm) at the 5th percentile (39.3 vm), representing the chance level of a random participant. Finally, we tested whether the observed group performance was below this permutation-based chance level using a one-sample $t$-test.

## Eye tracking acquisition and data processing
To account for the potential impact of eye movements on grid-like codes[41–44], and to validate that participants were attending to the demonstrator's paths, we recorded horizontal and vertical eye gaze, as well as pupil size of each participant's right eye using a video-based infrared eye tracker (EyeLink 1000 Plus, SR Research, Ontario, Canada). Prior to each recording, raw eye movement data was mapped onto screen coordinates by means of a calibration procedure. Participants sequentially fixated on nine fixation points on the screen, arranged in a 3 × 3 grid. This was followed by a validation procedure during which the nine fixation points were presented once more while the differences between the current and previously obtained gaze fixations (from the calibration period) were measured. If these differences were <1° of visual angle, the calibration settings were accepted and the eye tracker recording was started. During recording, the data was digitized at a sampling rate of 1000 Hz and a potential drift in eye movements was corrected for every four trials (i.e., approx. every

1.4 min). Due to technical problems eye tracking was only possible in a subsample of 47 participants.

Data was corrected for eye blinks by removing samples for which the pupil size deviated more than one standard deviation (s.d.) from the mean across the entire time series. To determine saccadic eye movements, vertical and horizontal eye gaze was transformed into velocities (implemented using Fieldtrip's "ft_detect_movement"; latest software version downloaded on 4 March 2021, updated daily; https://www.fieldtriptoolbox.org). In brief, velocities exceeding a threshold of 6 × the s.d. of the velocity distribution and exceeding a duration of 12 ms were defined as saccades[84]. Saccade onsets during individual path segments (while observing or navigating, excluding any standing or rotation periods) were extracted. To avoid potential artifacts from other eye movements, and since eye movements typically occur every 200-300 ms[85], only events free of saccades and blinks in a 200 ms-interval prior to saccade onset were included (in other words, only events with a pre-saccadic fixation period were considered). We detected a total of 8720 saccades across all participants, across the four task runs, and across both observation and navigation periods ($N = 47$, mean number of saccades ± s.d., overall: 181.7 ± 57.7; observation periods: 97.2 ± 36; navigation periods: 85.45 ± 27.9).

## Imaging parameters

All imaging data were collected at the Neuroimaging Center of the University of Vienna, using a 3T Skyra MR-scanner (Siemens, Erlangen, Germany) equipped with a 32-channel head coil. During each of the four task runs, we acquired on average 474 (±7, s.d.) T2*-weighted blood oxygenation level-dependent (BOLD) images, using a partial-volume echo-planar imaging (EPI) sequence with the following parameters: repetition time (TR) = 2.029 s, echo time (TE) = 30 ms, number of slices = 30 axial slices, slice order = interleaved acquisition, field of view (FoV) = 216 mm, flip angle = 90°, slice thickness = 3 mm, in-place resolution = 2 × 2 mm, using parallel imaging with a GRAPPA acceleration factor of 2. Slices were oriented parallel to the long axis of the hippocampus.

Since the entorhinal cortices are susceptible to image distortions due to their vicinity to air-filled cavities, we acquired 30 images for post-hoc artifact correction using the abovementioned functional sequence but reversing the phase-encoding direction (thereby stretching potential image distortions into the opposite direction). Additionally, to facilitate the co-registration of anatomical entorhinal cortex masks to the partial-volume EPI images, we acquired 10 whole-brain EPI images with the following parameters: repetition time (TR) = 2.832 s, echo time (TE) = 30 ms, number of slices = 42 axial slices, slice order = interleaved acquisition, field of view (FoV) = 216 mm, flip angle = 90°, slice thickness = 3 mm, in-place resolution = 2 × 2 mm, using parallel imaging with a GRAPPA acceleration factor of 2. This sequence was thus similar to the partial-volume EPIs (but had a longer TR to allow for whole-brain coverage). As above, slices were oriented parallel to the long axis of the hippocampus.

The T1-weighted structural image was acquired using a Magnetization-Prepared Rapid Gradient Echo (MPRAGE) sequence with the following parameters: TR = 2.3 s; TE = 2.43 ms; FoV = 240 mm, flip angle = 8°, voxel size = 0.8 mm isotropic. To delineate the entorhinal cortex, we acquired a T2-weighted structural image using a turbo-spin-echo (TSE) Sampling Perfection with Application optimized Contrasts using different flip angle Evolution (SPACE) sequence with the following parameters: TR = 3.2 s; TE = 564 ms; FoV = 256 mm, voxel size = 0.8 mm isotropic. These slices were oriented perpendicular to the long axis of the hippocampus.

## MRI data preprocessing

The MRI data were processed using SPM (software version 12, http://www.fil.ion.ucl.ac.uk/spm/) in combination with Matlab (software version R2019b, The Mathworks, Natick, MA, USA) and the Functional Magnetic Resonance Imaging of the Brain (FMRIB) Software Library

(FSL, software version 5.0.1; https://fsl.fmrib.ox.ac.uk/fsl/fslwiki/)[86]. The first six volumes were excluded to allow for T1-equilibration. The remaining volumes were slice-time-corrected to the middle slice and realigned to the mean image calculated across the four task runs. Potential image distortions were corrected by applying FSL's "topup" command: we calculated the mean image based on the additional volumes acquired (phase-encoding direction reversed). Together with the original fMRI data, this image was then used to estimate and correct susceptibility-induced distortions. Since grid-like codes were analyzed in the participant-specific image space, we refrained from normalizing the data but applied a 3D Gaussian smoothing kernel (5 mm full-width at half maximum, FWHM).

For the whole-brain group analyses, the distortion-corrected data was additionally normalized into standard space. The structural scan was co-registered to the mean functional image and segmented into gray matter, white matter, and cerebrospinal fluid using the "New Segmentation" algorithm. All images (functional and structural) were spatially normalized to the Montreal Neurological Institute (MNI) EPI template (MNI-152) using Diffeomorphic Anatomical Registration Through Exponentiated Lie Algebra (DARTEL)[87], and functional images were smoothed with a 3D Gaussian kernel (5 mm FWHM).

## Whole-brain univariate fMRI analysis

We first set out to test whole-brain activation changes during observation and navigation, independent of grid-like codes. Using a Generalized Linear Modeling (GLM) approach, the BOLD response during the modified navigation task was modeled using separate task regressors, time-locked to the onsets of the respective events (cues, observation periods, navigation periods, feedback). All events were estimated as boxcar functions of specific durations and were convolved with the SPM default canonical hemodynamic response function (HRF). Cue and Feedback periods were modeled with a duration of 2 and 1 s, respectively. The duration of observation and navigation periods varied depending on the path length and the participant's behavior, and was defined through the on- and offsets of the VR environment on the computer screen (ranging between 18–40 s; see above). This included events such as orientation adjustments (rotations), walked path segments (translation periods), and time periods during which no movement occurred (short standing periods in-between). Thus, the implicit baseline consisted of the fixation cross. To account for noise due to head movement, we included the six realignment parameters, their first derivatives, and the squared first derivatives into the design matrix. A high-pass filter with a cutoff at 128 s was applied. The four runs of the modified navigation task were combined into one first-level model and contrasts were created ([observation ∩ navigation] > implicit baseline, observation/navigation > implicit baseline, observation > navigation and vice versa), collapsing across the different runs. To test for group effects, these contrast images were submitted to one-sample $t$-tests.

Additionally, we were interested in whether activation changes during observation and/or navigation scaled with individual performance. We thus ran two linear regression analyses (contrasts observation/navigation > implicit baseline) and added the cumulative distance error (vm, obtained during navigation periods and averaged across all three points of a path trajectory) as a covariate of interest.

## Definition of regions-of-interest (ROIs): entorhinal cortex ROIs and participant exclusions

We used the T2-weighted structural scans to anatomically delineate individual entorhinal cortices. First, ROIs were automatically generated using Automated Segmentation of the Hippocampal Subfields (ASHS, software version 1.0.0, https://sites.google.com/site/hipposubfields/)[88]. Second, to verify the ASHS-based segmentation, we also performed manual delineation of the entorhinal cortex by tracing its anatomical borders on the structural image. This was done using ITK-SNAP

(software version 3.6.; www.itksnap.org)[89], following the segmentation protocol provided by Berron et al.[90].

As we initially did not have a specific hypothesis regarding the laterality of brain effects, we collapsed the left and right masks into a bilateral entorhinal cortex image (for both the ASHS- and the ITK-SNAP-based delineations; but see Supplementary Information for separate analyses of ROIs in the left and right hemispheres). These masks were then binarized and transformed into the participant-specific space of the functional images. Since the functional images were only partial-volume slabs, co-registration was aided by an additional intermediate step that involved the mean whole-brain functional image[8]. First, each participant's T2-weighted structural image (together with the individual entorhinal mask) was co-registered to match the orientation of the mean whole-brain functional image. Second, the mean whole-brain functional image (together with the co-registered individual entorhinal mask) was co-registered to match the orientation of the mean partial-volume functional image. The quality of co-registration was confirmed through visual inspection of each mask's overlap with the individual (co-registered) structural and functional data (mean ± s.d.; ASHS, 56 ± 13 voxels; ITK-SNAP, 104 ± 23 voxels).

The entorhinal cortex lies in close proximity to the temporal horn of the lateral ventricle. Such tissue borders are often associated with lower signal-to-noise ratio, which is also what we experienced in a subsample of participants. To circumvent the issue of including noise, we only considered voxels that exceeded a signal-to-noise threshold of 0.8, leading to the fact that voxels along the anterior-medial entorhinal cortex border were partly dropped from the analyses (participants were excluded if there were less than 5 voxels left in the mask, and two participants were fully excluded from all grid code analyses involving the entorhinal cortex). We thus exclusively focused all analyses on the posterior-medial entorhinal cortex. After applying these restrictions, the final participant sample for which entorhinal cortex data was available comprised 49 (ASHS; 23 ± 13 voxels) or 51 (ITK-SNAP; 38 ± 25 voxels) participants.

## Control ROIs

To test whether grid-like codes were also detectable in other regions, we chose several control ROIs known to be involved in spatial navigation and visual processing but for which no significant grid-like coding was reported so far. These included the adjacent hippocampus, the parahippocampal cortex, the anterior thalamus, and the primary visual cortex (V1). Both the hippocampus and parahippocampal cortex masks were defined using the ASHS algorithm (the hippocampus was defined by merging the hippocampal subfields cornu ammonis (CA) 1–4 and the subiculum). To delineate the anterior thalamus, we used the stereotactic mean anatomical atlas provided by Krauth et al.[91] (© University of Zurich and ETH Zurich, Axel Krauth, Rémi Blanc, Alejandra Poveda, Daniel Jeanmonod, Anne Morel, Gábor Székely), which is based on histological, cytoarchitectural features defined ex vivo[92]. We specified the anterior thalamus by combining the anterior dorsal, -medial, and -ventral nucleus masks. The V1 mask was created using the Automatic Anatomical Labeling (AAL) atlas[93].

As above, left and right masks were combined into bilateral volumes and were transformed into the participant-specific image space (hippocampus, 447 ± 56 voxels; parahippocampal cortex, 160 ± 26 voxels; anterior thalamus, 41 ± 6 voxels; V1, 1136 ± 297 voxels). The quality of co-registration was confirmed through visual inspection of each mask's overlap with the individual structural and functional data of each participant (final sample, $N = 58$ participants).

## Analysis of grid-like codes

We next asked whether grid-like codes supported the tracking of others. All analyses were based on the openly available source code of the Grid Code Analysis Toolbox (GridCAT, software version 1.0.4, https://www.nitrc.org/projects/gridcat)[94], which follows the procedures established by Doeller et al.[7].

## Estimating grid orientations (GLM1)

Data was modeled identical to above with the exception that each path segment was included as a separate event and was modulated by its direction. In brief, the BOLD response during the modified navigation task was modeled using separate task regressors, time-locked to the onsets of the respective events (cues, path segments observed during observation periods, path segments walked during navigation periods, feedback). Translational events (i.e., individual path segments observed by the participant/walked by the demonstrator, and path segments walked by the participant) were modeled from the start of the movement until the next point was reached (thus, the duration was depended on the path segment length). To obtain the direction of each segment (i.e., translational event $t$), we calculated the translation angle ($\alpha_t$) based on the path segment coordinates within the movement area and referenced them to an arbitrary zero coordinate on the horizontal VR plane. The translational direction was then modeled using two parametric modulators, defined as $\sin(\alpha_t*6)$ and $\cos(\alpha_t*6)$. Orientation adjustments (rotations) and time periods during which no movement occurred (standing periods) were not explicitly modeled as these durations were typically very short (<2 s). The implicit baseline thus consisted of fixation periods as well as short rotation/standing events. Grid code analysis was performed in the native-space of each participant.

We then estimated individual grid orientations by partitioning each fMRI task run into estimation and test sets. To allow for stable estimations, we maximized the data available for grid orientation estimation by leveraging the inherent trial-design of each run. Specifically, we used a 12-fold cross-validation (CV) regime during which grid orientations were estimated on the path segments of 11 trials (consisting of 11 observation and 11 navigation periods) and tested on the path segments of the remaining trial (consisting of 1 observation and 1 navigation period). This was iterated until every trial was tested once.

During each CV-fold, voxel-wise grid orientations for observation/navigation conditions were estimated by fitting the fMRI data using GLM1. We then calculated the mean grid orientation ($\phi$) of each participant based on the average beta estimates ($\beta_1$ and $\beta_2$) of the two parametric modulators, extracted from all voxels within the respective ROI and calculated using $\arctan[\text{mean}(\beta_1)/\text{mean}(\beta_2)]/6$.

We specifically opted for a cross-validation regime within-runs in order to attenuate potential confounds often associated with the analysis of fMRI-based grid-like codes. First, we aimed to minimize noise due to signal drifts, as well as noise due to potential imprecisions stemming from the spatial co-registration across task runs. Second, our goal was to avoid changes in grid orientation, or "remapping", which might occur under specific circumstances. Although rodent work suggests that grid cell remapping is triggered by changes in the environment[15], the precise conditions that trigger grid cell remapping in humans are so far unknown. One possibility is that memory and spatial representations remap based on the temporal order of events (for example, due to so-called "event-boundaries", such as the beginning and the end of a scanning session or a task run), presumably to facilitate a cognitive separation of different experiences even in the same environment[95,96]. We did not set out to test any hypotheses regarding the temporal stability of such signals and thus addressed our questions by cross-validating within runs, reasoning that this approach would provide us with the most reliable measure of grid-like activation.

## Testing grid orientations (GLM2)

This model was identical to GLM1 but instead modeled each translational event modulated by the difference between the event's translational direction ($\alpha_t$) and the participant's mean grid orientation ($\phi$) using $\cos[6*(\alpha_t - \phi)]$. This analysis yielded a voxel-wise grid magnitude

value for each path segment (i.e., the strength of grid-like signal per path segment that was observed or walked).

Finally, results were averaged across CV-folds and across voxels within the respective ROI, and data was analyzed using a set of one-sample *t*-tests. We hypothesized that significant grid-like coding should be associated with an *increase* in the entorhinal cortex signal. This is based on the assumption that participants cross more grid cell firing fields as they, for instance, walk aligned with the underlying grid axes. The choice of statistical test thus reflected an a priori expectation, which is why we adopted an α-level of 0.05 (one-tailed). Additionally, we applied Bonferroni-correction to account for multiple comparisons (2 entorhinal cortex ROIs and 4 control ROIs), using a threshold of $\alpha_{Bonferroni} = 0.05/6$ ROIs = 0.008. Grid magnitude values exceeding the median value ± 3 × the median absolute deviation were excluded from the analyses[97]. Results remained stable when basing calculations on the full data set (i.e., not excluding any outliers, see Supplementary Tables S2 and S3) and the raw data is publicly provided (see below). To establish the reliability of results obtained from the 6-fold symmetrical model, we repeated the analysis with different control symmetries that were not expected to yield significantly increased grid magnitudes (see also ref. [8]).

### Representational stability of grid-like codes
Grid-like codes might be affected by variations in spatial and temporal signal stability[8]. In the case of high spatial instability, the estimated grid orientations might differ across the voxels within a ROI, resulting in random mean grid orientations and decreased grid magnitudes of grid-like codes. In the presence of low temporal stability, decreased grid magnitudes could be caused by varying grid orientations over time (for example, different grid orientations present in the estimation vs. test set).

To estimate the extent to which such stability aspects affected our results, we repeated the abovementioned analysis (GLM1, which served to quantify voxel-wise grid orientations) but partitioned each run into data halves (this way, we were able to estimate temporal stability in equally-sized data parts which would not have been possible with the CV-regime). To obtain a metric of spatial stability, we then calculated the coherence of estimated voxel-wise grid orientations between all voxels within the ROI using Rayleigh's test for non-uniformity of circular data. Higher *z*-values were associated with higher spatial stability (thus, significantly clustered grid orientations within the ROI). To obtain a metric of temporal stability, we compared the voxel-wise grid orientation between the first and second data half. A voxel was classified as "stable" if the differences in grid orientations was within ± 15° and temporal stability was quantified by the proportion (%) of "stable" voxels within the ROI.

### Brain activation time-locked to grid-like codes
Next, we asked whether the activity of entorhinal grid-like codes was time-locked to voxel-wise changes in cortical activation and entorhinal-cortical connectivity, and whether such dynamics were modulated by individual performance. To obtain a value of grid-like codes for each path segment observed or walked, we leveraged GLM2 (which was estimated in each of the 12 CV-folds) and extracted the grid magnitudes from the parametric modulation regressor for each path segment (i.e., relying on the difference between each event's translational direction and the mean grid orientation of the participant, whereby a smaller difference should be associated with a stronger grid-like signal within the entorhinal cortex ROI that was automatically segmented with ASHS). Data was then modeled similar to above (GLM1, GLM2) but translational events were modulated by the path segment-specific grid magnitudes. To perform group-level analyses, the GLM3 was estimated based on fMRI data in MNI standard-space.

We contrasted the parametric modulation regressors that the captured path segment-wise fluctuations in entorhinal grid magnitude against baseline (entorhinal grid magnitude during observation/navigation > implicit baseline) and tested for group effects by submitting the individual contrast images to separate one-sample *t*-tests. Additionally, to test whether activation changes time-locked to fluctuations in entorhinal grid magnitude scaled with performance, we ran linear regression analyses and added the cumulative distance error (vm, obtained during navigation periods and averaged across all three points of a path) as a covariate of interest.

### Connectivity time-locked to grid-like codes
Moreover, our goal was to probe whether fluctuations in entorhinal grid magnitude (i.e., the strength of the grid-like signal during each path segment) would be associated with entorhinal-cortical functional connectivity changes. To achieve this, we used the above-mentioned model (GLM3) and performed generalized psychophysiological interaction analysis (gPPI)[98]. We took the anatomical boundaries of the bilateral posterior-medial entorhinal cortex as a seed[99], extracted the first eigenvariate of its functional timecourse and adjusted for average activation levels using an *F*-contrast. The timecourse was then deconvolved to estimate the putative neural activity of the seed region (i.e., the physiological factor) and was multiplied with boxcar functions that defined the specific task events (i.e., the psychological factor). The resulting vectors were convolved with the canonical HRF, yielding one psychophysiological interaction regressor per condition-of-interest (i.e., for parametric modulation regressors that captured fluctuations in entorhinal grid magnitude during path segments of observation/navigation periods), and were contrasted against the implicit baseline.

Group-level connectivity analyses were performed using a set of one-sample *t*-tests. Again, linear regression analysis was used to test for connectivity changes time-locked to fluctuations in entorhinal grid magnitude that potentially scaled with performance by adding the cumulative distance error as a covariate of interest.

### Statistical analysis of behavioral data and ROI-based fMRI results
Analyses of navigation performance (distance error, vm) and ROI-based fMRI results (magnitude of grid-like codes, directional bins and representational stability) were carried out using R (software version 4.0.5; https://www.r-project.org), using a set of *t*-tests and ANOVA models. Significant interaction effects were followed-up with pair-wise comparisons using the R-package *emmeans* (software version 1.5.5.; https://cran.r-project.org/web/packages/emmeans/index.html) and were corrected for multiple comparisons (Tukey's HSD). Effect sizes were calculated as Cohen's *d* or eta squared ($\eta^2$) for *t*-test and ANOVA models, respectively[100]. As mentioned above, we hypothesized that significant grid-like coding should be associated with an *increase* in the entorhinal cortex signal. This is based on the assumption that participants cross more grid cell firing fields as they, for instance, walk aligned with the underlying grid axes. The choice of statistical test thus reflected an a priori expectation, which is why we adapted an α-level of 0.05 (one-tailed). Additionally, we applied Bonferroni-correction to account for multiple comparisons, using a threshold of $\alpha_{Bonferroni} = 0.05/6$ ROIs = 0.008. For all other cases, the α-level was set to 0.05 (two-tailed). Any exploratory analyses are explicitly described as such. Grid magnitude values exceeding the median value ± 3 × the median absolute deviation (MAD) were excluded from the analyses. We chose this method because the mean and standard deviation are particularly sensitive to outliers whereas the median is not[97]. Results regarding grid-like codes remained stable when basing calculations on the full data set (i.e., not excluding any outliers) and are reported in Supplementary Tables S2 and S3.

**Statistical thresholding of whole-brain fMRI results and anatomical labeling**

Unless stated otherwise, significance for all whole-brain fMRI analyses was assessed using cluster-inference with a cluster-defining threshold of $p < 0.001$ and a cluster-probability of $p < 0.05$ family-wise error (FWE) corrected for multiple comparisons. We only deviate from this a priori defined threshold to show exploratory results that might be of interest to the field, which is still in its early stages and where we thus think that a balance between exploratory and confirmatory findings is necessary. The corrected cluster size (i.e., the spatial extent of a cluster that is required in order to be labeled as significant) was calculated using the SPM extension "CorrClusTh.m" and the Newton–Raphson search method (script provided by Thomas Nichols, University of Warwick, United Kingdom, and Marko Wilke, University of Tübingen, Germany; http://www2.warwick.ac.uk/fac/sci/statistics/staff/academic-research/nichols/scripts/spm/). Anatomical nomenclature for all tables was obtained from the Laboratory for Neuro Imaging (LONI) Brain Atlas (LBPA40, http://www.loni.usc.edu/atlases/).

**Reporting summary**

Further information on research design is available in the Nature Portfolio Reporting Summary linked to this article.

## Data availability

Raw, anonymized fMRI data are available upon request to the corresponding author (isabella.wagner@univie.ac.at). At present, participant informed consent does not allow for depositing the full data set. All analyses are based on openly available software (see section "Code availability" below). Source data to reproduce all graphs (behavioral performance and ROI-based results of all grid analyses), as well as (un-) thresholded statistical whole-brain fMRI maps and their corresponding results tables are openly available at the Open Science Framework (https://osf.io/mhtgp/). Source data are provided with this paper.

## Code availability

All analysis is based on openly available software or custom code which is provided on the Open Science Framework (https://osf.io/mhtgp/). Software codes for the modified navigation task are available upon request to the corresponding author (isabella.wagner@univie.ac.at).

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

## Acknowledgements

The authors would like to thank Paul Anderson for excellent discussions, Ronald Sladky for support with the MRI sequences, and Magdalena Boch for advice on the eye-tracker setup. I.C.W. is supported by the Austrian Science Fund (FWF, P 34775). M.S. is supported by the National Institute of Neurological Disorders and Stroke (NINDS) of the National Institutes of Health (NIH, K99NS126715).

## Author contributions

Conceptualization: I.C.W. and D.B.O.; Methodology: I.C.W. and M.S.; Software: I.C.W., M.S., and A.L.; Validation: I.C.W. and L.P.G.; Formal analysis: I.C.W.; Investigation: I.C.W., B.T., and L.P.G.; Resources: I.C.W. and C.L.; Data curation: I.C.W.; Writing – Original draft: I.C.W.; Writing – reviewing & editing: all authors; Visualization: I.C.W.; Supervision: I.C.W.; Project administration: I.C.W.; Funding acquisition: I.C.W.

## Competing interests

The authors declare no competing interests.
