## [Peer Review File · Nature Communications]

Entorhinal grid-like codes and time-locked network dynamics
track others navigating through spaceREVIEWER COMMENTS

Reviewer #1 (Remarks to the Author):

Wagner et al. describe results from an fMRI study investigating grid-like BOLD signals while navigating and while observing others as they navigate through space. Participants viewed an avatar (demonstrator) as it navigated to three waypoints along the perimeter of a circular space. Following this observation period, participants were tasked with retracing the path of the demonstrator. The authors report evidence of grid-like signals in the entorhinal cortex that track others as they navigate. Contrary to prior reports, grid-like signals were not evident during self-navigation as participants attempted to re-trace the demonstrator's path. The magnitude of grid-like signals during observation were positively correlated with distance errors during the route-retracing task, that is, stronger grid-like signals were associated with worse behavioral performance. Moreover, grid-like signals in the entorhinal cortex during observation covaried with activity in the striatum, hippocampus, parahippocampal gyrus, and posterior parietal cortex (among other cortical regions). Functional coupling between entorhinal grid-like signals and BOLD activity in these regions was also positively associated with distance errors during route retracing (i.e., stronger functional connectivity associated with worse performance).

This is a very interesting and timely paper, and the hypotheses are well stated and justified. I do have a few questions and concerns about some of the statistical choices and interpretations, which I describe below.

1. In the opening paragraph (lines 44-46), the authors state that "...it has been shown that the neural firing signature of grid cell populations can be measured non-invasively using fMRI, in the form of so-called 'grid-like codes'". I realize this is a common interpretation of six-fold modulation of the fMRI BOLD signal, but without any direct evidence linking these signals, it strikes me as a leap to assume that modulation of voxel-wise BOLD activity necessarily reflects coordinated grid cell activity. It might be worth softening this language a bit, and exploring alternative possibilities in the discussion, or at least acknowledging the current gap in these two levels of analysis.

2. The navigation epoch always occurred shortly after the observation period. Could the lack of grid-like signals during active navigation simply reflect a neural adaptation effect? This might also explain my mean univariate BOLD activity in the hippocampus, striatum, and other regions such as the posterior parietal cortex was greater during observation relative to navigation.

3. What is the rationale of performing one-tailed t-tests when examining the magnitude of grid-like codes, especially when (as the authors acknowledge in the supplementary materials), prior studies (Horner et al., 2016; Nau et al., 2018) have reported evidence for negative grid-like signals in the entorhinal cortex? Moreover, if I understand correctly, there was evidence for negative grid-like modulation in the left entorhinal cortex during navigation. I think a more thorough justification of these statistical choices is necessary.

4. I realize this is common practice with the 6-fold modulation analyses, but what is the rationale for choosing 5- and 7-fold symmetry for control analyses? Especially given that there is at least some prior evidence of 4-fold symmetry during navigation (He and Brown, 2019, Current Biology).

5. The magnitude of 6-fold BOLD modulation was significantly different from zero, whereas 5- and 7-fold modulation was not. However, visual inspection of Fig 3B suggests that these magnitudes do not significantly differ from one another. I'm not entirely sure how to interpret this, can the authors elaborate?

6. I thought the inclusion of eye tracking was a great addition to the paper, but I have a few questions about the approach and interpretation of the results:

A) Why not simply treat saccades as a covariate in the principal analyses, similar to motion artifacts?

B) It looks as though there is at least some evidence of saccade-related grid-like modulation during navigation trials when considering the full sample ($p = .027$, line 156 of supplemental materials), but the effect is described as not significant.

C) The authors conclude that these control analyses confirm grid-like modulation of the BOLD signal during observation was driven by spatial rather than visual information. Is it really possible to dissociate spatial from visual information with this type of paradigm, which is limited to visual input (i.e., restricted self-motion feedback as is necessitated by the MRI environment)?

7. How did the authors choose ≤ 20 vm as criterion threshold for distance error?

8. In the discussion (lines 447-451), the authors note that “The ventromedial prefrontal and posterior cingulate cortices were previously linked to memory...likely supporting memory encoding of the different paths in this socio-spatial setting.” Several of the studies cited by the authors refer specifically to memory retrieval, not encoding. Moreover, subsequent memory effects in these regions are often negative, that is, a relative decrease in MPFC/PCC activity is predictive of successful encoding.

9. The authors note (line 481) that results from the whole brain and gPPI analyses “...suggests a mechanism of time-locked network dynamics that are triggered by the activity of grid-like codes in the entorhinal cortex, potentially coordinating information transfer about the current socio-spatial map across the brain...” It might be worth adding a sentence or two qualifying these results as speculative and in need of follow-up verification, especially since none of the analyses provide any insights into the causal role of entorhinal grid-like signals.

10. Of 60 participants, only 7 were right-handed, is this correct? Is there any reason to think that this may have led to the general left-lateralized grid-like effects during observation, compared to prior reports that grid-like signals are stronger in the right hemisphere? On a related note, was the grid-signal by hemisphere interaction significant? If not then the authors seem well justified in collapsing across hemispheres.

Reviewer #2 (Remarks to the Author):

In their article, “Entorhinal grid-like codes and time-locked network dynamics track others navigation through space,” Wagner et al. use fMRI ($n = 60$ participants) to show that grid-like codes in the entorhinal cortex track another individual who navigates through a virtual spatial environment. The task contained two main periods: observation (in which the subject observed the navigation path of a demonstrator) and retracing (in which the subject aimed at navigating the same path as the demonstrator). The study is well motivated and the manuscript is written clearly. The sample size seems appropriate, the authors use previously established techniques for analyzing the fMRI data, and the statistics appear sound. In comparison to previous studies on grid-like codes, there are several surprising results in this study including the finding of no grid-like codes during self-navigation, the finding of significant grid-like codes during a condition in which the environment is experienced in a distorted way (as the subject sees the demonstrator from a first-person perspective and not from a third-person overhead perspective during observation), and a negative correlation between the subjects’ performance and grid-like codes during observation (as well as other surprising relationships between fMRI activity and behavioral performance). My specific comments are as follows.

Lines 107-108: Why is 20 vm the appropriate threshold to which the subjects’ performance should be compared? Currently it seems like an arbitrary choice and the significantly better performance of the subjects thus appears meaningless.

Line 135: The choice of the contrast observation versus navigation is not optimal, as it does not allow to understand to what extent the result is actually driven by the contrast 0 versus navigation. The

comparison against a control condition with the same visual input as the observation period but without the need to pay attention to the subjects' navigation path would have been advantageous. The same comment applies to the contrast navigation versus observation. I would thus suggest showing the contrasts observation versus baseline and navigation versus baseline in Fig. 2 (also with regard to my next comment).

Fig. 2: It is unclear why the authors first show the results from the contrasts observation versus navigation and navigation versus observation, but then switch to the contrast observation versus baseline when aiming at demonstrating a correlation between fMRI activity and behavioral performance. The choice of this procedure is unclear.

Line 175: The choice of references to previous fMRI studies demonstrating grid-like codes is non-exhaustive and it is unclear why only these four studies are cited and not also the other relevant studies (also elsewhere such as in line 238 and line 273).

Line 198: Were grid-like codes stronger in the posteromedial part of the entorhinal cortex than in the anterolateral part (see, for example, Bierbrauer et al., *Science Advances*, 2020)? The same comment applies to lines 241-242.

Fig. 3A: Overlaying the hexadirectional pattern on the firing pattern of one grid cell is misleading regarding the basis of the hexadirectional fMRI signal (as it incorrectly suggests that the hexadirectional modulation emerges due to more firing fields being traversed when moving along the grid axes). I would thus suggest removing this panel from this figure (the basis of the hexadirectional fMRI signal has been discussed, for example, in Kunz et al., *Trends in Cognitive Sciences*, 2019).

Lines 279-288: The authors describe reduced spatial stability of grid orientations during the navigation condition. Hence, are there significant grid-like codes during navigation when the grid analysis is performed separately for each voxel (using different grid orientations for different voxels) and finally averaging across grid magnitudes from different voxels?

Sections after line 307 and after line 345: The analyses and results are interesting and novel, but it is surprising and counterintuitive that the activation and connectivity changes are negatively associated with performance (and in contrast to previous results including Doeller et al., *Nature*, 2010; Kunz et al., *Science*, 2015; Stangl et al., *Current Biology*, 2018; Maidenbaum et al., *PNAS*, 2018; Bierbrauer et al., *Science Advances*, 2020; etc.). This makes me wonder whether bad-performance trials are actually associated with some behavioral pattern of the demonstrators that lead to stronger grid-like codes? For example, do paths associated with bad performance more often cross the center of the environment? Chance performance for re-tracing a path differs between different parts of the environment (for a similar issue see Miller et al., *Nature Communications*, 2018), which should be considered and accounted for before computing relationships between neural activity and behavioral performance.

Lines 394-395: Why are different cluster-defining thresholds used ($p < 0.001$ and $p < 0.005$)?

Line 549: The authors may acknowledge that a recent studies with bats did not find evidence for social place cells (Dotson et al., *Science*, 2021).

Reviewer #3 (Remarks to the Author):

Wagner et al. studied grid-like codes in the human entorhinal cortex during a spatial route retracing task. Participants completed a virtual reality navigation task while in the fMRI scanner. On each trial they first observed a human-like avatar traverse the virtual space and after a short delay were asked to retrace the same path from a first-person perspective. The authors found increased entorhinal grid codes during observation of the avatar, but not while participants navigated through the space themselves. The increase in entorhinal grid-codes was associated to increased activity and

connectivity to the striatum, hippocampus, parahippocampal cortex and right posterior parietal cortex. This relationship was stronger for participants with lower accuracy on the task.

The paper investigates an interesting and very timely topic, seeking to extend findings from animal research to humans. Nevertheless, there are some major issues concerning the interpretability of the findings.

MAJOR Points

I have concerns regarding the grid code analyses, which have yielded several unusual results.

- First, no grid codes could be observed during navigation. I believe this makes the existence of grid codes during observational period difficult to interpret. The dissociation between observation and navigation indicates that something present only during the observational period was driving this signal. One possible explanation could be that the analyses pick up mostly on visual signals. Viewed from participants' fixed location during the observational period, the motion of the demonstrator was reflected in visual angles. This would also explain why there are quite strong effects in V1. Indeed, I am wondering whether the negative relationship between grid codes and performance reflects that during navigation participants have to translate the observed angles into navigational angles given the viewpoint differences (simply put: observing the demonstrator for instance walk to the right might imply I have to walk left during navigation, if the demonstrator was standing opposite of me). Another possibility is that it is related to the rather passive nature of the navigational task, given participants could not control the movement through the arena. Have the authors looked at whether grid-codes appear during rotational movements in the navigation phase? There are many other possible speculations, e.g. that the planning and memorizing of the movements happened during observation, or that there is some sort of interference between observation and navigation phase, i.e. effects would be visible when considering only the trials with longer intervals between observation and navigation. It is difficult to know. Given all this, the interpretation of the main finding of entorhinal grid codes during observation is difficult in my view. We don't know the reason for this difference between observation and navigation, and the strong negative signals in V1 and the negative relations to behavior hint at possible confounds.

- A second issue related to entorhinal grid codes concerns the statistical reliability of the findings. The cross-validation was done within runs, not between. I think the authors should repeat the analysis in a between run manner. The authors exclude different numbers of outliers for different analyses (I could not find a clear definition what constituted an outlier), and using a 1-tailed test found a p-value of .005, which surpassed chance baseline of .008. I think the authors should not exclude any outliers, and use a robust test framework instead if they are worried about violations of distributional assumptions. It would also be helpful to see a whole brain map of where the authors found putative grid effects.

- Lastly, I believe the paper would benefit if the authors would provide more insights into the V1 effect, and to be more upfront about as well. It would be interesting to understand the negative sign of the visual effects, and whether there could be any possible reason for a possible 30-degree rotated grid re-orientation that could have caused this effect (see my point above about translating between visual and navigational angles). In addition, the authors write "In summary, we found significantly increased grid-like signals in the entorhinal cortex, but not in any of the control regions, when participants were observing and putatively encoding the demonstrator's paths. ", but I believe is not true?

A separate major issue is that the social aspect of the navigation is given too much weight in my opinion. While the authors do mention in the discussion that the grid-codes might be tracking relevant non-spatial cues and that their design does not allow to dissociate these, this point should be made clearer in the abstract and introduction. The abstract currently says "Navigating through crowded, dynamically changing social environments requires the ability to keep track of other individuals", which I think needs to be toned down given the particulars of the experiment.

MINOR Points

I am not sure whether the cumulative distance error is appropriate for the present experiment, given

that participants were forced to walk in straight paths and the accuracy of the second and third segments strongly depended on the accuracy of the first (I assume segment endpoint is the starting point for the next segment, which explains why the 2nd and 3rd segment distance errors have almost 4 times as much cumulative distance). Another issue is that 20 cm is an arbitrary threshold, so the t-test against it is difficult to interpret. I don't doubt that participants perform above chance, but it would be useful to know what the true chance baseline of a random agent would be? In general, it would be nice to show behavior in main paper.

The finding of negative grid codes in left entorhinal cortex during navigation is intriguing, especially as during observation grid-codes are left lateralised. The authors expand a bit on the possible reasons for finding negative grid codes in V1 during the observation phase, but what are possible interpretations in the case of the entorhinal cortex?

The relationship between brain activation patterns and performance could be described more clearly on page 15, line 487/488: "[...] connectivity patterns, were positively, rather than negatively associated with performance (i.e., distance error [...])"

How was the (implicit) baseline determined for the fMRI analyses?

The concept of re-using the same neural structures for the (planning of) own movement and tracking that of others strongly reminds me of mirror neurons. Can authors comment on how these relate to each other?

When reporting that grid orientations during observation did not match those during navigation it would also be interesting to see the distribution of grid orientations across participants.

Reviewer #1

Wagner et al. describe results from an fMRI study investigating grid-like BOLD signals while navigating and while observing others as they navigate through space. Participants viewed an avatar (demonstrator) as it navigated to three waypoints along the perimeter of a circular space. Following this observation period, participants were tasked with retracing the path of the demonstrator. The authors report evidence of grid-like signals in the entorhinal cortex that track others as they navigate. Contrary to prior reports, grid-like signals were not evident during self-navigation as participants attempted to re-trace the demonstrator's path. The magnitude of grid-like signals during observation were positively correlated with distance errors during the route-retracing task, that is, stronger grid-like signals were associated with worse behavioral performance. Moreover, grid-like signals in the entorhinal cortex during observation covaried with activity in the striatum, hippocampus, parahippocampal gyrus, and posterior parietal cortex (among other cortical regions). Functional coupling between entorhinal grid-like signals and BOLD activity in these regions was also positively associated with distance errors during route retracing (i.e., stronger functional connectivity associated with worse performance).

This is a very interesting and timely paper, and the hypotheses are well stated and justified. I do have a few questions and concerns about some of the statistical choices and interpretations, which I describe below.

Author reply:

We thank the reviewer for the overall positive evaluation of our work and the constructive comments to improve it.

Reviewer 1, comment 1

In the opening paragraph (lines 44-46), the authors state that "...it has been shown that the neural firing signature of grid cell populations can be measured non-invasively using fMRI, in the form of so-called 'grid-like codes'". I realize this is a common interpretation of six-fold modulation of the fMRI BOLD signal, but without any direct evidence linking these signals, it strikes me as a leap to assume that modulation of voxel-wise BOLD activity necessarily reflects coordinated grid cell activity. It might be worth softening this language a bit, and exploring alternative possibilities in the discussion, or at least acknowledging the current gap in these two levels of analysis.

We fully agree with the reviewer that it remains an open and most important question how fMRI-based measurements precisely translate to coordinated grid cell activity. We softened our language in the introduction and discussion accordingly. Please see below for the relevant text passages.

Introduction (page 3, line 43): *"In humans, it has been discussed that the neural firing signature of grid cell populations might relate to measures obtained non-invasively using fMRI (for a recent discussion, see Kunz et al., 2019), in the form of so-called "grid-like codes" (Doeller et al., 2010)."*

Supplementary Materials (page 3, line 94): *"Multiple explanations of how grid cell firing activity relates to macroscopic (fMRI-based) signals exist (for a review, see Kunz et al., 2019). For example, increased grid-like coding might be related to the crossing of multiple firing fields (leading to an elevated fMRI signal). In contrast to this, different results can be obtained if repetition suppression mechanisms are at play (for a review see Barron et al., 2016). Grid cells would repeatedly fire during aligned movement and this repetition would cause a suppression of neuronal firing (leading to a decreased fMRI signal), given that inhibitory mechanisms appear vital for grid-like coding (Covey et al., 2013). (...)"*

Supplementary Materials (page 5, line 196): *"Additionally, there is an ongoing debate in the field on how grid cell firing relates to macroscopic (fMRI-based) signals (Kunz et al., 2019). (...) One possibility could be that increased grid-like codes during observation are suppressed during subsequent navigation, as could be explained by a neural adaption*

mechanism. (...) We encourage future studies to scrutinize the finding of negative grid-like codes and to determine how fMRI-based grid-like codes relate to underlying grid cell firing."

Discussion (page 18, line 587): *"Multiple explanations of how grid cell firing activity relates to macroscopic (fMRI-based) signals exist (for a discussion, see Kunz et al., 2019). For example, one possibility is that grid cells repeatedly fire during aligned movement and that this repetition would cause neural adaption (leading to a relative increase in the fMRI signal for misaligned directions). Signals might potentially be triggered by grid cells in the visual system (although it is currently unclear whether grid cells exist in V1) or by (entorhinal) mechanisms upstream. (...)"*

Reviewer 1, comment 2

The navigation epoch always occurred shortly after the observation period. Could the lack of grid-like signals during active navigation simply reflect a neural adaptation effect? This might also explain my mean univariate BOLD activity in the hippocampus, striatum, and other regions such as the posterior parietal cortex was greater during observation relative to navigation.

The reviewer raises an excellent point. To briefly summarize, neural adaption (or "repetition suppression") describes the phenomenon that the neural signal is reduced upon repeated presentation of the same stimulus or when associated stimuli are presented in succession (compared to the sequential presentation of two unrelated stimuli), potentially reflecting the presence of a "memory trace" (for a review, see Barron et al., 2016).

Rationale:

In the current task, neural adaption during navigation (which always followed observation) might indicate an association (or similarity) between the two periods. To be able to compare between navigation periods with high/low similarity to the previous observation period, we took into account individual performance: we expected that the observation-navigation similarity should be stronger for trials on which participants performed well (compared to trials during which participants performed less well; i.e., smaller vs. larger cumulative distance error, respectively). In other words, during high-performance trials, participants should have encoded the path trajectory well during observation, indicated by a stronger neural adaption during the subsequent navigation period.

We addressed this question with two new analyses. First, we performed a neural adaption analysis by comparing the entorhinal cortex signal between high- and low-performance navigation trials. Second, we drew upon our initial analysis of whole-brain univariate activation levels (independent of grid-like coding, see also **Fig. 2**) and assessed aspects of neural adaption as well as the relationship between entorhinal activation levels and grid-like codes.

Analysis approach #1, neural adaption:

We created a model that was similar to the initial whole-brain univariate fMRI analysis (independent of grid-like coding) but that incorporated a different regressor structure. To briefly explain, we used a Generalized Linear Model (GLM) to model the BOLD response during the modified navigation task with task regressors that were time-locked to the onsets of the respective events (cue + feedback periods, high-performance observation, low-performance observation, high-performance navigation, low-performance navigation). High- and low-performance trials were defined according to the feedback threshold employed in the modified navigation task (≤ 20 vm or > 20 vm cumulative distance error). Participants generally performed well [they showed significantly more high- (mean \pm standard error of the mean, s.e.m., 7.59 ± 0.29 trials per run) compared to low-performance trials (4.41 ± 0.29 trials per run); $N = 44$, paired-sample t -test, $t(43) = 5.46$, Cohen's $d = 0.82$, 95% confidence interval (CI) = [1.99, 4.34], $p_{\text{two-tailed}} < 0.0001$]. Note that this analysis did not include 14 participants who did not display any low-

performance trials (which would otherwise yield empty regressors in the GLM and render the whole idea of the analysis meaningless).

All task events were estimated as boxcar functions of specific durations and were convolved with the SPM default canonical hemodynamic response function (HRF). Cue and feedback periods were modelled with a duration of 2 and 1 seconds, respectively, and were collapsed within one regressor. The duration of observation and navigation periods varied depending on the path length and the participants' behavior and was defined through the on- and offsets of the VR environment on the computer screen (ranging between 18-40 seconds). This included events such as orientation adjustments (rotations), walked path segments (translation periods), and time periods during which no movement occurred (short standing periods in-between). To account for noise due to head movement, we included the six realignment parameters, their first derivatives, and the squared first derivatives into the design matrix. A high-pass filter with a cutoff at 128 s was applied. The four runs of the modified navigation task were combined into one first-level model and contrasts collapsing across the different task runs were created (high-/low-performance navigation > implicit baseline). Next, we took the anatomical boundaries of the bilateral posterior-medial entorhinal cortex (Maass et al., 2015), extracted the average activation levels during high- and low-performance navigation periods and compared them.

Results:

Results did not reveal a significant difference in entorhinal cortex activation between high- (0.41 ± 0.12 parameter estimates, arbitrary units, a.u.) and low-performance trials (0.33 ± 0.08) as participants re-traced the demonstrator's path trajectories (**Fig. R1a**; paired-samples *t*-test, $N = 44$ (full sample), $p_{\text{two-tailed}} = 0.456$; results did not change when excluding 2 outliers, $N = 42$, $p_{\text{two-tailed}} = 0.229$). Based on this analysis approach, we did not find strong evidence for neural adaption during navigation.

Analysis approach #2, entorhinal cortex activation:

In our initial manuscript, we had performed a whole-brain analysis to test for changes in brain activation during the modified navigation task (**Fig. 2**). We now leveraged this data to perform additional analyses focused on the entorhinal cortex.¹

First, we performed an analysis similar to the neural adaption analysis above. To avoid any artificial segregation into high- vs. low-performance trials, we extracted the average activation levels during navigation from the entorhinal cortex (same anatomical mask as above) and measured their cross-participant correlation with individual performance (vm, averaged across all three points of a path trajectory; similar to the whole-brain linear regression results presented in **Fig. 2d**). In the case of neural adaption, we would expect decreased activation at higher individual performance (in line with our finding of decreased grid-like coding at higher individual performance). Second, we determined whether activation levels during observation/navigation differed in the entorhinal cortex (generally lower activation during navigation could explain our null-finding regarding grid-like codes).

Results:

There was no significant correlation between navigation-related activation levels in the entorhinal cortex and individual performance ($N = 58$ (full sample), Pearson correlation, $p_{\text{two-tailed}} = 0.755$; **Fig. R1b**), confirming the results from the neural adaption analysis above. Next, we found that

¹ Note that we previously had applied whole-brain, cluster-based correction for multiple comparisons and thus might have missed any significant results pertaining to the relatively small entorhinal cortex region.

entorhinal activation levels were not significantly different between observation and navigation ($N = 58$ (full sample), paired-sample t -test, $p_{\text{two-tailed}} = 0.831$; **Fig. R1c**). Hence, our finding regarding no significant grid-like codes during navigation appears unrelated to general activation differences between the conditions.

Action taken:

We included these analyses in our updated manuscript (below we show the relevant text passages included in the updated results section). A full description of the analyses, as well as the figure, are included in our supplemental materials (not shown here since the text matches our reply above).

Results (page 12, line 338): “Another possible reason for the lack of significant entorhinal grid-like codes during navigation could be neural adaption (or “repetition suppression”) during navigation. Neural adaption describes the phenomenon that the neural signal is reduced upon repeated presentation of the same stimulus or when associated stimuli are presented in succession (compared to the sequential presentation of two unrelated stimuli), potentially reflecting the presence of a “memory trace” (for a review, see Barron et al., 2016). In the current task, observation periods were followed by navigation periods and neural adaption might thus indicate an association (or similarity) between the periods. To be able to compare between navigation periods with high/low similarity to the previous observation period, we took into account individual performance: we expected that the observation-navigation similarity should be stronger for trials during which participants performed well (compared to trials during which participants performed less well; i.e., smaller vs. larger cumulative distance error, respectively). In other words, during high-performance trials, participants should have encoded the path trajectory well during observation, indicated by a stronger neural adaption during the subsequent navigation period.

However, we did not find evidence for neural adaption (Supplementary Results S5, Supplementary Fig. S9), suggesting that this phenomenon did not impact our ability to detect grid-like codes during navigation periods. Additionally, it could be the case that overall entorhinal activation levels were stronger during observation compared to navigation periods, leading to significantly increased grid-like codes during observation (and no significant findings during navigation). Analysis showed that we could rule out this potential explanation as overall activation levels appeared comparable between the conditions (Supplementary Results S5, Supplementary Fig. S9).”

Fig. R1. Neural adaption analyses. (a) No significant difference in activation levels (quantified as parameter estimates, arbitrary units, a.u.) extracted from the entorhinal cortex during high- and low-performance navigation. **(b)** No significant association between the average activation levels in the entorhinal cortex during navigation and the cumulative distance error (vm) across participants. **(c)** No significant difference between activation levels in the entorhinal cortex during observation and navigation. Error bars reflect the standard error of the mean, s.e.m; n.s., not significant.

Reviewer 1, comment 3

What is the rationale of performing one-tailed t-tests when examining the magnitude of grid-like codes, especially when (as the authors acknowledge in the supplementary materials), prior studies (Horner et al., 2016; Nau et al., 2018) have reported evidence for negative grid-like signals in the entorhinal cortex? Moreover, if I understand correctly, there was evidence for negative grid-like modulation in the left entorhinal cortex during navigation. I think a more thorough justification of these statistical choices is necessary.

We hypothesized that significant grid-like coding should be associated with an *increase* in the entorhinal cortex signal. This is based on the assumption that participants cross more grid cell firing fields as they, for instance, walk aligned with the underlying grid axes. The choice of statistical test thus reflected an *a priori* expectation.

Results of negative grid-like coding in the visual or entorhinal cortices emerged as surprising side-findings in our work, and from what we understand, also had emerged unexpectedly in previous work (Horner et al., 2016). A hypothesis regarding negative grid-like coding would strike us as premature as the field currently is lacking evidence on how this could relate to underlying grid cell activity.

We clarified this important aspect in the manuscript.

Methods (page 27, line 962): *“Finally, results were averaged across CV-folds and across voxels within the respective ROI, and data was analyzed using a set of one-sample t-tests. We hypothesized that significant grid-like coding should be associated with an increase in the entorhinal cortex signal. This is based on the assumption that participants cross more grid cell firing fields as they, for instance, walk aligned with the underlying grid axes. The choice of statistical test thus reflected an a priori expectation, which is why we adopted an α -level of 0.05 (one-tailed).”*

Methods (page 29, line 1034): *“As mentioned above, we hypothesized that significant grid-like coding should be associated with an increase in the entorhinal cortex signal. This is based on the assumption that participants cross more grid cell firing fields as they, for instance, walk aligned with the underlying grid axes. The choice of statistical test thus reflected an a priori expectation, which is why we adapted an α -level of 0.05 (one-tailed).”*

Reviewer 1, comment 4

I realize this is common practice with the 6-fold modulation analyses, but what is the rationale for choosing 5- and 7-fold symmetry for control analyses? Especially given that there is at least some prior evidence of 4-fold symmetry during navigation (He and Brown, 2019, Current Biology).

As the reviewer states correctly, it is common practice to establish the reliability of entorhinal grid-like coding (i.e., 6-fold symmetry) by repeating the analysis with different control symmetries that are not expected to yield significantly increased grid magnitudes. We chose the 5- and 7-fold symmetries as control symmetries since we followed the approach by Stangl and colleagues (2018). We clarified this in the manuscript.

We thank the reviewer for pointing us towards the paper by He and Brown (2019). We performed the additional analysis and tested for a 4-fold symmetrical modulation of the fMRI signal in the entorhinal cortex during observation and navigation periods. All remaining parts of the analysis were identical to the original pipeline described in the manuscript.

To briefly explain, a significant 4-fold symmetrical modulation of the fMRI signal could reflect increased activation when moving in the cardinal directions (north, south, east, west), which might be unrelated to grid cells. We speculate that the cardinal directions might act as mental coordinate system, allowing us to compare other movement directions with these major axes.

Future studies should investigate the 4-fold modulation further and whether grid cells are driving this effect.

Analysis approach:

Identical to our main analysis, we split the fMRI data of each task run into independent sets and estimated/tested individual grid orientations using a 12-fold cross-validation regime (for a detailed description, please see **Methods**). We used 11 trials (each trial included one observation and navigation period) to estimate individual grid orientations based on the demonstrator's (or the participant's) movement trajectories through VR-space using a General Linear Model (GLM1; note that grid orientations were estimated separately for observation and navigation periods). The grid orientations' fit was then tested on the path segments of the remaining trial, which served to quantify the magnitude of grid-like codes in the left-out data set (GLM2). This procedure was repeated for each cross-validation fold and the resulting grid magnitudes were averaged across the different iterations. We performed the analyses for the entorhinal cortex region-of-interest (ROI) obtained through ASHS-based segmentation ($N = 49$), identical to our main analysis (**Methods**).

Results, observation:

As expected, the 4-fold symmetrical model did not yield significant results in the entorhinal cortex during observation [separate one-sample t -tests; $N = 47$ (2 outliers excluded), mean magnitude (arbitrary units, a.u.) \pm s.e.m., 0.07 ± 0.05 , $p_{\text{one-tailed}} = 0.082$, **Fig. R2**; $N = 49$ (full sample), 0.07 ± 0.06 , $p_{\text{one-tailed}} = 0.113$, not shown in figure].

Results, navigation:

During navigation, the 4-fold symmetrical model yielded significantly increased responses in the entorhinal cortex [$N = 43$ (6 outliers excluded), 0.1 ± 0.05 , $t(42) = 1.83$, Cohen's $d = 0.28$, 95% CI = $[-0.01, 0.21]$, $p_{\text{one-tailed}} = 0.037$, **Fig. R2**; $N = 49$ (full sample), 0.15 ± 0.13 , $p_{\text{one-tailed}} = 0.137$, not shown in figure]. Thus, results resonate with previous reports of 4-fold symmetrical signal modulation in the entorhinal cortex during navigation (He and Brown, 2019).

Action taken:

We clarified our choice of control symmetries. Further, we incorporated the novel analysis in the manuscript, along with a brief discussion, and now also cite the paper by He and Brown (2019).

Results (page 8, line 200): *"To establish the reliability of results obtained from the 6-fold symmetrical model, we repeated the analysis with different control symmetries that were not expected to yield significantly increased grid magnitudes. (...) Additionally, prior work reported 4-fold modulation of entorhinal cortex signal during self-related navigation (He and Brown, 2019). We thus tested whether such an effect was also present during observation but did not find evidence for a 4-fold signal periodicity in the entorhinal cortex [$N = 47$ (2 outliers excluded), 0.07 ± 0.05 , $p_{\text{one-tailed}} = 0.082$, Supplementary Fig. S4; $N = 49$ (full sample), 0.07 ± 0.06 , $p_{\text{one-tailed}} = 0.113$]."*

Results (page 10, line 281): *"As above, we also tested for a potential 4-fold signal periodicity since previous work demonstrated such an effect during navigation (He and Brown, 2019). Indeed, we found that the 4-fold symmetrical model yielded significantly increased navigation-related responses in the entorhinal cortex [$N = 43$ (6 outliers excluded), 0.1 ± 0.05 , $t(42) = 1.83$, Cohen's $d = 0.28$, 95% CI = $[-0.01, 0.21]$, $p_{\text{one-tailed}} = 0.037$, Supplementary Fig. S4; $N = 49$ (full sample), 0.15 ± 0.13 , $p_{\text{one-tailed}} = 0.137$, not shown]."*

Discussion (page 19, line 616): *"Interestingly, we found a significant 4-fold symmetrical modulation of the entorhinal cortex signal during navigation, in line with previous work (He and Brown, 2019). A significant 4-fold modulation could reflect increased activation when moving in the cardinal directions (north, south, east, west). It is currently unclear whether such modulation is driven by grid cells but we speculate that the cardinal directions might act as mental coordinate system, allowing us to compare other movement directions with these major axes. He and Brown (2019)*

reported such 4-fold modulation when barriers compartmentalized the environment and disrupted grid-like coding. Thus, it is possible that the “borders” of our movement area (indicated on the sandy desert plane) within the larger environment disrupted grid-like codes, pushing towards a 4-fold modulation of the entorhinal cortex signal.”

Methods (page 27, line 972): “To establish the reliability of results obtained from the 6-fold symmetrical model, we repeated the analysis with different control symmetries that are not expected to yield significantly increased grid magnitudes (see also Stangl et al., 2018).”

Fig. R2. Results of the 4-fold symmetrical model. Magnitude of symmetrical codes (4-fold) within the entorhinal cortex (arbitrary units, a.u.) during observation and navigation. Error bars reflect the standard error of the mean, s.e.m.; * $p < 0.05$.

Reviewer 1, comment 5

The magnitude of 6-fold BOLD modulation was significantly different from zero, whereas 5- and 7-fold modulation was not. However, visual inspection of Fig 3B suggests that these magnitudes do not significantly differ from one another. I’m not entirely sure how to interpret this, can the authors elaborate?

Given that grid cells per definition show an approximate angle of 60° between neighboring firing fields, we expected to see increased BOLD activation for the 6-fold model (which tests for increased activation depending on movement direction in steps of 60°), but not for any of the control models (4-, 5-, and 7-fold, where these control models assume activation peaks at directions that are different from 60°, for example, 90° for the 4-fold model).

In line with these theoretical assumptions, our results for the 6-fold modulation in the entorhinal cortex during observation were significantly increased whereas the results for all control analyses, including the 5-, 7-, and also the 4-fold modulation (see our reply to the comment directly above) were not. To our knowledge, all published studies so far showed that the 6-fold model can significantly explain variability in the BOLD signal (potentially indexing grid-cell-like representations), whereas the explanatory power of the control models (4-, 5-, and 7-fold) is not significantly different from zero (Bierbrauer et al., 2020; Doeller et al., 2010; He and Brown, 2019; Horner et al., 2016; Kunz et al., 2015; Stangl et al., 2018). We are not aware of any studies that have tested/shown that the 6-fold symmetry effect is significantly higher than the control models.

Although grid cells are supposed to show 60° angles between their firing fields on average, electrophysiology data of rats suggests that these firing fields are not expected to be perfectly

regular but instead often show variation and rather noisy firing patterns². This means that firing fields often do not show perfect 60° angles in practice (but instead often show some angle variability).

When testing for our main effect of grid-like codes, a 6-fold symmetry would represent a 60° angle between the putative firing fields of grid cells, while 4-, 5-, and 7-fold symmetry would represent 90°, 72°, and ~51° angles, respectively. Given some variability in these angles between the putative firing of grid cells, it is not surprising that also control models (e.g., a 7-fold model) might explain some of the variability in the BOLD signal as the model assumptions are relatively similar (e.g., 51° instead of 60° symmetry).

Conclusion and action taken:

For this reason, we think that it is appropriate to test the hypothesis whether any model can significantly fit the data (i.e., model fit significantly > 0) and we do not test the hypothesis whether a model fits the data significantly better than the control models. In our study, we found that only the 6-fold model significantly fits the data whereas the control models cannot. Importantly, these results are in line with results from previous studies in the field (Bierbrauer et al., 2020; Doeller et al., 2010; He and Brown, 2019; Horner et al., 2016; Kunz et al., 2015; Stangl et al., 2018), as in all of these studies the explanatory power of control models that tested for other periodicities was not significantly different from zero.

Reviewer 1, comment 6

I thought the inclusion of eye tracking was a great addition to the paper, but I have a few questions about the approach and interpretation of the results:

A) Why not simply treat saccades as a covariate in the principal analyses, similar to motion artifacts?

We opted for the analysis of saccade-based grid-like codes to directly test for a hexadirectional signal modulation during the modified navigation task, which would have indicated a potential contribution of visual grid cells in the entorhinal cortex. We would not have been able to answer this question otherwise and are generally unsure about the approach that the reviewer suggests.

To elaborate further, by including saccades as a covariate in the General Linear Model (GLM), similar to motion artifacts, all saccade-related signal would have been removed from the voxel-wise time course. Even though saccades might be related to visual grid-like codes in the entorhinal cortex and we wished to show that our results are independent of such a potential effect, saccadic eye movements serve to direct attention towards relevant features of the visual scene and thus appear crucial for visual information processing (Henderson, 2017; Pertzov et al., 2009; Renninger et al., 2007). Therefore, even if we did not find saccade-based grid-like codes in the entorhinal cortex during observation, regressing out any saccade-related signal might essentially translate into removing other, task-related signal from the data as our modified navigation paradigm relied on the visual domain.

B) It looks as though there is at least some evidence of saccade-related grid-like modulation during navigation trials when considering the full sample ($p = .027$, line 156 of supplemental materials), but the effect is described as not significant.

Indeed, analysis based on the entire participant sample showed significantly increased saccade-related grid-like coding in the entorhinal cortex during navigation [$N = 37$ (full sample), $p_{\text{one-tailed}} =$

² Personal communication of co-author Matthias Stangl with several rodent electrophysiologists, for example Kate Jeffery, as well as inspection of unpublished data from Kate Jeffery's lab.

0.027; **Supplementary Results S6**]. Further analyses, however, revealed that this result was driven by two extreme values, which could be classified as “outliers”. Outliers (for all analyses in the paper, including all grid analyses) were *a priori* defined as values exceeding a difference of +/- 3 absolute deviations from the median (Leys et al., 2013). After removing these outliers, the results were not significant [$N = 35$ (excluding two outliers), $p_{\text{one-tailed}} = 0.096$; **Supplementary Fig. S8c**].

We present both results to the reader as it is our aim to paint a clear and maximally transparent picture of the data.

C) The authors conclude that these control analyses confirm grid-like modulation of the BOLD signal during observation was driven by spatial rather than visual information. Is it really possible to dissociate spatial from visual information with this type of paradigm, which is limited to visual input (i.e., restricted self-motion feedback as is necessitated by the MRI environment)?

We recognize that this statement might be misleading and rephrased the text accordingly.

Results (page 13, line 370): “(...) altogether suggesting that our result of significant grid-like codes in the entorhinal cortex when observing the demonstrator’s paths was based on spatial information rather than the number or direction of saccadic eye movements.”

Supplementary Materials (page 8, line 304): “This suggests that our finding of significant grid-like codes in the entorhinal cortex when observing the demonstrator’s paths (Fig. 3d) was primarily based on spatial information rather than the direction of saccadic eye movements.”

Discussion (page 16, line 488): “(...) and specific to spatial information rather than being driven by the direction of saccadic eye movements (Julian et al., 2018; Killian et al., 2012; Nau et al., 2018; Staudigl et al., 2018).”

Reviewer 1, comment 7

How did the authors choose ≤ 20 vm as criterion threshold for distance error?

The choice of a 20 vm feedback threshold was inspired by previous work from Stangl and colleagues (2018) who investigated grid-like coding as participants navigated through a virtual reality environment to place objects at their correct location (Stangl et al., 2018). We subsequently confirmed that this threshold was appropriate by performing a small behavioral pre-test (with $N = 4$ participants) which showed that participants’ performance stabilized at a cumulative distance error of ~ 20 vm after one training run. We clarified this in the manuscript.

We would like to highlight though that this threshold did not affect the main analyses or the brain-behavior associations that we report. In other words, any analyses pertaining to an association between grid-like codes and behavioral performance are focused on individual differences, irrespective of how the feedback threshold was defined.

Methods (page 22, line 730): “The choice of a 20 vm feedback threshold was inspired by previous work from Stangl and colleagues (2018) who investigated grid-like coding as participants navigated through a virtual reality environment to place objects at their correct location. We subsequently confirmed that this threshold was appropriate by performing a small behavioral pre-test (with $N = 4$ participants) which showed that performance stabilized at a cumulative distance error of ~ 20 vm after one training run.”

Also, please note that we provide additional analysis regarding this point further down (reply to reviewer 3, comment 5). Specifically, we determined chance level based on navigation of a random participant. We accumulated a permutation distribution by simulating random performance 5000 times and determined chance level as the 5th percentile of this distribution (chance = ~ 36 vm cumulative distance error). We then compared the obtained performance of our participant group with random chance and could show that participants on average

performed significantly better. We added this analysis to the manuscript (text changes are not shown here).

Reviewer 1, comment 8

In the discussion (lines 447-451), the authors note that “The ventromedial prefrontal and posterior cingulate cortices were previously linked to memory...likely supporting memory encoding of the different paths in this socio-spatial setting.” Several of the studies cited by the authors refer specifically to memory retrieval, not encoding. Moreover, subsequent memory effects in these regions are often negative, that is, a relative decrease in MPFC/PCC activity is predictive of successful encoding.

The reviewer raises a valid point. We realize that our statement was unclear and rephrased it accordingly (please see below for changes in the text).

Indeed, previous studies reported relatively decreased activation in the medial prefrontal and posterior cingulate cortices (MPFC, PCC) predictive of successful memory encoding, and relatively increased activation of the same regions during successful memory retrieval (Daselaar et al., 2004, 2009; Huijbers et al., 2012). However, we previously showed that brain regions that are typically associated with encoding and retrieval work in parallel while participants form durable memories (Wagner et al., 2016). Encoding and retrieval processes might be intertwined and the validity of separating encoding/retrieval is currently being debated (Kragel and Voss, 2022). More specifically, parallel encoding/retrieval processes might benefit memory formation in certain tasks that, for example, require the association of novel information with previously encountered material or the embedding into prior knowledge structures (van Kesteren et al., 2012). Relatedly, previous work demonstrated relatively increased MPFC activation during memory encoding when participants studied information that was congruent with prior knowledge (van Kesteren et al., 2013).

In the present modified navigation task, participants were required to encode a series of three path segments that constituted an entire path trajectory. While observing the demonstrator moving from one path segment to the next, participants likely retrieved previous path segments as well. This might explain why we found increased activation in MPFC and PCC regions during observation. We clarified this in the text.

Discussion (page 17, line 514): *“The ventromedial prefrontal and posterior cingulate cortices are often linked to successful memory retrieval (King et al., 2015; Rugg and Vilberg, 2012; Watrous et al., 2013) rather than encoding (Daselaar et al., 2004, 2009; Huijbers et al., 2012). However, encoding and retrieval processes may act in parallel (Kragel and Voss, 2022; Wagner et al., 2016), especially for tasks that require the association of novel information with previously encountered material (van Kesteren et al., 2012). In the present modified navigation task, participants were required to encode a series of three path segments that constituted an entire path trajectory. While observing the demonstrator moving from one path segment to the next, participants probably retrieved previous path segments as well. Furthermore, the ventromedial prefrontal and posterior cingulate cortices have also been linked to social cognition (Amodio and Frith, 2006; Gaesser et al., 2019; Gallagher et al., 2000, 2002; Saxe and Powell, 2006; Spreng and Mar, 2012; Wagner et al., 2020), suggesting that they likely supported memory encoding of the different paths in the present socio-spatial setting.”*

Reviewer 1, comment 9

The authors note (line 481) that results from the whole brain and gPPI analyses "...suggests a mechanism of time-locked network dynamics that are triggered by the activity of grid-like codes in the entorhinal cortex, potentially coordinating information transfer about the current socio-spatial map across the brain..." It might be worth adding a sentence or two qualifying these results as speculative and in need of follow-up verification, especially since none of the analyses provide any insights into the causal role of entorhinal grid-like signals.

We now highlight that this interpretation is speculative and awaits verification.

Discussion (page 17, line 555): *"We would like to highlight that this interpretation is speculative and in need of follow-up verification, and that we encourage others to provide more insights into the causal role of entorhinal grid-like signals within the broader network of brain regions representing (socio-)spatial information."*

Reviewer 1, comment 10

Of 60 participants, only 7 were right-handed, is this correct? Is there any reason to think that this may have led to the general left-lateralized grid-like effects during observation, compared to prior reports that grid-like signals are stronger in the right hemisphere? On a related note, was the grid-signal by hemisphere interaction significant? If not then the authors seem well justified in collapsing across hemispheres.

We thank the reviewer for catching this. We made an error in the original manuscript and corrected it in the revised version. What we meant to state, was that 7/60 participants were *left-handed*.

Nevertheless, we performed the additional analysis to test whether participants' handedness affected the lateralization of grid-like coding in the entorhinal cortex.

Analysis approach:

Grid magnitude values were compared between right- and left-handed participants using a set of independent-samples *t*-tests. Analyses were based on the same participant samples as presented in our initial manuscript ($N = 37$; 5 left-handed participants; sample size after applying restrictions for voxel selection and outlier correction, using participants for whom both left and right lateralized data was available; **Supplementary Results S2 and S4**). We did not directly compare between hemispheres as potential ROI-differences might stem from unrelated factors such as a different number of voxels included in the ROIs or potential differences in signal-to-noise ratio (Poldrack, 2007).

Results

There were no significant group differences for any of the comparisons (all $p > 0.05$; summarized in **Table R1**), suggesting no effect of handedness on grid-like coding in the left/right entorhinal cortices. Due to the small sample size of left-handed participants ($N = 5$) we consider these results preliminary and advocate for interpreting them with caution.

*(please find **Table R1** on the next page)*

Table R1. Entorhinal grid magnitudes in right- and left-handed participants.

Statistical comparison	Handedness	Group size (N)	Grid magnitude, arbitrary units (mean \pm s.e.m.)	p -value
Observation, left hemisphere (N _{total} = 37)	Right	32	0.08 \pm 0.03	0.781
	Left	5	0.04 \pm 0.12	
Observation, right hemisphere (N _{total} = 37)	Right	33	-0.01 \pm 0.05	0.247
	Left	4	-0.12 \pm 0.05	
Navigation, left hemisphere (N _{total} = 37)	Right	32	-0.08 \pm 0.04	0.564
	Left	5	-0.02 \pm 0.08	
Navigation, right hemisphere (N _{total} = 37)	Right	33	0.06 \pm 0.04	0.323
	Left	4	0.31 \pm 0.2	

Reviewer #2:

In their article, “Entorhinal grid-like codes and time-locked network dynamics track others navigation through space,” Wagner et al. use fMRI (n = 60 participants) to show that grid-like codes in the entorhinal cortex track another individual who navigates through a virtual spatial environment. The task contained two main periods: observation (in which the subject observed the navigation path of a demonstrator) and retracing (in which the subject aimed at navigating the same path as the demonstrator). The study is well motivated and the manuscript is written clearly. The sample size seems appropriate, the authors use previously established techniques for analyzing the fMRI data, and the statistics appear sound. In comparison to previous studies on grid-like codes, there are several surprising results in this study including the finding of no grid-like codes during self-navigation, the finding of significant grid-like codes during a condition in which the environment is experienced in a distorted way (as the subject sees the demonstrator from a first-person perspective and not from a third-person overhead perspective during observation), and a negative correlation between the subjects’ performance and grid-like codes during observation (as well as other surprising relationships between fMRI activity and behavioral performance). My specific comments are as follows.

Author reply:

We thank the reviewer for the positive feedback and the constructive comments to improve it.

Reviewer 2, comment 1

Lines 107-108: Why is 20 vm the appropriate threshold to which the subjects’ performance should be compared? Currently it seems like an arbitrary choice and the significantly better performance of the subjects thus appears meaningless.

The choice of a 20 vm feedback threshold was initially inspired by previous work from Stangl and colleagues who investigated grid-like coding as participants navigated through a virtual reality environment to place objects at their correct location (Stangl et al., 2018). We subsequently confirmed that this threshold was appropriate by performing a small behavioral pre-test (with *N* = 4 participants) which showed that participants’ performance stabilized at a cumulative distance error of ~20 vm after one training run. We clarified this in the manuscript. Also, we removed the comparison of performance to the 20 vm threshold. Please see below for text changes.

We would like to highlight that this threshold did not affect the main analyses or the brain-behavior associations that we report. In other words, any analyses pertaining to an association between grid-like codes and behavioral performance are focused on individual differences, irrespective of how the feedback threshold was defined.

Methods (page 22, line 730): *“The choice of a 20 vm feedback threshold was inspired by previous work from Stangl and colleagues (2018) who investigated grid-like coding as participants navigated through a virtual reality environment to place objects at their correct location. We subsequently confirmed that this threshold was appropriate by performing a small behavioral pre-test (with N = 4 participants) which showed that participants’ performance stabilized at a cumulative distance error of ~20 vm after one training run.”*

Also, please note that we provide additional analysis regarding this point further down (reply to reviewer 3, comment 5). Specifically, we determined chance level based on navigation of a random participant. We accumulated a permutation distribution by simulating random performance 5000 times and determined chance level as the 5th percentile of this distribution (chance = ~36 vm cumulative distance error). We then compared the obtained performance of our participant group with random chance and could show that participants on average performed significantly better. We added this analysis to the manuscript (text changes are not shown here).

Reviewer 2, comment 2

Line 135: The choice of the contrast observation versus navigation is not optimal, as it does not allow to understand to what extent the result is actually driven by the contrast 0 versus navigation. The comparison against a control condition with the same visual input as the observation period but without the need to pay attention to the subjects’ navigation path would have been advantageous. The same comment applies to the contrast navigation versus observation. I would thus suggest showing the contrasts observation versus baseline and navigation versus baseline in figure 2 (also with regard to my next comment).

We thank the reviewer for this thoughtful comment.

We restructured **Fig. 2** and now include an overview of the contrasts observation > baseline and navigation > baseline (the full results are included in the supplementary material, please see the relevant text changes below). Despite the fact that visual input was not matched between observation/navigation conditions, we reasoned that the direct comparison would be interesting to the reader and we thus would like to retain these results in the figure.

We agree that the comparison against a control condition with the same visual input as during observation (but without participants’ requirement to pay attention to the demonstrator) would have been ideal. When designing the task, we opted against the inclusion of more conditions as our main goal was to maximize the time participants were observing/navigating while keeping the scanning duration at reasonable length. We briefly discuss this issue in our updated discussion.

Results (page 6, line 142): *“Since visual input was considerably different between observation and navigation conditions (stationary viewpoint vs. navigation), we also compared each of the conditions against the (fixation) baseline. Activation changes appeared largely similar for both conditions, including increased activation in visual, parietal and lateral prefrontal regions (Fig. 2c, Supplementary Fig. S2, Supplementary Table S1).”*

Reviewer 2, comment 3

Figure 2: It is unclear why the authors first show the results from the contrasts observation versus navigation and navigation versus observation, but then switch to the contrast observation versus baseline when aiming at demonstrating a correlation between fMRI activity and behavioral performance. The choice of this procedure is unclear.

We apologize for being unclear and adjusted this in the text.

We tested the association between whole-brain univariate activation levels and behavioral performance during observation and navigation periods using linear regression. To be able to clearly interpret the direction of potential effects, we performed this analysis in two steps, first using the contrast observation > baseline, then navigation > baseline. This way, we were able to conclude that higher activation levels during observation were associated with individual differences in behavioral performance whereas this was not the case during navigation.

Results (page 6, line 156): *“We performed this analysis in two steps to be able to clearly interpret the direction of potential effects, first focusing on observation and then on navigation periods (each contrasted against baseline).”*

Reviewer 2, comment 4

Line 175: The choice of references to previous fMRI studies demonstrating grid-like codes is non-exhaustive and it is unclear why only these four studies are cited and not also the other relevant studies (also elsewhere such as in line 238 and line 273).

We acknowledge that our references to prior work were not exhaustive and now added the following studies:

References regarding fMRI-based studies that demonstrate grid-like codes during navigation:

Bierbrauer, A., Kunz, L., Gomes, C.A., Luhmann, M., Deuker, L., Getzmann, S., Wascher, E., Gajewski, P.D., Hengstler, J.G., Fernandez-Alvarez, M., et al. (2020). Unmasking selective path integration deficits in Alzheimer’s disease risk carriers. *Sci. Adv.* 6, eaba1394.

Kunz, L., Schroder, T.N., Lee, H., Montag, C., Lachmann, B., Sariyska, R., Reuter, M., Stirnberg, R., Stocker, T., Messing-Floeter, P.C., et al. (2015). Reduced grid-cell-like representations in adults at genetic risk for Alzheimer’s disease. *Science* 350(6259), 430–433.

He, Q., Brown, T. I. (2019). Environmental Barriers Disrupt Grid-like Representations in Humans during Navigation. *Current Biology* 29, 2718-2722.e3.

Julian, J. B., Doeller, C. F. (2021). Remapping and realignment in the human hippocampal formation predict context-dependent spatial behavior. *Nature Neuroscience* 24, 863–872.

Introduction (page 3, line 45): *“Moreover, these grid-like codes were shown to support spatial (Bierbrauer et al., 2020; Doeller et al., 2010; He and Brown, 2019; Julian and Doeller, 2021; Kunz et al., 2015; Stangl et al., 2018) as well as mental self-related navigation (Bellmund et al., 2016; Horner et al., 2016), (...).”*

Results (page 8, line 179): *“(…) as well as by previous work probing entorhinal grid cell activity (or grid-like codes) during spatial (and mental) self-navigation in both animals (Hafting et al., 2005) and humans (Bellmund et al., 2016; Bierbrauer et al., 2020; Doeller et al., 2010; He and Brown, 2019; Horner et al., 2016; Julian and Doeller, 2021; Kunz et al., 2015; Stangl et al., 2018).”*

Results (page 10, line 268): *“In analogy to previous work that reported entorhinal grid cell activity, or grid-like codes, during spatial (and mental) self-navigation (Bellmund et al., 2016; Bierbrauer et al., 2020; Doeller et al., 2010; He and Brown, 2019; Horner et al., 2016; Julian and Doeller, 2021; Kunz et al., 2015; Stangl et al., 2018), (...).”*

Discussion (page 16, line 498): *“Besides hippocampal place cells, grid cell activity (or grid-like coding) is considered central to spatial (Bierbrauer et al., 2020; Doeller et al., 2010; Hafting et al., 2005; Jacobs et al., 2013; Kunz et al., 2015; Stangl et al., 2018) and mental navigation (Bellmund et al., 2016; Horner et al., 2016) but was so far not discussed in the context of socio-spatial navigation.”*

Discussion (page 18, line 596): *“Contrary to previous work (Bellmund et al., 2016; Bierbrauer et al., 2020; Doeller et al., 2010; He and Brown, 2019; Horner et al., 2016; Julian and Doeller, 2021; Kunz et al., 2015; Stangl et al., 2018), we did not find significantly increased entorhinal grid-like codes during self-navigation (...).”*

References regarding the association of grid-like codes and behavior:

Bierbrauer, A., Kunz, L., Gomes, C.A., Luhmann, M., Deuker, L., Getzmann, S., Wascher, E., Gajewski, P.D., Hengstler, J.G., Fernandez-Alvarez, M., et al. (2020). Unmasking selective path integration deficits in Alzheimer's disease risk carriers. *Sci. Adv.* 6, eaba1394.

Kunz, L., Schroder, T.N., Lee, H., Montag, C., Lachmann, B., Sariyska, R., Reuter, M., Stirnberg, R., Stocker, T., Messing-Floeter, P.C., et al. (2015). Reduced grid-cell-like representations in adults at genetic risk for Alzheimer's disease. *Science* 350(6259), 430–433.

Maidenbaum, S., Miller, J., Stein, J.M., and Jacobs, J. (2018). Grid-like hexadirectional modulation of human entorhinal theta oscillations. *Proc. Natl. Acad. Sci. U. S. A.* 115, 10798–10803.

Results (page 10, line 250): *“Previous studies highlighted a significant association between entorhinal grid-like codes during navigation and behavioral performance (Bierbrauer et al., 2020; Doeller et al., 2010; Kunz et al., 2015; Maidenbaum et al., 2018; Stangl et al., 2018).”*

References regarding the relationship between grid cells and fMRI-based grid-like codes:

Kunz, L., Maidenbaum, S., Chen, D., Wang, L., Jacobs, J., and Axmacher, N. (2019). Mesoscopic Neural Representations in Spatial Navigation. *Trends Cogn. Sci.* 23, 615–630.

Introduction (page 3, line 43): *“In humans, it has been discussed that the neural firing signature of grid cell populations might relate to measures obtained non-invasively using fMRI (for a recent discussion, see Kunz et al., 2019), in the form of so-called “grid-like codes” (Doeller et al., 2010).”*

Supplementary Materials (page 3, line 94): *“Multiple explanations of how grid cell firing activity relates to macroscopic (fMRI-based) signals exist (for a review, see Kunz et al., 2019). (...)”*

Supplementary Materials (page 5, line 196): *“Additionally, there is an ongoing debate in the field on how grid cell firing relates to macroscopic (fMRI-based) signals (Kunz et al., 2019). (...)”*

Discussion (page 18, line 597): *“Multiple explanations of how grid cell firing activity relates to macroscopic (fMRI-based) signals exist (for a discussion, see Kunz et al., 2019). For example, (...)”*

Reviewer 2, comment 5

Line 198: Were grid-like codes stronger in the posteromedial part of the entorhinal cortex than in the anterolateral part (see, for example, Bierbrauer et al., *Science Advances*, 2020)? The same comment applies to lines 241-242.

Unfortunately, we were unable to compare between grid-like codes in posterior-medial and anterior-lateral entorhinal regions as our data was affected by signal drop-outs in the anterior-lateral part. We had already described this in our initial version of the manuscript but made sure to clarify this further in the revised version.

Methods (page 26, line 890): *“(…) leading to the fact that voxels along the anterior-medial entorhinal cortex border were partly dropped from the analyses (participants were excluded if there were less than 5 voxels left in the mask, and two participants were fully excluded from all grid code analyses involving the entorhinal cortex). We thus exclusively focused all analyses on the posterior-medial entorhinal cortex.”*

Reviewer 2, comment 6

Figure 3A: Overlaying the hexadirectional pattern on the firing pattern of one grid cell is misleading regarding the basis of the hexadirectional fMRI signal (as it incorrectly suggests that the hexadirectional modulation emerges due to more firing fields being traversed when moving along the grid axes). I would thus suggest removing this panel from this figure (the basis of the hexadirectional fMRI signal has been discussed, for example, in Kunz et al., *Trends in Cognitive Sciences*, 2019).

We removed the panel from **Fig. 3** (not shown here) and added the citation of Kunz and colleagues (2019) to our manuscript.

Introduction (page 3, line 43): *“In humans, it has been discussed that the neural firing signature of grid cell populations might relate to measures obtained non-invasively using fMRI (for a recent discussion, see Kunz et al., 2019), in the form of so-called “grid-like codes” (Doeller et al., 2010).”*

Supplementary Materials (page 3, line 94): *“Multiple explanations of how grid cell firing activity relates to macroscopic (fMRI-based) signals exist (for a review, see Kunz et al., 2019). (...)”*

Supplementary Materials (page 5, line 196): *“Additionally, there is an ongoing debate in the field on how grid cell firing relates to macroscopic (fMRI-based) signals (Kunz et al., 2019). (...)”*

Discussion (page 18, line 597): *“Multiple explanations of how grid cell firing activity relates to macroscopic (fMRI-based) signals exist (for a discussion, see Kunz et al., 2019). For example, (...)”*

Reviewer 2, comment 7

Lines 279-288: The authors describe reduced spatial stability of grid orientations during the navigation condition. Hence, are there significant grid-like codes during navigation when the grid analysis is performed separately for each voxel (using different grid orientations for different voxels) and finally averaging across grid magnitudes from different voxels?

We thank the reviewer for this excellent suggestion and performed the analysis.

Analysis approach:

We leveraged the data from our initial analysis of representational stability for which we had partitioned each run into data halves. Voxel-wise grid orientations were estimated on the first data half of each run (GLM1). We then tested the fit of the voxel-wise grid orientations using the independent, second half of the data (GLM2). Concretely, we adopted this approach to quantify grid magnitudes during navigation on a voxel-by-voxel basis and finally averaged across these to obtain a summary score of grid-like codes within the bilateral entorhinal cortex (ASHS-based segmentation).

Results:

There were no significant grid-like codes in the entorhinal cortex during navigation when performing the grid analysis for each voxel separately (one-sample t -test, $N = 46$ (3 outliers excluded), -0.02 ± 0.03 , $p_{\text{one-tailed}} = 0.199$; $N = 49$ (full sample), 0.09 ± 0.08 , $p_{\text{one-tailed}} = 0.14$), altogether corroborating our initial results.

Action taken:

We added this to our manuscript:

Results (page 11, line 323): *“This led us to reason that it might still be possible to obtain significant grid-like coding during navigation when performing the analysis separately for each voxel (i.e., when estimating and testing grid orientations on a voxel-by-voxel basis rather than averaging across the entorhinal cortex ROI). We thus repeated the abovementioned analysis of representational stability for which we had partitioned each task run into data halves and adopted the approach to quantify grid magnitudes during navigation on a voxel-by-voxel basis (that is, we repeated GLM2 for each voxel within the bilateral entorhinal cortex ROI), and then averaged across grid magnitudes to obtain a summary score. However, in line with all previous analyses, results did not yield significant grid-like coding during navigation (one-sample t -test, $N = 46$ (3 outliers excluded), mean grid magnitude (arbitrary units, a.u.) \pm s.e.m., -0.02 ± 0.03 , $p_{\text{one-tailed}} = 0.199$; $N = 49$ (full sample), 0.09 ± 0.08 , $p_{\text{one-tailed}} = 0.14$).”*

Reviewer 2, comment 8

Sections after line 307 and after line 345: The analyses and results are interesting and novel, but it is surprising and counterintuitive that the activation and connectivity changes are negatively associated with performance (and in contrast to previous results including Doeller et al., Nature, 2010; Kunz et al., Science, 2015; Stangl et al., Current Biology, 2018; Maidenbaum et al., PNAS, 2018; Bierbrauer et al., Science

Advances, 2020; etc.). This makes me wonder whether bad-performance trials are actually associated with some behavioral pattern of the demonstrators that lead to stronger grid-like codes? For example, do paths associated with bad performance more often cross the center of the environment? Chance performance for re-tracing a path differs between different parts of the environment (for a similar issue see Miller et al., Nature Communications, 2018), which should be considered and accounted for before computing relationships between neural activity and behavioral performance.

We thank the reviewer for this insightful comment. We structured our reply in two parts.

Part #1, Paths of good vs. bad trials:

We performed additional analysis to clarify whether our finding of increased entorhinal grid-like codes during observation at lower performance might depend on path differences between high- vs. low-performance trials (i.e., trials with a smaller vs. larger cumulative distance error, respectively). The reviewer suggested that paths during low-performance trials might have crossed the center of the environment more often, leading to a longer average path length, and potentially contributing to increased grid-like coding.

We tested whether this was the case and calculated the average path length of high- vs. low-performance trials (defined as trials with ≤ 20 vm or > 20 vm cumulative distance error). There was no significant difference between the average path length of high- vs. low-performance trials [paired-sample *t*-test, $N = 59$ (full sample), $p_{\text{two-tailed}} = 0.095$; high-performance trials: 110.2 ± 0.03 vm, low-performance trials: 105.3 ± 0.38 vm]. Also, there was no significant association between the magnitude of entorhinal grid-like codes during observation and the average path length observed across participants [$N = 46$ (3 outliers excluded, same sample as in original analysis), $p_{\text{two-tailed}} = 0.071$].

We conclude that the brain-behavior association is unlikely to stem from differences in path length between high- and low-performance trials and added this analysis to the manuscript (text changes are shown at the end of this reply).

Part #2, Chance performance in different parts of the environment:

We thank the reviewer for pointing us towards the approach by Miller and colleagues (2018) and performed the additional analysis, taking into account that chance performance for re-tracing a given path correctly differs for different areas of the environment.

To recapitulate, Miller and colleagues employed an object-location paradigm and computed the distance between the location where the object was dropped and its actual location (response error in virtual meters, vm). However, the probability for locating an object correctly differed within different areas in the environment. Considering the environmental boundaries of their virtual reality space, the distribution of possible error distances was smaller for objects located in (or close to) the center of the environment compared to objects located close to the environmental borders. They accounted for this difference by normalizing all objects to the same response range, which yielded an accuracy score per participant response. We adopted their approach for our modified navigation task.

Analysis approach:

In our paradigm, participants were asked to re-trace the demonstrator's paths as accurately as possible, moving from a starting point towards specific endpoints along the circumference of the circular movement area (that is, participants were not able to stop and turn within the movement area). A single path consisted of three path segments.

For each path segment that a participant re-traced after having observed the demonstrator, we calculated the probability distribution of all possible Euclidean distance errors. In other words, we simulated all possible endpoints that a participant could have navigated towards, spanning the entire circumference of the movement arena in 1°-degree steps (**Fig. R3a**). We then calculated the Euclidean distance between the correct endpoint and all other possible points, including the endpoint that the participant actually navigated towards. Based on this probability distribution, we determined the percentile rank of the actual error distance of a given path segment (i.e., the endpoint that a given participant actually navigated towards) relative to all possible error distances and determined its accuracy score (1-percentile/100; **Fig. R3b**). Higher values thus indicated that participants re-traced the observed path segments more precisely. Altogether, this yielded accuracy scores that were normalized with respect to differences in the probability of obtaining a correct response at different areas of the circular environment.

Results:

The mean accuracy across participants was ~86%. Next, we tested whether this different method for assessing individual performance affected our main result regarding the brain-behavior relationship of increased grid-like magnitudes at lower individual performance across participants and repeated the correlation analysis. Results remained virtually unchanged, once again showing increased grid-like codes in participants with lower overall accuracy ($N = 47$, same sample as for main analysis: $r_{\text{Pearson}} = -0.43$, 95% CI = [-0.64, -0.16], $p_{\text{two-tailed}} = 0.003$; **Fig. R3c**).

Action taken:

We included the additional analyses in our manuscript (please see below). A full description of the approach, as well as the figure, are included in our supplemental materials (not shown here since the text matches our reply above).

Lastly, we would like to highlight a related finding by Nau and colleagues (2022) who reported stronger fMRI-based directional coding in the parahippocampal gyrus and medial temporal regions (including the posterior-medial entorhinal cortex) in participants who displayed low memory performance in a virtual navigation task where objects had to be placed at their correct location (Nau et al., 2020).

Results (page 10, line 257): *“We performed several control analyses to validate the brain-behavior relationship. First, we could show that the results did not stem from specific path patterns, such as potentially longer paths during low-compared to high-performance trials (the paths of low-performance trials could have crossed the center of the movement area more often, yielding a longer average path length and thus perhaps a stronger grid-like signal, but this was not the case; Supplementary Results S3). Second, we repeated the correlation analysis with a newly-defined performance measure, taking into account that chance performance for re-tracing a given path correctly differed for different endpoints along the circumference of the movement area. Using this normalized accuracy measure (Miller et al., 2018), results remained virtually identical (Supplementary Results S3, Supplementary Fig. S6). Altogether, these analyses corroborated our result of stronger grid-like codes the worse participants performed.”*

Discussion (page 18, line 562): *“This fits with a recent finding by Nau and colleagues (2022) who reported stronger fMRI-based directional coding in the medial temporal lobe (including the posterior-medial entorhinal cortex) in participants who displayed low memory performance in a virtual spatial navigation task (i.e., measured by a larger drop-error when trying to place objects at their correct locations).”*

*(please find **Fig. R3** on the next page)*

Fig. R3. Normalized accuracy measure and correlation with entorhinal grid-like codes during observation. (a) Color gradient indicates chance level at different points on the circumference of the movement area in virtual reality space (120 vm = maximum error distance). Black and red circles indicate exemplary endpoints of the demonstrator and participant, respectively, yielding an error distance of 46 vm for this specific path segment. (b) Probability distribution of all possible error distances. The normalized accuracy is calculated as the percentile rank of the actual path segment (i.e., endpoint that participant actually navigated towards) relative to all possible error distances (1-percentile/100). (c) Initial results remained virtually unchanged, once more showing increased grid-like codes in the entorhinal cortex during observation at lower overall accuracy.

Reviewer 2, comment 9

Lines 394-395: Why are different cluster-defining thresholds used ($p < 0.001$ and $p < 0.005$)?

Significance for all whole-brain fMRI analyses was assessed using cluster-inference with a cluster-defining threshold of $p < 0.001$ and a cluster-probability of $p < 0.05$ family-wise error (FWE) corrected for multiple comparisons.

In Fig. 4c we presented results regarding an association between voxel-wise activation changes and entorhinal grid-like codes during observation, and how this relationship depended on individual performance. These results were the only ones that did not survive our abovementioned significance threshold but instead survived a cluster-defining threshold of $p < 0.005$. We explicitly noted this already in our initial manuscript but made sure to highlight it more clearly (please see below). We would like to keep these results in the manuscript as we believe that they will be of interest to the field, which is still in its early stages and where we thus think that a good balance between exploratory findings (requiring more liberal controls so as to not be too tight on *type II*-errors) and confirmatory findings (requiring more strict controls so as to not be too liberal on *type I*-errors) is necessary.

Results (page 13, line 411): "(...) please note that we here deviated from our a priori cluster-defining threshold of $p < 0.001$, showing results that reached significance only at a more lenient statistical threshold, $p < 0.05$ FWE-corrected at cluster level using a cluster-defining threshold of $p < 0.005$, cluster size = 157 voxels; Fig. 4c)."

Methods (page 29, line 1047): "Unless stated otherwise, significance for all whole-brain fMRI analyses was assessed using cluster-inference with a cluster-defining threshold of $p < 0.001$ and a cluster-probability of $p < 0.05$ family-wise error (FWE) corrected for multiple comparisons. We only deviate from this a priori defined threshold to show exploratory results that might be of interest to the field, which is still in its early stages and where we thus think that a balance between exploratory and confirmatory findings is necessary."

Reviewer 2, comment 10

Line 549: The authors may acknowledge that a recent studies with bats did not find evidence for social place cells (Dotson et al., Science, 2021).

We thank the reviewer for pointing this out and we have included this study in our discussion.

Discussion (page 16, line 494): *“Further evidence comes from animal work with bats (Omer et al., 2018) and rodents (Danjo et al., 2018) showing so-called “social” place cells in the hippocampus that were specifically tuned to the location of others in space. In contrast, separate subpopulations of CA1 pyramidal cells exclusively coded for self-location or showed a conjoint firing pattern for both self- and other-related location information (but see Dotson and Yartsev, 2021 who did not find evidence for “social” coding).”*

Reviewer #3:

Wagner et al. studied grid-like codes in the human entorhinal cortex during a spatial route retracing task. Participants completed a virtual reality navigation task while in the fMRI scanner. On each trial they first observed a human-like avatar traverse the virtual space and after a short delay were asked to retrace the same path from a first-person perspective. The authors found increased entorhinal grid codes during observation of the avatar, but not while participants navigated through the space themselves. The increase in entorhinal grid-codes was associated to increased activity and connectivity to the striatum, hippocampus, parahippocampal cortex and right posterior parietal cortex. This relationship was stronger for participants with lower accuracy on the task.

The paper investigates an interesting and very timely topic, seeking to extend findings from animal research to humans. Nevertheless, there are some major issues concerning the interpretability of the findings.

Author reply:

We thank the reviewer for the overall positive evaluation of our work and the constructive comments to improve it.

MAJOR Points

I have concerns regarding the grid code analyses, which have yielded several unusual results.

Reviewer 3, comment 1

- First, no grid codes could be observed during navigation. I believe this makes the existence of grid codes during observational period difficult to interpret. The dissociation between observation and navigation indicates that something present only during the observational period was driving this signal. One possible explanation could be that the analyses pick up mostly on visual signals. Viewed from participants' fixed location during the observational period, the motion of the demonstrator was reflected in visual angles. This would also explain why there are quite strong effects in V1. Indeed, I am wondering whether the negative relationship between grid codes and performance reflects that during navigation participants have to translate the observed angles into navigational angles given the viewpoint differences (simply put: observing the de-monstrator for instance walk to the right might imply I have to walk left during navigation, if the demonstrator was standing opposite of me). Another possibility is that it is related to the rather passive nature of the navigational task, given participants could not control the movement through the arena. Have the authors looked at whether grid-codes appear during rotational movements in the navigation phase? There are many other possible speculations, e.g. that the planning and memorizing of the movements happened during observation, or that there is some sort of interference between observation and navigation phase, i.e. effects would be visible when considering only the trials

with longer intervals between observation and navigation. It is difficult to know. Given all this, the interpretation of the main finding of entorhinal grid codes during observation is difficult in my view. We don't know the reason for this difference between observation and navigation, and the strong negative signals in V1 and the negative relations to behavior hint at possible confounds.

We thank the reviewer for raising these important points. In the following, we highlight the individual reviewer comments once more together with our response to each of them:

Reviewer comment: *“First, no grid codes could be observed during navigation. I believe this makes the existence of grid codes during observational period difficult to interpret. The dissociation between observation and navigation indicates that something present only during the observational period was driving this signal. One possible explanation could be that the analyses pick up mostly on visual signals. Viewed from participants’ fixed location during the observational period, the motion of the demonstrator was reflected in visual angles. This would also explain why there are quite strong effects in V1.”*

We fully agree with the reviewer that the absence of grid-like signals in the entorhinal cortex during navigation is unexpected. One reason for this could be that our modified navigation task required participants to re-trace the demonstrator’s path trajectories rather than freely navigation towards object locations in virtual reality space, rendering it substantially different from previous work (Bellmund et al., 2016; Bierbrauer et al., 2020; Doeller et al., 2010; He and Brown, 2019; Horner et al., 2016; Julian and Doeller, 2021; Kunz et al., 2015).

Initial control analysis, no saccade-based grid-like codes:

We also agree with the reviewer that eye movements could be a potential factor driving effects during observation, as eye movements were shown to influence grid-like signals in the entorhinal cortex (Julian et al., 2018; Killian et al., 2012; Nau et al., 2018; Staudigl et al., 2018). We had addressed this analytically by re-analyzing grid-like signals during observation periods using saccadic eye movements. More specifically, we based our grid analysis on the angles between successive saccades that participants made when following the demonstrator’s path trajectories during observation (rather than using the actual path trajectories in space, see also **Methods**). This analysis did not show significant grid-like signals in the entorhinal cortex during observation [$N = 37$ (full sample, no outliers), $p_{\text{one-tailed}} = 0.35$; see also **Supplementary Results S6** and **Supplementary Fig. S8**].

Additional analysis, no association between grid-like signals and eye movements:

Prompted by the reviewer’s comment, we performed additional analysis and tested whether the magnitude of grid-like codes in the entorhinal cortex during observation scaled with the average number of saccades that participants performed. Correlation analysis did not show a significant association between the magnitude of grid-like response during observation and the average number of saccades across participants [$N = 37$ (reduced sample since eye-tracking data was missing for some participants; $p_{\text{two-tailed}} = 0.427$].

Additional analysis, 4-fold modulation of the entorhinal cortex signal during navigation:

Irrespective of visual signals but tapping into the reviewer’s comment regarding the lack of significantly increased grid-like representations during navigation, we also performed analysis regarding a potential 4-fold modulation of the fMRI signal in the entorhinal cortex during navigation (also inspired by reviewer 1, comment 4). To briefly explain, such modulation could index increased activation when moving in the cardinal directions (north, south, east, west). We speculate that the cardinal directions might act as mental coordinate system, allowing us to

compare other movement directions with these major axes. Previous work (He and Brown, 2019) showed such a 4-fold modulation during navigation when environmental barriers disrupted grid-like coding in the entorhinal cortex.

In line with their findings, the 4-fold symmetrical model yielded significantly increased responses in the entorhinal cortex during navigation [$N = 43$ (6 outliers excluded), 0.1 ± 0.05 , $t(42) = 1.83$, Cohen's $d = 0.28$, 95% CI = [-0.01, 0.21], $p_{\text{one-tailed}} = 0.037$, **Fig. R2**; $N = 49$ (full sample), 0.15 ± 0.13 , $p_{\text{one-tailed}} = 0.137$, not shown in figure] but not during observation ($p > 0.05$). Thus, it is possible that the “borders” of our movement area (indicated on the sandy desert plane) within the larger environment disrupted grid-like codes, pushing towards a 4-fold modulation of the entorhinal cortex signal.

Fig. R2. Results of the 4-fold symmetrical model. Magnitude of symmetrical codes (4-fold) within the entorhinal cortex (arbitrary units, a.u.) during observation and navigation. Error bars reflect the standard error of the mean, s.e.m.; * $p < 0.05$.

Conclusion & action taken:

Altogether, these results strongly suggest that grid-like codes in the entorhinal cortex during observation are not driven by eye movements. We also added the additional correlation analysis to our manuscript. Moreover, we included the analysis regarding the 4-fold symmetrical modulation of the entorhinal cortex signal during navigation and discuss it briefly.

To elucidate the finding of negative grid-like coding in V1 during observation, we performed additional analyses further down (in response to a related comment, see below reviewer 3, comment 3).

Results (page 12, line 367): “Additionally, we tested whether the magnitude of grid-like codes in the entorhinal cortex during observation scaled with the average number of saccades that participants performed. Correlation analysis showed that this was not the case [$N = 37$ (same sample as above); $p_{\text{two-tailed}} = 0.427$], altogether suggesting that our result of significant grid-like codes in the entorhinal cortex when observing the demonstrator’s paths was based on spatial information rather than the direction of saccadic eye movements.”

Results (page 10, line 281): “(...) we also tested for a potential 4-fold signal periodicity since previous work demonstrated such an effect during navigation (He and Brown, 2019). Indeed, we found that the 4-fold symmetrical model yielded significantly increased navigation-related responses in the entorhinal cortex [$N = 43$ (6 outliers excluded), 0.1 ± 0.05 , $t(42) = 1.83$, Cohen’s $d = 0.28$, 95% CI = [-0.01, 0.21], $p_{\text{one-tailed}} = 0.037$, Supplementary Fig. S4; $N = 49$ (full sample), 0.15 ± 0.13 , $p_{\text{one-tailed}} = 0.137$, not shown in figure].”

Discussion (page 18, line 603): “The finding might also be explained by the specific setup of our modified navigation task, which differed from previous setups in several points. Participants were passively moved (after indicating and

confirming their intended walking direction with a button press) and were required to re-trace path trajectories rather than freely navigating towards object locations in virtual reality space.”

Discussion (page 19, line 616): “Interestingly, we found a significant 4-fold symmetrical modulation of the entorhinal cortex signal during navigation, in line with previous work (He and Brown, 2019). A significant 4-fold modulation could reflect increased activation when moving in the cardinal directions (north, south, east, west). It is currently unclear whether such modulation is driven by grid cells but we speculate that the cardinal directions might act as mental coordinate system, allowing us to compare other movement directions with these major axes. He and Brown (2019) reported such 4-fold modulation when barriers compartmentalized the environment and disrupted grid-like coding. Thus, it is possible that the “borders” of our movement area (indicated on the sandy desert plane) within the larger environment disrupted grid-like codes, pushing towards a 4-fold modulation of the entorhinal cortex signal.”

Reviewer comment: “Indeed, I am wondering whether the negative relationship between grid codes and performance reflects that during navigation participants have to translate the observed angles into navigational angles given the viewpoint differences (simply put: observing the demonstrator for instance walk to the right might imply I have to walk left during navigation, if the demonstrator was standing opposite of me).”

We agree with the reviewer that participants likely had to account for differences in viewpoints when re-tracing the previously observed paths during navigation periods (as they would have not been able to successfully complete the task otherwise). That said, we apologize that we are unable to entirely follow the reviewer’s reasoning at this point, as it is not clear to us how a potential viewpoint translation during observation could explain the relationship between increased grid-like codes during observation and worse performance across participants.

Additional analyses:

To elaborate on the brain-behavior relationship, we performed several control analyses helping us to understand that this effect does not stem from eye movements or from other, potentially confounding factors.

First, we assessed whether performance was related to visual signals by calculating the correlation between performance (cumulative distance error in vm) and the number of saccades during observation trials. We found no significant association between performance and the number of saccades across participants [$N = 37$ (reduced sample since eye-tracking data was missing for some participants; $p_{\text{two-tailed}} = 0.385$)]. This is in parallel with our reply directly above, showing no significant relationship between the magnitude of grid-like codes during observation and the average number of saccades ($p_{\text{two-tailed}} = 0.427$).

Second, we addressed whether the effect could be caused by specific aspects of how we had designed the demonstrator’s path trajectories. For instance, it could be the case that grid-like codes during observation were associated with the length of the observed path segments (i.e., shorter paths might lead to weaker grid-like signals compared to longer paths, and shorter paths would be easier to re-trace yielding better performance). However, there was no significant association between the magnitude of entorhinal grid-like codes during observation and the average path length observed across participants [$N = 46$ (3 outliers excluded, same sample as in original analysis), $p_{\text{two-tailed}} = 0.071$], suggesting that the negative brain-behavior relationship was not significantly driven by the length of the observed paths.

Third, we re-calculated our accuracy measure, taking into account that chance performance for re-tracing a given path correctly differs for different areas of the environment (Miller et al., 2018). We then tested whether this alternative method for assessing individual performance affected our result regarding the negative brain-behavior relationship of decreased grid-like magnitudes at better individual performance across participants. Results remained virtually unchanged, once

more showing decreased grid-like codes in participants with better overall accuracy ($N = 47$, same sample as for main analysis: $r_{\text{Pearson}} = -0.43$, 95% CI = $[-0.64, -0.16]$, $p_{\text{two-tailed}} = 0.003$; see also **Fig. R3c**, and please see our reply to reviewer 2, comment 8 for a more detailed description of the approach).

Conclusions & action taken:

In sum, these findings suggest that the brain-behavior relationship is not driven by the number of eye movements produced by the participants during observation, by the task design, or by the type of performance measure used. We partly integrated these analyses into our updated manuscript. A complete description is included in our supplemental materials (see **Supplementary Results S3** for full details; due to space issues, the changes made are not fully reproduced here).

Lastly, we would like to highlight a related finding by Nau and colleagues (2022) who reported stronger fMRI-based directional coding in the parahippocampal gyrus and medial temporal regions (including the posterior-medial entorhinal cortex) specifically in participants who displayed low memory performance in a virtual navigation task where objects had to be placed at their correct location (Nau et al., 2020).

Results (page 10, line 257): *“We performed several control analyses to validate the brain-behavior relationship. First, we could show that the results did not stem from specific path patterns, such as potentially longer paths during low-compared to high-performance trials (the paths of low-performance trials could have crossed the center of the movement area more often, yielding a longer average path length and thus perhaps a stronger grid-like signal, but this was not the case; Supplementary Results S3). Second, we repeated the correlation analysis with a newly-defined performance measure, taking into account that chance performance for re-tracing a given path correctly differed for different endpoints along the circumference of the movement area. Using this normalized accuracy measure (Miller et al., 2018), results remained virtually identical (Supplementary Results S3, Supplementary Fig. S6). Altogether, these analyses corroborated our result of stronger grid-like codes the worse participants performed.”*

Discussion (page 18, line 562): *“This fits with a recent finding by Nau and colleagues (2022) who reported stronger fMRI-based directional coding in the medial temporal lobe (including the posterior-medial entorhinal cortex) in participants who displayed low memory performance in a virtual spatial navigation task (i.e., measured by a larger drop-error when trying to place objects at their correct locations).”*

Reviewer comment: *“Another possibility is that it is related to the rather passive nature of the navigational task, given participants could not control the movement through the arena. Have the authors looked at whether grid-codes appear during rotational movements in the navigation phase?”*

We agree that the passive nature of the task could be a reason why we did not find significant grid-like codes during navigation. We made sure to highlight this possibility more clearly in the revised version.

Discussion (page 18, line 603): *“The finding might also be explained by the specific setup of our modified navigation task, which differed from previous setups in several points. Participants were passively moved (after indicating and confirming their intended walking direction with a button press) and were required to re-trace path trajectories rather than freely navigating towards object locations in virtual reality space.”*

Unfortunately, we are unable to test whether grid-like codes were associated with rotational movements in the navigation phase as these periods were generally short (< 2 seconds), which is also why we had explicitly excluded rotational (and standing) periods from all grid analyses. Besides, it is unclear to us whether we would be able to detect grid-like codes during such periods when no positional change is observed (or when no navigation is taking place).

Reviewer comment: “There are many other possible speculations, e.g. that the planning and memorizing of the movements happened during observation, or that there is some sort of interference between observation and navigation phase, i.e. effects would be visible when considering only the trials with longer intervals between observation and navigation. It is difficult to know.”

As the reviewer states correctly, participants could have planned the upcoming path during observation and most likely memorized the observed paths (in fact, they were explicitly instructed to do so).

We are unsure though about the reviewer’s concern regarding a possible interference between observation and navigation periods. We designed our modified navigation task with jittered inter-trial-intervals that ranged from 2-7 seconds (mean = 5 seconds). Thus, we specifically optimized our trial sequences to be able to separate blood-oxygen-level-dependent (BOLD) signal changes related to observation vs. navigation periods (Amaro and Barker, 2006).

Reviewer comment: “Given all this, the interpretation of the main finding of entorhinal grid codes during observation is difficult in my view. We don’t know the reason for this difference between observation and navigation, and the strong negative signals in V1 and the negative relations to behavior hint at possible confounds.”

While we are open to alternative interpretations (as also included in the discussion and in our comprehensive response to the reviewers), we do not agree that the brain-behavior relationship or the negative V1 signal represent substantial confounds to our main result. Rather, we are convinced that they pose novel and exciting findings.

Lower grid-like codes at better performance resonate with a recent report on a similar relationship between directional coding in the medial temporal lobe and spatial memory performance (Nau et al., 2020), and could be explained through efficient neural coding similar to our recent finding of lower brain activation after memory training that specifically predicted durable memory formation (Wagner et al., 2021). Regarding negative grid-like codes in V1, we provide additional analysis below and discuss possible mechanisms that could drive this effect (please see our reply further down, reviewer 3, comment 3).

To conclude, we present novel and partly also unexpected findings (which we believe are important to show), along with a discussion of potential limitations. We think that exactly such (unexpected) findings can push the field forward and may spark new ideas. This, however, does not question the foundations of our results as we are convinced that our findings are valid.

Reviewer 3, comment 2

- A second issue related to entorhinal grid codes concerns the statistical reliability of the findings. The cross-validation was done within runs, not between. I think the authors should repeat the analysis in a between run manner. The authors exclude different numbers of outliers for different analyses (I could not find a clear definition what constituted an outlier), and using a 1-tailed test found a p-value of .005, which surpassed chance baseline of .008. I think the authors should not exclude any outliers, and use a robust test framework instead if they are worried about violations of distributional assumptions. It would also be helpful to see a whole brain map of where the authors found putative grid effects.

We thank the reviewer for this insightful comment. We structured our reply in three parts.

Part #1, Cross-validation between runs:

With our analysis, we followed the current state-of-the-art approaches used by multiple previous studies (for further discussion, see Stangl et al., 2019) and explicitly opted against an approach that cross-validates between runs. The main reason for this decision was to prevent inherent methodological issues with the analysis of human grid-like signals in fMRI data. First, it has been shown that voxel-based activation patterns of fMRI signals are not stable but display a substantial temporal pattern drift throughout measurement sessions (Alink et al., 2015), which introduces noise in comparisons of voxel-wise activity patterns across longer time periods. Second, noise may be introduced due to potential imprecisions from the spatial co-registration of task runs, where noise is presumably smaller within task runs (i.e., shorter time periods) compared to between task runs (i.e., longer time periods, especially as participants likely moved their heads during scanning breaks between task runs). Third, previous studies showed that the orientation of grid cell firing patterns is not stable but changes under specific conditions (commonly referred to as “remapping”, akin to the remapping of place cell firing fields), but the precise conditions that trigger grid cell remapping in humans are so far unknown. While rodent work suggests that grid cell remapping is triggered by changes in the environment, new hypotheses have emerged regarding event-based remapping of memory and spatial representations in humans (e.g., due to so-called “event-boundaries”, such as the beginning and end of a scanning session/task run), presumably to facilitate a cognitive separation of different experiences even in the same environment (Brunec et al., 2018; Radvansky and Zacks, 2017).

Together, these factors can all contribute to an instability of grid orientations detected in fMRI signals over time. Since the estimation of these grid orientations is at the core of our analyses of human grid-like activity, we reasoned that within-run cross-validation (i.e., considering smaller time windows and thus a lower likelihood for changes in grid orientations) provides the most reliable measurement of grid-like activation, in contrast to between-run cross-validation that is inherently more affected by the abovementioned confounding factors.

Moreover, our own experience from analyzing human grid-like activity in previous studies (for example, Stangl et al., 2018), as well as discussions with other groups who perform these analyses, suggest that estimated grid orientations fluctuate more strongly over longer time periods, and show less stability between versus within fMRI runs. To date, however, the precise reason for this fluctuation is not clear, and future studies will need to determine to what extent factors like general drift in fMRI patterns, remapping of spatial representations, spatial co-registration of images, or an interplay of different factors affects grid-like codes in human fMRI signals.

Conclusion & action taken:

The main aim of the present study was to detect whether grid-like signals in the human brain track movements of other individuals. We did not set out to test any hypotheses regarding the temporal stability or remapping of such signals. For this reason, we think that the research question in this manuscript can be reliably addressed using a within-run cross-validation approach. Based on our findings, future studies will be able to further investigate whether grid-like activity during the observation of others is stable over time, as well as the conditions under which these spatial representations might show remapping.

We now clearly state the rationale for our cross-validation approach and added a more extensive overview of the different methodological aspects to our revised manuscript.

Methods (page 27, line 945): *“We specifically opted for a cross-validation regime within-runs in order to attenuate potential confounds often associated with the analysis of fMRI-based grid-like codes. First, we aimed to minimize noise*

due to signal drifts, as well as noise due to potential imprecisions stemming from the spatial co-registration across task runs. Second, our goal was to avoid changes in grid orientation, or “remapping”, which might occur under specific circumstances. Although rodent work suggests that grid cell remapping is triggered by changes in the environment (Fyhn et al., 2007), the precise conditions that trigger grid cell remapping in humans are so far unknown. One possibility is that memory and spatial representations remap based on the temporal order of events (for example, due to so-called “event-boundaries”, such as the beginning and the end of a scanning session or a task run), presumably to facilitate a cognitive separation of different experiences even in the same environment (Brunec et al., 2018; Radvansky and Zacks, 2017). We did not set out to test any hypotheses regarding the temporal stability of such signals and thus addressed our questions by cross-validating within runs, reasoning that this approach would provide us with the most reliable measure of grid-like activation.”

Part #2, Outlier detection:

We apologize for any unclarity in the description of our outlier detection method. We now made sure to include all details in our revised **Methods** section. To reiterate, outliers (for all analyses in the paper, including all grid analyses) were *a priori* defined as values exceeding a difference of +/- 3 absolute deviations from the median (Leys et al., 2013). We are convinced that outlier exclusion is a necessary step to undertake in order to avoid adding noise to the analyses. However, in the spirit of maximal transparency and reproducibility, we provide all results with and without the exclusion of outliers (see **Results** section and supplementary materials), and provide our raw data on the Open Science Framework³.

We would like to highlight in this context that overall, our approach and the findings it produced is reliable and yielded converging results. This is shown by the numerous control analyses that we had already provided in our initial manuscript, and by the comprehensive reply and additional analyses to the reviewers’ comments, which altogether left our initial results unchanged.

Methods (page 27, line 969): “Grid magnitude values exceeding the median value $\pm 3 \times$ the median absolute deviation were excluded from the analyses (Leys et al., 2013). Results remained stable when basing calculations on the full data set (i.e., not excluding any outliers, see Supplementary Tables S2 and S3) and the raw data is publicly provided (see below).”

Methods (page 29, line 1040): “Grid magnitude values exceeding the median value $\pm 3 \times$ the median absolute deviation were excluded from the analyses (Leys et al., 2013). Results remained stable when basing calculations on the full data set (i.e., not excluding any outliers) and are reported in Supplementary Tables S2 and S3.”

Part #3, Whole-brain analysis of grid-like coding⁴:

We thank the reviewer for this thoughtful comment, and we agree that it would be of interest to determine the precise location of human grid-like codes in a whole-brain analysis. We would like to point out, however, that such a whole-brain analysis of fMRI grid-like activity is methodologically extremely challenging, which presumably is the reason why (to the best of our knowledge) no previous study has performed such an analysis.

The contemporary approach to analyze grid-like activity in human fMRI signals (which has been used by us and a series of other groups in the field) comprises two steps: First, voxel-wise grid orientations are estimated and averaged within an anatomically defined brain region where grid-like activity is expected. Second, this estimated mean grid orientation is tested on unseen fMRI

³ https://osf.io/mhtgp/?view_only=ff7038b65c0345cd9ffa4ccd3813d7ba; Note that the OSF project website is currently set to private and will be made publicly available upon publication of the manuscript.

⁴ In our initial manuscript, we already asked whether the activity of entorhinal grid-like codes during observation was paralleled by whole-brain, voxel-wise activation changes of regions typically involved in spatial navigation and visuospatial processing. Results showed that striatal activation was strong when participants observed the demonstrator walking aligned with their individual grid orientation in the entorhinal cortex (**Fig. 4b**).

data from the same anatomical region to determine the magnitude of grid-like activity within that region. To provide a methodologically sound estimation and test of grid orientations, this is done using a cross-validation regime (in our case, we applied 12-fold cross validation, meaning that the above-described procedure was repeated 12 times on different data subsets, please see our **Methods** section for more details).

For a single brain region (e.g., the entorhinal cortex) and the entire participant sample, this cross-validation procedure takes approximately 72 hours on a powerful computing server (such as the one provided by the University of Vienna that we used for the present study). Performing this procedure on a whole-brain level would dramatically increase the processing time up to many weeks and would require immense computational power and resources (72 hours*approximately 100 000 voxels in the brain). Given that the computing server is shared across different research groups, blocking this valuable resource would create considerable administrative and financial issues.

Even though potential findings of such an analysis may provide interesting additional insights, we conclude that this additional analysis would only be justifiable if it was absolutely necessary to answer a specific research hypothesis. In the present study, we hypothesized that grid-like codes in the entorhinal cortex would be associated with observing others' movement through space and we believe that this hypothesis can be tackled without this additional analysis.

We would like to highlight, however, that a recent study utilized intracranial electrophysiological recordings in human epilepsy patients and provided a map of brain locations where hexadirectional modulation of oscillatory power (indexing grid-like representations) was detected (Chen et al., 2021). While this still cannot be considered a whole-brain approach (since not all brain regions were covered by the electrode contacts), the study of Chen and colleagues can nevertheless provide first insights into the distribution of grid-like activity across (a restricted set of) brain regions. The distribution map (see Fig. S7 in their paper) also highlights the particular importance of the entorhinal cortex, showing that grid-like signals were detected mostly in this region.

Altogether, this supports our targeted (and *a priori* defined) approach of testing for grid-like codes in the entorhinal cortex. However, we encourage future studies to test hypotheses about the presence of grid-like activity in the human spatial navigation network beyond the entorhinal cortex, and how this network supports keeping track of not only oneself but also other individuals.

Action taken:

To stimulate such a discussion, we briefly elaborate on this topic in our updated manuscript.

Discussion (page 17, line 549): *“Given that human grid-like signals have been detected not only in the entorhinal cortex but also in other brain regions (Chen et al., 2021; Constantinescu et al., 2016), it will be interesting to test how regions beyond the entorhinal cortex support the tracking of others.”*

Reviewer 3, comment 3

- Lastly, I believe the paper would benefit if the authors would provide more insights into the V1 effect, and to be more upfront about as well. It would be interesting to understand the negative sign of the visual effects, and whether there could be any possible reason for a possible 30-degree rotated grid re-orientation that could have caused this effect (see my point above about translating between visual and navigational angles). In addition, the authors write “In summary, we found significantly increased grid-like signals in the entorhinal cortex, but not in any of the control regions, when participants were observing and putatively encoding the demonstrator’s paths. “, but I believe is not true?

We thank the reviewer for this comment which prompted us to provide more insights into the V1 effect by performing additional analyses and by discussing the results in more detail.

As possible reasons for negative grid-like codes in V1 during observation, we had explored the following possibilities in our initial manuscript (**Supplementary Results S2**): (1) specific directions could be driving the effect (a strong response in a specific direction compared to smaller responses in other directions); (2) grid magnitudes could be increased for directions that are rotated 30° (resulting in increased grid magnitudes for misaligned compared to aligned movement directions).

To determine whether these aspects actually contributed to our effect, we repeated the main grid analysis but now modelled V1 activation levels with separate regressors for the different movement directions that participants observed.

Analysis approach:

Voxel-wise grid orientations were estimated identically to our main analysis. Specifically, we used a 12-fold cross-validation (CV) regime during which grid orientations were estimated on the path segments of 11 trials (GLM1) and tested on the path segments of the remaining trial (GLM2) within each task run. This was repeated until every trial was tested once. Importantly, GLM2 was now used to contrast directions aligned (0 modulo 60°) versus misaligned (0 modulo 30°) relative to the estimated grid orientation. Thus, V1 activation levels were modelled with separate regressors for the different movement directions that participants observed (instead of parametrically modulating each path segment using the difference between each segment’s translational direction and the participant’s mean grid orientation, as we had done previously, see **Methods**). Next, we averaged the obtained grid magnitudes across CV-folds, runs, and voxels within the V1 ROI. We plotted the parameter estimates (arbitrary units, a.u.) for each directional bin, and also assessed the averaged signal change across all aligned and misaligned directional bins.

Results:

We found that V1 activation levels appeared increased for observed directions that were misaligned with the individual grid orientation (0 modulo 30°) compared to aligned directions (0 modulo 60°). This effect was not driven by single directional bins but appeared relatively consistent across the range of misaligned directions (**Fig. R4a**).

We also assessed the average activation levels across all aligned and misaligned directions (**Fig. R4b**). Activation for misaligned directions (mean \pm s.e.m., 0.84 ± 0.22) was significantly higher than for aligned directions (0.4 ± 0.2 ; paired-sample *t*-test, $N = 58$ (full sample), $t(57) = -2.48$, $p_{\text{two-tailed}} = 0.016$). This explains our finding of negative grid-like codes in V1 during observation.

Conclusions:

Our effects appear related to decreased V1 activity when participants observed the demonstrator walking aligned with their individual grid orientation. Multiple explanations of how grid cell firing activity relates to macroscopic (fMRI-based) signals exist (for a review, see Kunz et al., 2019). For example, increased grid-like coding might be related to the crossing of multiple firing fields (leading to an elevated fMRI signal). In contrast to this, different results can be obtained if repetition suppression mechanisms are at play (for a review see Barron et al., 2016). Grid cells would repeatedly fire during aligned movement and this repetition would cause a suppression of neuronal firing (leading to a decreased fMRI signal). Thus, it is possible that the decreased V1 activity for aligned < misaligned directions could represent hexadirectional signal modulation in the form of neural adaption.

In summary, we found significantly increased and decreased observation-based grid-like codes in the entorhinal cortex and V1, respectively (**Fig. 3cd**). This divergence could potentially suggest different mechanisms of how grid cell activity translates to macroscopic fMRI-based signal changes in different brain regions (Kunz et al., 2019). Additionally, grid-like neural adaption effects in V1 could be related to grid cells in the visual system (it is currently unclear whether grid cells exist in V1, but see a recent preprint reporting on grid cells in V2, Long et al., 2021) or whether effects are caused by grid cell activity upstream (from the entorhinal cortex). We encourage future research to elucidate the relationship of grid-like codes in the medial temporal lobe and visual system.

Fig. R4. V1 activation during aligned and misaligned paths observed. (a) V1 activation levels (quantified by parameter estimates, arbitrary units, a.u.) when observing aligned (0 modulo 60°) and misaligned (0 modulo 30°, indicated in bold) directions, sorted according to the putative grid orientation (0°) and obtained using 12-fold cross-validation. The green line indicates a hypothetical signal modulation as would be the case when assuming increased grid-like codes when observing/walking aligned (0 modulo 60°) with the internal grid orientation. (b) Average V1 activation levels for aligned and misaligned directions observed. Error bars reflect the standard error of the mean, s.e.m; *, $p < 0.05$.

Action taken:

We included the analysis and figure in our manuscript. Below we show an excerpt of the most important text passages, all further details that were included in the supplementary materials are not shown (as the text largely overlap with our description above, but see **Supplementary Results S2**).

Results (page 9, line 222): *“We further explored the finding of negative grid-like codes in V1 during observation with additional analyses. First, results remained stable when using a different, less-nested cross-validation regime (mitigating the potential effect of specific directions that could have affected grid magnitudes disproportionately, Supplementary Results S2). Second, we found that V1 activation levels appeared increased for observed directions that were misaligned with the individual V1 grid orientation (0 modulo 30°) compared to aligned directions (0 modulo 60°). This effect was not driven by single directional bins but appeared relatively consistent across the range of all misaligned directions (Supplementary Fig. S5). We provide an in-depth discussion of this finding in our supplementary materials (Supplementary Results S2).”*

Discussion (page 18, line 584): *“Furthermore, we found a strong effect of negative grid-like codes in V1 during observation (Fig. 3e, Supplementary Fig. S5). V1 activation levels were increased for observed directions that were misaligned with the individual V1 grid orientation (0 modulo 30°) compared to aligned directions (0 modulo 60°). Multiple explanations of how grid cell firing activity relates to macroscopic (fMRI-based) signals exist (for a discussion, see Kunz et al., 2019). For example, one possibility is that grid cells repeatedly fire during aligned movement and that this repetition would cause neural adaption (leading to a relative increase in the fMRI signal for misaligned directions). Signals might potentially be triggered by grid cells in the visual system (although it is currently unclear whether grid cells exist in V1) or by (entorhinal) mechanisms upstream. At present, we do not know which factors drive this effect and thus cannot provide a firm interpretation of why negative grid-like coding in V1 appears associated with tracking others navigating through space. We encourage future research to elucidate this finding, as well as the relationship between grid-like codes in the medial temporal lobe and visual systems.”*

Reviewer comment: *“In summary, we found significantly increased grid-like signals in the entorhinal cortex, but not in any of the control regions, when participants were observing and putatively encoding the demonstrator’s paths. “, but I believe is not true?”*

We were initially alluding to significant increases in grid-like coding (which was not the case for V1) but now changed this statement accordingly.

Results (page 9, line 231): *“In summary, we found significantly increased grid-like signals in the entorhinal cortex (as well as significantly decreased grid-like codes in V1) when participants were observing and putatively encoding the demonstrator’s paths.”*

Reviewer 3, comment 4

A separate major issue is that the social aspect of the navigation is given too much weight in my opinion. While the authors do mention in the discussion that the grid-codes might be tracking relevant non-spatial cues and that their design does not allow to dissociate these, this point should be made clearer in the abstract and introduction. The abstract currently says “Navigating through crowded, dynamically changing social environments requires the ability to keep track of other individuals”, which I think needs to be toned down given the particulars of the experiment.

We acknowledge that we might have put too much weight on the social aspect at certain points of the manuscript and toned down our language accordingly.

Introduction (page 3, line 45): *“Moreover, these grid-like codes were shown to support spatial (Bierbrauer et al., 2020; Doeller et al., 2010; He and Brown, 2019; Julian and Doeller, 2021; Kunz et al., 2015; Stangl et al., 2018) as well as mental self-related navigation (Bellmund et al., 2016; Horner et al., 2016), but whether they also track others’ movement (or the movement of non-social features) through space is unclear.”*

And further,

Introduction (page 3, line 48): *“Here, we propose that entorhinal grid cells (and related grid-like codes) make an essential contribution to (socio-)spatial navigation in humans.”*

Discussion (page 16, line 469): *“In the current study, we investigated whether grid-like codes in the human entorhinal cortex supported (socio-)spatial navigation as participants tracked and subsequently re-traced the paths of a virtual demonstrator.”*

Finally, we highlighted this issue once more in our conclusions:

Discussion (page 20, line 655): “While we are currently unable to answer whether these results are specifically related to social processing, these findings might indicate that grid-like codes could be involved in socio-spatial navigation, concerned with tracking others’ complex and goal-directed movements through space.”

We would, however, prefer to keep the sentence in the abstract as it highlights our motives for designing this experiment and why we set out to test the question of whether grid-like codes play a role during observation.

MINOR Points

Reviewer 3, comment 5

I am not sure whether the cumulative distance error is appropriate for the present experiment, given that participants were forced to walk in straight paths and the accuracy of the second and third segments strongly depended on the accuracy of the first (I assume segment endpoint is the starting point for the next segment, which explains why the 2nd and 3rd segment distance errors have almost 4 times as much cumulative distance). Another issue is that 20 vm is an arbitrary threshold, so the t-test against it is difficult to interpret. I don’t doubt that participants perform above chance, but it would be useful to know what the true chance baseline of a random agent would be? In general, it would be nice to show behavior in main paper.

We addressed this comment in several parts.

Part #1, Path segments:

The reviewer assumes correctly that the endpoint of the first path segment represents the starting point of the next path segment (and so forth). We clarified this.

Methods (page 22, line 754): “Starting points of paths were randomly generated within a sector and each sector hosted a start position twice within the same run, whereby the endpoint of the first path segment represented the starting point of the next path segment (and so forth).”

While it is true that error can compound across multiple path segments, it is still possible for participants to “correct” an error on the 2nd or 3rd path segments (that is, participants are still able to navigate towards the correct endpoints of the remaining path segments even if they showed poor performance at the start).

Part #2, Feedback threshold:

The choice of a 20 vm feedback threshold was initially inspired by previous work from Stangl and colleagues who investigated grid-like coding as participants navigated through a virtual reality environment to place objects at their correct location (Stangl et al., 2018). We subsequently confirmed that this threshold was appropriate by performing a small behavioral pre-test (with $N = 4$ participants) which showed that participants’ performance stabilized at a cumulative distance error of ~20 vm after one training run. We clarified this in the manuscript.

Methods (page 22, line 730): “The choice of a 20 vm feedback threshold was inspired by previous work from Stangl and colleagues (2018) who investigated grid-like coding as participants navigated through a virtual reality environment to place objects at their correct location. We subsequently confirmed that this threshold was appropriate by performing a small behavioral pre-test (with $N = 4$ participants) which showed that participants’ performance stabilized at a cumulative distance error of ~20 vm after one training run.”

We would like to highlight though that this threshold did not affect the main analyses or the brain-behavior associations that we report. In other words, any analyses pertaining to an association

between grid-like codes and behavioral performance are focused on individual differences, irrespective of how the feedback threshold was defined.⁵

Part #3, Performance of a random agent:

We thank the reviewer for this excellent suggestion. We performed the following analysis to assess chance performance of a random agent.

Analysis approach:

We pooled all paths that were presented to participants during the observation period (a path consisted of three path segments). We then randomly selected paths and simulated the performance of a “random agent”. That is, we randomly chose three endpoints (for each of the three path segments) out of all possible endpoints along the circumference of the movement area (spanning the circular border in 1°-degree steps). We then calculated the Euclidean distance between each randomized endpoint and the “correct” endpoint of a specific path segment that was selected from the pool of possible paths, averaged across the three error distances, and accumulated a permutation distribution by iterating over these steps 5000 times. Next, we determined the performance value (cumulative distance error, vm) at the 5th percentile (39.3 vm), representing the chance level of a random participant. Finally, we determined whether the observed group performance was below this permutation chance level (see **Fig. R5** below).

Results and action taken:

Group performance (mean ± s.e.m., 17.57 ± 0.54 vm, calculated as the average distance between the demonstrator’s and the participant’s endpoints across the three different segments of a given path, **Fig. 1d**) was significantly below the permutation-based chance level of ~39 vm (one-sample t-test, $N = 58$, $t(57) = -39.6$, $p_{\text{two-tailed}} < 0.0001$). We added this analysis to our updated manuscript.

Results (page 5, line 108): *“To determine whether participants performed better than chance, we simulated the performance of a random agent using permutation testing (Methods). Navigation performance in our sample was significantly below the permutation-based chance level of 39.3 vm (one-sample t-test, $N = 58$, $t(57) = -39.6$, $p_{\text{two-tailed}} < 0.0001$; Fig. 1e).”*

Results (page 23, line 764): *“To determine whether participants performed better than chance (i.e., whether they were able to re-trace the demonstrator’s paths more accurately than a random agent), we performed permutation-based testing. We pooled all paths that were presented to participants during the observation period, randomly selected paths, and simulated the performance of a “random agent”. That is, we randomly chose three endpoints (for each of the three path segments) out of all possible endpoints along the circumference of the movement area (spanning the circular border in 1°-degree steps). We then calculated the Euclidean distance between each randomized endpoint and the “correct” endpoint of a specific path segment that was selected from the pool of possible paths, averaged across the three error distances, and accumulated a permutation distribution by iterating over these steps 5000 times (Fig. 1d). We then determined the performance value (cumulative distance error, vm) at the 5th percentile (39.3 vm), representing the chance level of a random participant. Finally, we tested whether the observed group performance was below this permutation-based chance level using a one-sample t-test.”*

Part #4, Behavioral performance in main paper:

As suggested by the reviewer, we included the behavioral performance as well as the comparison to performance of a random agent in our updated **Fig. 1** (not shown here).

⁵ Additionally, we re-calculated performance as normalized accuracy values taking into account that the probability to correctly re-trace a given path segment can be different in different areas of the environment (see reviewer 2, comment 8). This did not affect our main results but yielded virtually identical findings regarding the negative brain-behavior association (we included these analyses also in our updated supplemental materials, changes are not shown here).

Fig. R5. Performance of a random agent. Permutation distribution obtained from simulating paths of a random agent ($N = 5000$). Chance level was calculated as the 5th percentile of the distribution, yielding a value of ~39 virtual meters (vm; indicated as red line). The dashed line indicates the value of the observed group mean (17.6 vm). Group performance was significantly below the permutation-based chance level ($p_{\text{two-tailed}} < 0.0001$).

Reviewer 3, comment 6

The finding of negative grid codes in left entorhinal cortex during navigation is intriguing, especially as during observation grid-codes are left lateralised. The authors expand a bit on the possible reasons for finding negative grid codes in V1 during the observation phase, but what are possible interpretations in the case of the entorhinal cortex?

From what we understand, results of negative grid-like coding in the visual or entorhinal cortices emerged as surprising side-findings not only from our analyses, but had also emerged unexpectedly in previous work (Horner et al., 2016). Additionally, there is an ongoing debate in the field on how grid cell firing relates to macroscopic (fMRI-based) signals (Kunz et al., 2019).

Thus, we are generally unsure about what could cause a switch from increased to decreased grid-like codes in the left entorhinal cortex during observation and navigation, respectively. One possibility could be that increased grid-like codes during observation are suppressed during subsequent navigation, as could be explained by a neural adaption mechanism. However, we directly tested whether neural adaption could have led to a decrease in the overall entorhinal cortex signal during navigation but this was not the case (reviewer 1, comment 2).

We have included a more in-depth discussion of this result in our updated manuscript.

Supplementary Materials (page 5, line 194): “Results indicating negative grid-like coding in the visual or entorhinal cortices emerged as surprising side-findings not only from our analyses, but had already been reported as equally unexpected findings in previous work (Horner et al., 2016). Additionally, there is an ongoing debate in the field on how grid cell firing relates to macroscopic (fMRI-based) signals (Kunz et al., 2019). Thus, we are generally unsure about what could cause a switch from increased to decreased grid-like codes in the left entorhinal cortex during observation and navigation, respectively. One possibility could be that increased grid-like codes during observation are suppressed during subsequent navigation, as could be explained by a neural adaption mechanism. However, we directly tested whether neural adaption could have led to a decrease in the overall entorhinal cortex signal during navigation but this was not the case (Supplementary Results S5). We encourage future studies to scrutinize the finding of negative grid-like codes and to determine how fMRI-based grid-like codes relate to underlying grid cell firing.”

Reviewer 3, comment 7

The relationship between brain activation patterns and performance could be described more clearly on page 15, line 487/488: “[...] connectivity patterns, were positively, rather than negatively associated with performance (i.e., distance error) [...]”

We thank the reviewer for pointing this out, and we have clarified this in the revised manuscript.

Discussion (page 17, line 558): *“For instance, we found that entorhinal grid-like codes, as well as co-activation and entorhinal-cortical connectivity that were timed to the activity of entorhinal grid-like codes, were associated with performance in the modified navigation task. In other words, participants with higher grid magnitudes during observation performed worse when re-tracing the demonstrator’s paths (Fig. 3f).”*

Reviewer 3, comment 8

How was the (implicit) baseline determined for the fMRI analyses?

We apologize for being unclear.

For the standard univariate (grid-independent) analysis, all events during the fMRI task were explicitly modelled (cues, feedback periods, observation and navigation periods). Thus, the implicit baseline consisted of the fixation cross.

For the grid-based analysis, we explicitly modelled all path segments (i.e., translation periods) during observation and navigation periods, as well as cues and feedback periods (collapsed into one regressor of no interest). Rotation and standing periods were not explicitly modelled as these typically included short events (< 2 s). Thus, the implicit baseline consisted of fixation periods as well as short rotation/standing events.

We clarified these aspects in the manuscript.

Methods (page 25, line 856): *“(...) This included events such as orientation adjustments (rotations), walked path segments (translation periods), and time periods during which no movement occurred (short standing periods in-between). Thus, the implicit baseline consisted of the fixation cross.”*

Methods (page 27, line 932): *“(...) Orientation adjustments (rotations) and time periods during which no movement occurred (standing periods) were not explicitly modelled as these durations were typically very short (< 2 seconds). The implicit baseline thus consisted of fixation periods as well as short rotation/standing events.”*

Reviewer 3, comment 9

The concept of re-using the same neural structures for the (planning of) own movement and tracking that of others strongly reminds me of mirror neurons. Can authors comment on how these relate to each other?

The reviewer raises an intriguing point.

Mirror neurons both fire when a specific action is observed and when the same action is performed. They were first discovered in non-human primates (Rizzolatti et al., 1996) and subsequently were shown to span across a parieto-frontal network, including the precentral gyrus, adjacent lateral prefrontal and inferior parietal regions (for a review, see Rizzolatti and Sinigaglia, 2010). Crucially, mirror neurons are regarded as central to our understanding of other’s actions, tying into shared representations that support socio-cognitive functions, on which we have worked extensively (Lamm et al., 2016).

It is possible that mirror neuron mechanisms play a role in the observation of the demonstrator’s paths, their memorization, the planning and the execution during the subsequent navigation period. For instance, similar neuronal firing during observation/navigation periods (i.e., mirror

neuron activity) within the abovementioned fronto-parietal network might indicate that participants observed the demonstrator's paths to the extent that they were successfully able to re-trace them.

However, this reflects pure speculation as we are unable to directly test whether mirror neurons were actually engaged in our task. Additionally, although there is evidence that the activity of mirror neurons is modulated by the spatial location of actions (Caggiano et al., 2009), it is currently unclear whether and how such signals would interact with grid-like coding in the entorhinal cortex. We would thus prefer not to speculate about this aspect in the paper.

Reviewer 3, comment 10

When reporting that grid orientations during observation did not match those during navigation it would also be interesting to see the distribution of grid orientations across participants.

Thank you for this suggestion.

We extracted the individual grid orientations of all entorhinal cortex voxels during observation and navigation periods (based on data from analysis presented in **Supplementary Results S4**). We then calculated the circular mean across the different cross-validation folds and task runs and plotted the grid orientations for each participant and condition (**Fig. R6**). This data is now included in **Supplementary Fig. S7**.

Fig. R6. Different grid orientations during observation and navigation in the entorhinal cortex. (a) Individual grid orientations plotted for observation (outer ring) and navigation (inner ring) conditions. Color variations indicate the different participants. **(b)** Distribution of differences in grid orientation angles between observation/navigation across the sample. **(c)** Differences in grid orientation angles between observation/navigation visualized once more. Data matches the distribution in (b), color values match those in (a).

References

- Alink, A., Walther, A., Krugliak, A., Bosch, J.J.F. van den, and Kriegeskorte, N. (2015). Mind the drift - improving sensitivity to fMRI pattern information by accounting for temporal pattern drift. *BioRxiv* 032391. <https://doi.org/10.1101/032391>.
- Amaro, E., and Barker, G.J. (2006). Study design in fMRI: basic principles. *Brain Cogn* 60, 220–232. <https://doi.org/10.1016/j.bandc.2005.11.009>.
- Amodio, D.M., and Frith, C.D. (2006). Meeting of minds: the medial frontal cortex and social cognition. *Nature Reviews Neuroscience* 7, 268–277. <https://doi.org/10.1038/nrn1884>.
- Barron, H.C., Garvert, M.M., and Behrens, T.E.J. (2016). Repetition suppression: A means to index neural representations using BOLD? *Philosophical Transactions of the Royal Society B: Biological Sciences* 371. <https://doi.org/10.1098/rstb.2015.0355>.
- Bellmund, J.L., Deuker, L., Navarro Schröder, T., and Doeller, C.F. (2016). Grid-cell representations in mental simulation. *Elife* 5. <https://doi.org/10.7554/eLife.17089>.
- Bierbrauer, A., Kunz, L., Gomes, C.A., Luhmann, M., Deuker, L., Getzmann, S., Wascher, E., Gajewski, P.D., Hengstler, J.G., Fernandez-Alvarez, M., et al. (2020). Unmasking selective path integration deficits in Alzheimer’s disease risk carriers. *Science Advances* 6, eaba1394. <https://doi.org/10.1126/sciadv.aba1394>.
- Brunec, I.K., Moscovitch, M., and Barense, M.D. (2018). Boundaries Shape Cognitive Representations of Spaces and Events. *Trends Cogn Sci* 22, 637–650. <https://doi.org/10.1016/j.tics.2018.03.013>.
- Caggiano, V., Fogassi, L., Rizzolatti, G., Thier, P., and Casile, A. (2009). Mirror Neurons Differentially Encode the Peripersonal and Extrapersonal Space of Monkeys. *Science* (1979) 324, 403–406. <https://doi.org/10.1126/science.1166818>.
- Chen, D., Kunz, L., Lv, P., Zhang, H., Zhou, W., Liang, S., Axmacher, N., and Wang, L. (2021). Theta oscillations coordinate grid-like representations between ventromedial prefrontal and entorhinal cortex. *Science Advances* 7. <https://doi.org/10.1126/sciadv.abj0200>.
- Constantinescu, A.O., O’Reilly, J.X., and Behrens, T.E.J. (2016). Organizing conceptual knowledge in humans with a gridlike code. *Science* 352, 1464–1468. <https://doi.org/10.1126/science.aaf0941>.
- Couey, J.J., Witoelar, A., Zhang, S.J., Zheng, K., Ye, J., Dunn, B., Czajkowski, R., Moser, M.B., Moser, E.I., Roudi, Y., et al. (2013). Recurrent inhibitory circuitry as a mechanism for grid formation. *Nature Neuroscience* 2013 16:3 16, 318–324. <https://doi.org/10.1038/nn.3310>.
- Danjo, T., Toyozumi, T., and Fujisawa, S. (2018). Spatial representations of self and other in the hippocampus. *Science* (1979) 359, 213–218. <https://doi.org/10.1126/science.aao3898>.
- Daselaar, S.M., Prince, S.E., and Cabeza, R. (2004). When less means more: deactivations during encoding that predict subsequent memory. *Neuroimage* 23, 921–927. <https://doi.org/10.1016/j.neuroimage.2004.07.031>.
- Daselaar, S.M., Prince, S.E., Dennis, N.A., Hayes, S.M., Kim, H., and Cabeza, R. (2009). Posterior midline and ventral parietal activity is associated with retrieval success and encoding failure. *Front Hum Neurosci* 3, 13. <https://doi.org/10.3389/neuro.09.013.2009>.
- Doeller, C.F., Barry, C., and Burgess, N. (2010). Evidence for grid cells in a human memory network. *Nature* 463, 657–661. <https://doi.org/10.1038/nature08704>.
- Dotson, N.M., and Yartsev, M.M. (2021). Nonlocal spatiotemporal representation in the hippocampus of freely flying bats. *Science* (1979) 373, 242–247. <https://doi.org/10.1126/science.abg1278>.
- Fyhn, M., Hafting, T., Treves, A., Moser, M., and Moser, E. (2007). Hippocampal remapping and grid realignment in entorhinal cortex. *Nature* 446, 190–194. .
- Gaesser, B., Hirschfeld-Kroen, J., Wasserman, E.A., Horn, M., and Young, L. (2019). A role for the medial temporal lobe subsystem in guiding prosociality: the effect of episodic processes on willingness to help others. *Social Cognitive and Affective Neuroscience* <https://doi.org/10.1093/scan/nsz014>.
- Gallagher, H.L., Happé, F., Brunswick, N., Fletcher, P.C., Frith, U., and Frith, C.D. (2000). Reading the mind in cartoons and stories: an fMRI study of “theory of mind” in verbal and nonverbal tasks. *Neuropsychologia* 38, 11–21. [https://doi.org/10.1016/s0028-3932\(99\)00053-6](https://doi.org/10.1016/s0028-3932(99)00053-6).
- Gallagher, H.L., Jack, A.I., Roepstorff, A., and Frith, C.D. (2002). Imaging the intentional stance in a competitive game. *Neuroimage* 16, 814–821. .
- Hafting, T., Fyhn, M., Molden, S., Moser, M.-B., and Moser, E.I. (2005). Microstructure of a spatial map in the entorhinal cortex. *Nature* 436, 801–806. <https://doi.org/10.1038/nature03721>.
- He, Q., and Brown, T.I. (2019). Environmental Barriers Disrupt Grid-like Representations in Humans during Navigation. *Curr Biol* 29, 2718–2722.e3. <https://doi.org/10.1016/j.cub.2019.06.072>.
- Henderson, J.M. (2017). Gaze Control as Prediction. *Trends in Cognitive Sciences* 21, 15–23. <https://doi.org/10.1016/j.tics.2016.11.003>.
- Horner, A.J., Bisby, J.A., Zotow, E., Bush, D., and Burgess, N. (2016). Grid-like Processing of Imagined Navigation. *Current Biology* 26, 842–847. <https://doi.org/10.1016/j.cub.2016.01.042>.

Huijbers, W., Vannini, P., Sperling, R.A., C M, P., Cabeza, R., and Daselaar, S.M. (2012). Explaining the encoding/retrieval flip: memory-related deactivations and activations in the posteromedial cortex. *Neuropsychologia* 50, 3764–3774. <https://doi.org/10.1016/j.neuropsychologia.2012.08.021>.

Jacobs, J., Weidemann, C.T., Miller, J.F., Solway, A., Burke, J.F., Wei, X.-X., Suthana, N., Sperling, M.R., Sharan, A.D., Fried, I., et al. (2013). Direct recordings of grid-like neuronal activity in human spatial navigation. *Nature Neuroscience* 16, 1188–1190. <https://doi.org/10.1038/nn.3466>.

Julian, J.B., and Doeller, C.F. (2021). Remapping and realignment in the human hippocampal formation predict context-dependent spatial behavior. *Nat Neurosci* 24, 863–872. <https://doi.org/10.1038/s41593-021-00835-3>.

Julian, J.B., Keinath, A.T., Frazzetta, G., and Epstein, R.A. (2018). Human entorhinal cortex represents visual space using a boundary-anchored grid. *Nature Neuroscience* 21, 191–194. <https://doi.org/10.1038/s41593-017-0049-1>.

van Kesteren, M.T., Beul, S.F., Takashima, A., Henson, R.N., Ruiter, D.J., and Fernandez, G. (2013). Differential roles for medial prefrontal and medial temporal cortices in schema-dependent encoding: from congruent to incongruent. *Neuropsychologia* 51, 2352–2359. <https://doi.org/10.1016/j.neuropsychologia.2013.05.027> S0028-3932(13)00184-X [pii].

van Kesteren, M.T.R., Ruiter, D.J., Fernández, G., and Henson, R.N. (2012). How schema and novelty augment memory formation. *Trends Neurosci* 35, 211–219. <https://doi.org/10.1016/j.tins.2012.02.001>.

Killian, N.J., Jutras, M.J., and Buffalo, E.A. (2012). A map of visual space in the primate entorhinal cortex. *Nature* 491, 761–764. <https://doi.org/10.1038/nature11587>.

King, D.R., de Chastelaine, M., Elward, R.L., Wang, T.H., and Rugg, M.D. (2015). Recollection-Related Increases in Functional Connectivity Predict Individual Differences in Memory Accuracy. *Journal of Neuroscience* 35, 1763–1772. .

Kragel, J.E., and Voss, J.L. (2022). Looking for the neural basis of memory. *Trends in Cognitive Sciences* 26, 53–65. <https://doi.org/10.1016/j.tics.2021.10.010>.

Kunz, L., Schroder, T.N., Lee, H., Montag, C., Lachmann, B., Sariyska, R., Reuter, M., Stirnberg, R., Stocker, T., Messing-Floeter, P.C., et al. (2015). Reduced grid-cell-like representations in adults at genetic risk for Alzheimer's disease. *Science* (1979) 350, 430–433. <https://doi.org/10.1126/science.aac8128>.

Kunz, L., Maidenbaum, S., Chen, D., Wang, L., Jacobs, J., and Axmacher, N. (2019). Mesoscopic Neural Representations in Spatial Navigation. *Trends in Cognitive Sciences* 23, 615–630. <https://doi.org/10.1016/j.tics.2019.04.011>.

Lamm, C., Bukowski, H., and Silani, G. (2016). From shared to distinct self–other representations in empathy: evidence from neurotypical function and socio-cognitive disorders. *Philosophical Transactions of the Royal Society B: Biological Sciences* 371, 20150083. <https://doi.org/10.1098/rstb.2015.0083>.

Leys, C., Ley, C., Klein, O., Bernard, P., and Licata, L. (2013). Detecting outliers: Do not use standard deviation around the mean, use absolute deviation around the median. *Journal of Experimental Social Psychology* 49, 764–766. <https://doi.org/10.1016/j.jesp.2013.03.013>.

Long, X., Deng, B., Cai, J., Chen, Z.S., and Zhang, S.-J. (2021). A compact spatial map in V2 visual cortex. *BioRxiv* 2021.02.11.430687. <https://doi.org/10.1101/2021.02.11.430687>.

Maass, A., Berron, D., Libby, L.A., Ranganath, C., and Düzel, E. (2015). Functional subregions of the human entorhinal cortex. *Elife* 4, 1–20. <https://doi.org/10.7554/eLife.06426>.

Maidenbaum, S., Miller, J., Stein, J.M., and Jacobs, J. (2018). Grid-like hexadirectional modulation of human entorhinal theta oscillations. *Proc Natl Acad Sci U S A* 115, 10798–10803. <https://doi.org/10.1073/pnas.1805007115>.

Miller, J., Watrous, A.J., Tsitsiklis, M., Lee, S.A., Sheth, S.A., Schevon, C.A., Smith, E.H., Sperling, M.R., Sharan, A., Asadi-Pooya, A.A., et al. (2018). Lateralized hippocampal oscillations underlie distinct aspects of human spatial memory and navigation. *Nature Communications* 9, 2423. <https://doi.org/10.1038/s41467-018-04847-9>.

Nau, M., Navarro Schröder, T., Bellmund, J.L.S., and Doeller, C.F. (2018). Hexadirectional coding of visual space in human entorhinal cortex. *Nature Neuroscience* 21, 188–190. <https://doi.org/10.1038/s41593-017-0050-8>.

Nau, M., Navarro Schröder, T., Frey, M., and Doeller, C.F. (2020). Behavior-dependent directional tuning in the human visual-navigation network. *Nature Communications* 11, 3247. <https://doi.org/10.1038/s41467-020-17000-2>.

Omer, D.B., Maimon, S.R., Las, L., and Ulanovsky, N. (2018). Social place-cells in the bat hippocampus. *Science* (1979) 359, 218–224. <https://doi.org/10.1126/science.aao3474>.

Pertsov, Y., Avidan, G., and Zohary, E. (2009). Accumulation of visual information across multiple fixations. *Journal of Vision* 9. <https://doi.org/10.1167/9.10.2>.

Poldrack, R.A. (2007). Region of interest analysis for fMRI. *Soc Cogn Affect Neurosci* 2, 67–70. <https://doi.org/10.1093/scan/nsm006>.

Radvansky, G.A., and Zacks, J.M. (2017). Event boundaries in memory and cognition. *Current Opinion in Behavioral Sciences* 17, 133–140. <https://doi.org/10.1016/j.cobeha.2017.08.006>.

Renninger, L.W., Verghese, P., and Coughlan, J. (2007). Where to look next? Eye movements reduce local uncertainty. *Journal of Vision* 7, 6–6. <https://doi.org/10.1167/7.3.6>.

- Rizzolatti, G., and Sinigaglia, C. (2010). The functional role of the parieto-frontal mirror circuit: interpretations and misinterpretations. *Nature Reviews Neuroscience* 11, 264–274. <https://doi.org/10.1038/nrn2805>.
- Rizzolatti, G., Fadiga, L., Gallese, V., and Fogassi, L. (1996). Premotor cortex and the recognition of motor actions. *Cognitive Brain Research* 3, 131–141. [https://doi.org/10.1016/0926-6410\(95\)00038-0](https://doi.org/10.1016/0926-6410(95)00038-0).
- Rugg, M.D., and Vilberg, K.L. (2012). Brain networks underlying episodic memory retrieval. *Curr Opin Neurobiol* 23, 255–260. .
- Saxe, R., and Powell, L.J. (2006). It's the Thought That Counts. *Psychological Science* 17, 692–699. <https://doi.org/10.1111/j.1467-9280.2006.01768.x>.
- Spreng, R.N., and Mar, R.A. (2012). I remember you: A role for memory in social cognition and the functional neuroanatomy of their interaction. *Brain Research* 1428, 43–50. <https://doi.org/10.1016/j.brainres.2010.12.024>.
- Stangl, M., Achtzehn, J., Huber, K., Dietrich, C., Tempelmann, C., and Wolbers, T. (2018). Compromised Grid-Cell-like Representations in Old Age as a Key Mechanism to Explain Age-Related Navigational Deficits. *Current Biology* 28, 1108-1115.e6. <https://doi.org/10.1016/J.CUB.2018.02.038>.
- Stangl, M., Wolbers, T., and Shine, J.P. (2019). Population-Level Analysis of Human Grid Cell Activation. (*Humana*, New York, NY), pp. 257–279.
- Staudigl, T., Leszczynski, M., Jacobs, J., Sheth, S.A., Schroeder, C.E., Jensen, O., and Doeller, C.F. (2018). Hexadirectional Modulation of High-Frequency Electrophysiological Activity in the Human Anterior Medial Temporal Lobe Maps Visual Space. *Current Biology* 28, 3325-3329.e4. <https://doi.org/10.1016/j.cub.2018.09.035>.
- Wagner, I.C., van Buuren, M., Bovy, L., and Fernandez, G. (2016). Parallel Engagement of Regions Associated with Encoding and Later Retrieval Forms Durable Memories. *Journal of Neuroscience* 36, 7985–7995. <https://doi.org/10.1523/JNEUROSCI.0830-16.2016>.
- Wagner, I.C., Rütgen, M., and Lamm, C. (2020). Pattern similarity and connectivity of hippocampal-neocortical regions support empathy for pain. *Social Cognitive and Affective Neuroscience* 15, 273–284. <https://doi.org/10.1093/scan/nsaa045>.
- Wagner, I.C., Konrad, B.N., Schuster, P., Weisig, S., Repantis, D., Ohla, K., Kühn, S., Fernández, G., Steiger, A., Lamm, C., et al. (2021). Durable memories and efficient neural coding through mnemonic training using the method of loci. *Science Advances* 7. <https://doi.org/10.1126/sciadv.abc7606>.
- Watrous, A.J., Tandon, N., Conner, C.R., Pieters, T., and Ekstrom, A.D. (2013). Frequency-specific network connectivity increases underlie accurate spatiotemporal memory retrieval. *Nat Neurosci* 16, 349–356. <https://doi.org/10.1038/nn.3315>.

REVIEWER COMMENTS

Reviewer #1 (Remarks to the Author):

This is my second time reviewing this manuscript by Wagner and colleagues. Overall I believe the authors have done a commendable job in addressing my previous concerns. While I don't necessarily agree with some of their interpretations, I do feel the authors have done a great job explaining their logic and are very transparent in their methods and results.

Reviewer #2 (Remarks to the Author):

The authors addressed all my concerns from the first round of reviews and I have no further comments.

Reviewer #3 (Remarks to the Author):

The authors have addressed some of my previous concerns extensively. I thank the authors for their work; in many cases these additional analyses offer important additional insights into the reported findings.

But I do remain somewhat skeptical in some cases.

1. I do not find the argument regarding the within-run cross validation convincing. First, the idea of cross-validation is to have *independent* sets of data. In light of temporally autocorrelated fMRI noise, data from different runs is the only safe way to ensure independence – which is why between runs analyses are the gold standard for any cross-validation analysis. The authors have not provided any evidence that their folds are truly independent. Second, the instability of grid cell orientation argument is also somewhat weak, given that grid cells are known to be rather stable (see the original paper from Doeller et al. Nature, 2008; or Kunz et al. Science, 2015 for a more recent example; the animal literature suggest a similar stability (see the orig. work from Hafting et al. Nature, 2005).

2. I also struggle with the outlier exclusion approach. 3 SDs is a rather low cutoff, and I still believe a robust test framework would be highly preferable to the variable amounts of outlier exclusions. I simply have never come across a paper that has excluded participants on a per-analysis basis, resulting a different Ns for every other analysis.

3. I am not sure if the additional analyses really rule out that there is a (possibly visual) confound that increases the grid-like signal strength during observation. However, the authors have quite exhaustively tried to account for possible confounds (due to eye movements) and it doesn't seem like there are any additional control analyses that offer themselves. I nevertheless suggest the authors to use nuanced and toned-down language, in particular in the abstract. The absence of grid-like signals during navigation, the negative relation to behavior and the negative V1 grid like signals should be mentioned, and the difficulties arising for the interpretation of the entorhinal signals should be presented more prominently too.

4. I feel this point about the social aspects could be addressed better. The authors did tone down the language they used; however, the social aspect is still emphasized quite strongly throughout the manuscript. I find this problematic, given that the experiment lacks an appropriate non-social control condition that would allow to establish whether the effects found are specific to the 'social' aspect of the task.

Reviewer #1:

This is my second time reviewing this manuscript by Wagner and colleagues. Overall I believe the authors have done a commendable job in addressing my previous concerns. While I don't necessarily agree with some of their interpretations, I do feel the authors have done a great job explaining their logic and are very transparent in their methods and results.

Reviewer #2:

The authors addressed all my concerns from the first round of reviews and I have no further comments.

Reviewer #3:

The authors have addressed some of my previous concerns extensively. I thank the authors for their work; in many cases these additional analyses offer important additional insights into the reported findings.

But I do remain somewhat skeptical in some cases.

Author reply:

We thank the reviewer for the critical acclaim of our revisions and the additional time and thought invested in clarifying the few remaining concerns.

1. I do not find the argument regarding the within-run cross validation convincing. First, the idea of cross-validation is to have **independent** sets of data. In light of temporally autocorrelated fMRI noise, data from different runs is the only safe way to ensure independence – which is why between runs analyses are the gold standard for any cross-validation analysis. The authors have not provided any evidence that their folds are truly independent. Second, the instability of grid cell orientation argument is also somewhat weak, given that grid cells are known to be rather stable (see the original paper from Doeller et al. Nature, 2008; or Kunz et al. Science, 2015 for a more recent example; the animal literature suggest a similar stability (see the orig. work from Hafting et al. Nature, 2005).

Our main goal was to probe grid-like codes during observation. We explicitly opted for cross-validation *within* task runs since we expected higher temporal stability of grid orientations within rather than between runs, and since it was not our intention to investigate temporal stability over longer time periods.

Comparison to previous work:

The reviewer states that grid orientations are known to be rather stable. We respectfully disagree with this notion and would like to explain further.

First, it has been shown that several factors can affect the fMRI signal over longer time periods (e.g., within one fMRI session) which makes it difficult to compare between separate task runs. These factors include substantial signal drifts of fMRI-based activation patterns over time (Alink et al., 2015) as well as potential imprecisions in the spatial co-registration between different task runs. Most importantly, grid orientations can “remap” even in the same spatial environment (Radvansky and Zacks, 2017; Brunec et al., 2018) and this could be triggered by the end of one task run and the beginning of the next (indicating an “event boundary”). The precise reason for fluctuations in grid orientations are currently unclear. Given these specific constraints, cross-validation *within* runs has been used as the method of choice by multiple previous studies (see also Stangl et al., 2019 for a discussion). We opted for the same approach to be able to compare to previous work and we even showed that our results were robust across different types of cross-validation regimes.

Second, the reviewer mentioned the work by Doeller and colleagues (2010)¹ and Kunz and colleagues (2015). The latter study indeed used cross-validation *between* different runs (6 task runs, grid orientation estimated on runs 1, 3 and 5 and tested on the remaining). However, the temporal stability of grid orientations in their study was between 50-60% (chance level = 50%), which actually shows that grid orientations are not particularly stable. Although Doeller and colleagues (2010) used the term “runs”, they refer to navigation periods (aligned/misaligned with the internal grid orientation) and to periods during which human participants navigated with different running speeds (fast/medium/slow). To the best of our knowledge, the authors make no statement claiming that they performed cross-validation across separate task runs, which is why this paper does not show the stability of grid orientations over time.

It is important to mention that our modified navigation task is novel and entirely different compared to other tasks that are typically used to probe entorhinal grid-like codes during navigation (such as the object-location task used by Doeller et al., 2010; Kunz et al., 2015). In our study, participants observed short paths on each trial (consisting of three path segments each) and re-traced them thereafter. The switch between observation/navigation periods and the change from one path to the next could have enforced the perception of “event boundaries”, potentially contributing to “remapping” of grid orientations and generally lower temporal stability. Indeed, while we found no significant difference in the temporal stability of grid orientations between observation/navigation periods, temporal stability within runs was generally low (Supplementary Fig. 8, already reported as part of our initial submission). We expected this “remapping” to also occur between task runs as participants completed one task run and started the next one after a short break. This is the reason why we opted for cross-validation within runs.

Third, the reviewer compares to electrophysiological recordings from animals. We believe that this comparison is somewhat problematic since it is unclear how electrophysiological recordings of grid cell firing in rats are related to fMRI-based grid-like codes in humans. In fact, based on the comments of reviewers 1+2 in the previous round of revisions, we took special care to refrain from any direct comparisons throughout the paper. Additionally, any results on the stability of grid orientations are likely heavily influenced by the large amount of training that experimental animals undergo (Hafting et al., 2005: animals underwent multiple training sessions, although it is unclear how long the training period lasted; Doeller et al., 2010: animals were trained over a minimum of 3 days). In our study, participants completed only one task familiarization run directly before the start of the modified navigation task.

Temporal autocorrelation of the fMRI signal:

The reviewer mentions temporally autocorrelated fMRI noise within the same run as one reason to perform cross-validation between runs. We did not compare absolute fMRI signal intensity values across trials of the same run, which is why temporally autocorrelated fMRI noise is not at issue here. Rather, we tested whether entorhinal cortex signal was modulated by the individual grid orientation as participants observed movement aligned or misaligned with the grid orientation. Temporal autocorrelation would have decreased the difference between aligned and misaligned observation periods and would have resulted in no significant grid magnitudes. Opposite to this, we found significantly increased grid-like codes during observation.

¹ The reviewer mentions Doeller et al., *Nature*, 2008 but we believe that they actually meant to refer to the 2010 publication.

Cross-validation between runs:

Nevertheless, we repeated the main analysis and used a cross-validation regime that estimated and tested observation-related grid orientations in the entorhinal cortex (automatic segmentation, identical to main analysis) across separate task runs (there were four separate runs during the modified navigation task). We estimated grid orientations on even runs (i.e., runs 2 and 4, mean grid orientations estimated across both) and tested grid orientations on the remaining (i.e., runs 1 and 3). Results showed significantly increased grid-like codes during run 2 [mean grid magnitude (arbitrary units, a.u.) \pm s.e.m., 0.03 ± 0.01 ; one-sample t -test, $N = 48$ (excluding one formal outlier); $t(46) = 2.39$, $d = 0.36$, 95% CI = [0.005, 0.6], $p_{\text{one-tailed}} = 0.011$, Bonferroni-corrected for multiple comparisons using a threshold of $\alpha_{\text{Bonferroni}} = 0.05/2 = 0.025$] but not during run 4 ($N = 48$ (excluding one formal outlier), $p_{\text{one-tailed}} = 0.315$). Thus, entorhinal grid-like codes partly generalized across runs during observation, speaking for a certain degree of stability of grid orientations over longer time periods. The partial generalization does not come surprising to us given that the stability of grid orientations might be impacted by several factors, including the potential “remapping” at event boundaries such as the end/beginning of different task runs (Radvansky and Zacks, 2017; Brunec et al., 2018), as already discussed in our manuscript and in our reply above.

Importantly, this analysis does not affect our main results or the conclusions we draw from them in any way since our goal was to probe grid-like codes during observation but not to assess their temporal stability across longer time periods. Our results are solid and were confirmed by a multitude of control analyses, scrutinizing data from different angles, and convincingly highlighting grid-like coding during observation in the human entorhinal cortex.

Action taken:

We added the additional analysis to our updated manuscript and expanded the critical discussion on the longer-term temporal stability of grid orientations.

Results (page 8, line 208): “Grid orientations also partly generalized across separate task runs (Supplementary Results S2).”

Supplementary information (page 2, line 44): “Our main goal was to probe grid-like codes during observation. It has been shown that several factors can affect the fMRI signal over longer time periods (e.g., within one fMRI session) which makes it difficult to compare between separate task runs (see also the Methods section for a more in-depth discussion of this aspect). In brief, these factors include substantial signal drifts of fMRI-based activation patterns over time (...) as well as potential imprecisions in the spatial co-registration between different task runs. Most importantly, grid orientations can “remap” even in the same spatial environment (...) and this could be triggered by the end of one task run and the beginning of the next (indicating an “event boundary”). The precise reason for fluctuations in grid orientations are currently unclear. For these reasons, we explicitly opted for cross-validation within task runs.

Nevertheless, to test whether entorhinal grid-like codes during observation were stable over longer time periods we repeated the main analysis and used a cross-validation regime that estimated and tested observation-related grid orientations in the entorhinal cortex (automatic segmentation, identical to main analysis) across separate task runs (there were four separate runs during the modified navigation task). We estimated grid orientations on even runs (i.e., runs 2 and 4, mean grid orientations estimated across both) and tested grid orientations on the remaining (i.e., runs 1 and 3). Results showed significantly increased grid-like codes during run 2 [mean grid magnitude (arbitrary units, a.u.) \pm s.e.m., 0.03 ± 0.01 ; one-sample t -test, $N = 48$ (excluding one formal outlier); $t(46) = 2.39$, $d = 0.36$, 95% CI = [0.005, 0.6], $p_{\text{one-tailed}} = 0.011$, Bonferroni-corrected for multiple comparisons using a threshold of $\alpha_{\text{Bonferroni}} = 0.05/2 = 0.025$] but not during run 4 ($N = 48$ (excluding one formal outlier), $p_{\text{one-tailed}} = 0.315$). Thus, entorhinal grid-like codes partly generalized across runs during observation, speaking for a certain degree of stability of grid orientations over longer time periods.

The partial generalization does not come surprising to us. Our modified navigation task is novel and entirely different compared to other tasks that are typically used to probe entorhinal grid-like codes during navigation (such as the object-location task)^{4,5}. Participants observed short paths on each trial (consisting of three path segments each) and re-traced them thereafter. The switch between observation/navigation periods and the change from one path to the next could have enforced the perception of “event boundaries”, potentially contributing to “remapping” of grid orientations and generally lower temporal stability. Indeed, while we found no significant difference in the

temporal stability of grid orientations between the observation/navigation periods, temporal stability within runs was generally low (Supplementary Fig. 8). We expect this “remapping” to also be the case as participants complete one task run and start the next run after a short break, which could have contributed to the abovementioned results.”

2. I also struggle with the outlier exclusion approach. 3 SDs is a rather low cutoff, and I still believe a robust test framework would be highly preferable to the variable amounts of outlier exclusions. I simply have never come across a paper that has excluded participants on a per-analysis basis, resulting a different Ns for every other analysis.

Outlier cutoff:

To clarify, we did not define outliers as mean value $\pm 3 \times$ the standard deviation but rather defined them as median value $\pm 3 \times$ the median absolute deviation (MAD). We chose this method because the mean and standard deviation are particularly sensitive to outliers whereas the median is not (Leys et al., 2013). We now highlight this more clearly in the text.

Robust test framework:

We understand the reviewer’s concern. In order to assure that our findings are not driven by our outlier exclusion approach we have repeated all main analyses using a robust test framework that relied on the full data set (i.e., no outlier exclusion). Specifically, we used the one-sample Wilcoxon signed-rank test which is the non-parametric alternative to the one-sample *t*-test. The Wilcoxon signed-rank test is robust against outliers as it assesses whether the median of a given sample significantly differs from a given population median by considering the absolute differences between observed values and a given population mean, ranked by their magnitude. As expected, this analysis yielded virtually identical results, not affecting the findings in any way. Results are summarized in Rebuttal-Table R1 below (same data as in updated Supplementary Tables S1 and S2, which are not shown here).

Action taken:

We included the additional results in our updated supplement (Supplementary Tables S2 and S3). Together with our initial analysis (which already highlighted that our results were stable disregarding the inclusion/exclusion of formal outliers) we now provide a complete and maximally transparent picture of the data, which we think will allow readers to evaluate the robustness of the results with full information.

Methods (page 29, line 1044): *“Grid magnitude values exceeding the median value $\pm 3 \times$ the median absolute deviation (MAD) were excluded from the analyses. We chose this method because the mean and standard deviation are particularly sensitive to outliers whereas the median is not (Leys et al., 2013).”*

Results (page 8, line 199): *“(…) results for this and all following analyses of this section remained stable when using the full data set of $N = 49$ and when using a robust test framework, see also Supplementary Table S2 (…).”*

Results (page 10, line 275): *“(…) as above, results for this and all following analyses of this section remained the same when using the full data set and when using a robust test framework, Supplementary Table S3 (…).”*

(please see next page for Rebuttal-Table R1)

Rebuttal-Table R1: Grid-like codes during observation and navigation.

Grid-like codes (analysis type)	Observation			Navigation		
	Main analysis (excluding formal outliers)	Full data set (no outliers excluded)	Robust analysis (no outliers excluded)	Main analysis (excluding formal outliers)	Full data set (no outliers excluded)	Robust analysis (no outliers excluded)
Entorhinal cortex (ASHS), 6-fold	0.005 *	0.009 (*)	0.005 *	0.233	0.028 (*)	0.034 (*)
Entorhinal cortex (ASHS), 5-fold	0.245	0.146	0.248			
Entorhinal cortex (ASHS), 7-fold	0.489	0.23	0.422			
Entorhinal cortex (ITK-SNAP), 6-fold	0.032 (*)	0.032 (*)	0.06 (~)	0.323	0.191	0.487
Entorhinal cortex (ITK-SNAP), 5-fold	0.102	0.293	0.866			
Entorhinal cortex (ITK-SNAP), 7-fold	0.178	0.103	0.866			
Hippocampus, 6- fold	0.271	0.283	0.628	0.34	0.166	0.568
Parahippocampal cortex, 6-fold	0.264	0.196	0.583	0.087	0.162	0.818
Anterior thalamus, 6-fold	0.159	0.241	0.818	0.016 (*)	0.118	0.042(*)
V1, 6-fold	< 0.0001 *	< 0.0001 *	< 0.0001 *	0.222	0.118	0.852

Note: Results of the grid code analysis based on two entorhinal cortex ROIs (defined through ASHS and manually segmented using ITK-SNAP) and four control ROIs (hippocampus, parahippocampal cortex, anterior thalamus, V1) during observation/navigation and using three complementary analysis approaches (1: initial analysis using one-sample *t*-test and excluding formal outliers, 2: initial analysis using one-sample *t*-tests based on the full data set, 3: robust analysis using the Wilcoxon signed-rank test based on the full data set). Results were corrected for multiple comparisons using Bonferroni-correction ($\alpha_{\text{Bonferroni}} = 0.05/6 \text{ ROIs} = 0.008$). (*) $p < 0.05$ but not surviving Bonferroni-correction, (~) tendency for $p < 0.05$. Numbers indicate *p*-values. The main result is shown in yellow.

3. I am not sure if the additional analyses really rule out that there is a (possibly visual) confound that increases the grid-like signal strength during observation. However, the authors have quite exhaustively tried to account for possible confounds (due to eye movements) and it doesn't seem like there are any additional control analyses that offer themselves. I nevertheless suggest the authors to use nuanced and toned-down language, in particular in the abstract. The absence of grid-like signals during navigation, the negative relation to behavior and the negative V1 grid like signals should be mentioned, and the difficulties arising for the interpretation of the entorhinal signals should be presented more prominently too.

We made sure to adopt a more nuanced and toned-down language wherever appropriate. First, we paid particular attention to the abstract and now also mention the negative relationship between entorhinal grid-like codes during observation and behavioral performance.

Abstract (page 2, line 25): "The activity of grid-like codes was time-locked to increases in co-activation and entorhinal-cortical connectivity that included the striatum, the hippocampus, parahippocampal and right posterior parietal cortices. Surprisingly, the grid-related effects during observation were stronger the worse participants performed when subsequently re-tracing the demonstrator's paths."

Second, we made sure to mention the unexpected results more prominently.

Discussion (page 16, line 482): *“The study yielded several surprising findings as well: we found significantly negative grid-like codes in V1 during observation and we did not detect significant grid-like codes during navigation periods.”*

Also, we would like to highlight the thorough discussion on these aspects that we had already included in our manuscript:

Discussion (page 18, line 588): *“Furthermore, we found a strong effect of negative grid-like codes in V1 during observation (Fig. 3e, Supplementary Fig. S5). V1 activation levels were increased for observed directions that were misaligned with the individual V1 grid orientation (0 modulo 30°) compared to aligned directions (0 modulo 60°). Multiple explanations of how grid cell firing activity relates to macroscopic (fMRI-based) signals exist (...). For example, one possibility is that grid cells repeatedly fire during aligned movement and that this repetition would cause neural adaption (leading to a relative increase in the fMRI signal for misaligned directions). Signals might potentially be triggered by grid cells in the visual system (although it is currently unclear whether grid cells exist in V1) or by (entorhinal) mechanisms upstream. At present, we do not know which factors drive this effect and thus cannot provide a firm interpretation of why negative grid-like coding in V1 appears associated with tracking others navigating through space. We encourage future research to elucidate this finding, as well as the relationship between grid-like codes in the medial temporal lobe and visual systems.”*

And further:

Discussion (page 18, line 600): *“Contrary to previous work (...), we did not find significantly increased entorhinal grid-like codes during self-navigation (Fig. 3g; see Supplementary Results S2 for numerically increased, as well as significantly decreased grid-like codes in the right and left entorhinal cortex, respectively, and see Supplementary Fig. S3 for significant grid-like coding in the anterior thalamus). The spatial stability of entorhinal grid-like codes was significantly decreased during navigation compared to observation periods (Fig. 3h), indicating that grid orientations in the entorhinal cortex were less clustered towards a specific direction. Grid orientations hence displayed larger variability across voxels, which might have resulted in an overall decrease in grid magnitude values during navigation. The finding might also be explained by the specific setup of our modified navigation task, which differed from previous setups in several points. Participants were passively moved (after indicating and confirming their intended walking direction with a button press) and were required to re-trace path trajectories rather than freely navigating towards object locations in virtual reality space. Also, participant’s viewpoints were directly placed behind the demonstrator’s starting positions (but note that these randomly varied across trials). This design aspect could have enforced egocentric (striatal) rather than allocentric (hippocampal) processing (...), and could explain the striatal co-activation and connectivity time-locked to observation-related entorhinal grid-like codes. We speculate that decoupling the participant’s viewpoint from the starting position might have rendered the task more hippocampal dependent (...). In contrast to this argument, however, it appears that participants formed an allocentric representation since we detected entorhinal grid-like codes during observation, and since results showed generally increased activation levels in the hippocampus and entorhinal cortex during navigation (Fig. 2c). Future research might resolve this issue by expanding our current task-setup and decoupling view- and starting points (...).”*

4. I feel this point about the social aspects could be addressed better. The authors did tone down the language they used; however, the social aspect is still emphasized quite strongly throughout the manuscript. I find this problematic, given that the experiment lacks an appropriate non-social control condition that would allow to establish whether the effects found are specific to the ‘social’ aspect of the task.

We completely agree with the reviewer that we are unable to establish whether our findings are specific to “social” processing as our modified navigation task does not include a non-social control condition. We would like to reiterate that we mentioned this already in our first version of the manuscript, which we think quite explicitly makes the reader aware that the conclusions regarding social specificity need further evidence.

Discussion (page 19, line 643): *“Second, while it is plausible that entorhinal grid-like codes support socio-spatial navigation, we would like to emphasize that we are unable to make claims about the social specificity of our results. Entorhinal grid-like codes during observation might also be triggered by non-social features that need to be tracked (such as moving cars when crossing the road) and might be related to feature relevance (keeping an eye on the moving car is important to cross the road safely). While such results would speak for a general role of entorhinal grid-like codes in tracking moving features (...), the debate on the social specificity of brain processes (...) is fueled by initial evidence for “social” and “non-social” place cells (...). Omer and colleagues dissociated signals by either presenting another individual (a demonstrator bat) or an object (a football) moving through space while the observer bat remained stationary. A similar task design might be helpful to resolve this issue in follow-up work with*

humans. Hence, we cannot draw firm conclusions regarding the involvement of entorhinal grid-like coding specifically in socio-spatial navigation but consider our work an important first step in this direction.”

Additionally, we went over the manuscript once more and made sure to adopt a more careful phrasing throughout.

Abstract (page 2, line 19): “Navigating through crowded, dynamically changing environments requires the ability to keep track of other individuals.”

Abstract (page 2, line 29): “This suggests that network dynamics time-locked to entorhinal grid-cell-related activity might serve to distribute information about the location of others throughout the brain.”

Discussion (page 16, line 471): “In the current study, we investigated whether grid-like codes in the human entorhinal cortex supported participants as they tracked and subsequently re-traced the paths of a virtual demonstrator.”

Discussion (page 20, line 662): “We suggest that grid-like codes and their associated network dynamics could serve to distribute information about the location of others throughout the brain, laying the foundation for an internal “compass” that enables us to maneuver through crowded and dynamically changing environments as we encounter them in everyday situations.”

References

- Alink A, Walther A, Krugliak A, Bosch JJF van den, Kriegeskorte N (2015) Mind the drift - improving sensitivity to fMRI pattern information by accounting for temporal pattern drift. bioRxiv:032391–032391.
- Brunec IK, Moscovitch M, Barense MD (2018) Boundaries Shape Cognitive Representations of Spaces and Events. Trends in cognitive sciences 22:637–650.
- Doeller CF, Barry C, Burgess N (2010) Evidence for grid cells in a human memory network. Nature 463:657–661.
- Kunz L, Schroder TN, Lee H, Montag C, Lachmann B, Sariyska R, Reuter M, Stirnberg R, Stocker T, Messing-Floeter PC, Fell J, Doeller CF, Axmacher N (2015) Reduced grid-cell-like representations in adults at genetic risk for Alzheimer’s disease. Science 350:430–433.
- Leys C, Ley C, Klein O, Bernard P, Licata L (2013) Detecting outliers: Do not use standard deviation around the mean, use absolute deviation around the median. Journal of Experimental Social Psychology 49:764–766.
- Radvansky GA, Zacks JM (2017) Event boundaries in memory and cognition. Current Opinion in Behavioral Sciences 17:133–140.
- Stangl M, Wolbers T, Shine JP (2019) Population-Level Analysis of Human Grid Cell Activation. In, pp 257–279. Humana, New York, NY. Available at: http://link.springer.com/10.1007/7657_2019_27.